# Do degree and rate of silicate weathering depend on plant productivity?

Ralf A. Oeser[1], Friedhelm von Blanckenburg[1,2]

[1] GFZ German Research Centre for Geosciences, Section Earth Surface Geochemistry, Potsdam, D-14473, Germany
[2] Freie Universität Berlin; Institute of Geological Science, Berlin, D-12249, Germany

5  *Correspondence to*: Ralf A. Oeser (oeser@gfz-potsdam.de)

**Abstract.** Plants and their associated below-ground microbiota possess the tools for rock weathering. Yet the quantitative evaluation of the impact of these biogenic weathering drivers relative to abiogenic parameters, such as the supply of primary minerals, of water, and of acids is an open question in Critical Zone research. Here we present a novel strategy to decipher the relative impact of these drivers. We 10  quantified the degree and rate of weathering and compared these to nutrient uptake along the "EarthShape" transect in the Chilean Coastal Cordillera. These sites define a major north-south gradient in precipitation and primary productivity but overlie granitoid rock throughout. We present a dataset of the chemistry of Critical Zone compartments (bedrock, regolith, soil, and vegetation) to quantify the relative loss of soluble elements (the "degree of weathering") and the inventory of bio-available elements. 15  We use $^{87}Sr/^{86}Sr$ isotope ratios to identify the sources of mineral nutrients to plants. With rates from cosmogenic nuclides and biomass growth we determined fluxes ("weathering rates"), meaning the rate of loss of elements out of the ecosystems, averaged over weathering timescales (millennia), and quantified mineral nutrient recycling between the bulk weathering zone and the bulk vegetation cover. We found that neither the degree of weathering nor the weathering rates increase systematically with precipitation 20  from north to south along the climate and vegetation gradient. Instead, the increase in biomass nutrient demand is accommodated by faster nutrient recycling. In the absence of an increase in weathering rate despite a five-fold increase in precipitation and NPP, we hypothesize that plant growth might in fact dampen weathering rates. Because plants are thought to be key players in the global silicate weathering - carbon feedback, this hypothesis merits further evaluation.

 **1 Introduction**

Ever since the emergence of land plants, their dependence on mineral-derived nutrients has impacted rock weathering (used here to mean the combined processes of primary mineral dissolution, secondary solid formation, and the loss of elements in aqueous solution). This impact results from three types of interaction. The first is mechanical processes, that weaken rock or change the depth of the weathering zone through roots and microbial symbionts (e.g. mycorrhizal fungi; Blum et al., 2002; Brantley et al., 2017; Hasenmueller et al., 2017; Minyard et al., 2012; Quirk et al., 2014; van Schöll et al., 2007). The second is a variety of biogeochemical processes that alter the susceptibility of minerals to weathering. These mechanisms include root respiration that releases protons and $CO_2$, lowering soil pH, the exudation of organic ligands through roots that increases the solubility of nutrients through complexation, and the uptake, uplift, and recycling of pore fluids and nutrients from solution (e.g. Berner et al., 2003; Brantley et al., 2012; Drever, 1994; Kump et al., 2000; Lee and Boyce, 2010; Jobbágy, 2001; Giehl and von Wiren, 2014). The third interaction affects the water cycle which is impacted by rooting depth and seasonal water storage in saprolite and evapotranspiration (Kleidon et al., 2000; Ibarra et al., 2019). All of these interactions impact weathering, either directly by aiding plant acquisition of mineral nutrients from rock, or indirectly by modifying the water cycle (e.g. Brantley et al., 2011; Porder, 2019; Moulton et al., 2000). This means the presence and growth rate of land plants is commonly thought to have strongly impacted the evolution of Earth's atmosphere over geologic time by strengthening the negative feedback between silicate weathering rates and atmospheric $CO_2$ concentrations (Beerling and Berner, 2005; Doughty et al., 2014; Lenton et al., 2012; Pagani et al., 2009; Porada et al., 2016).

While biota in general and plants in particular are undoubtedly key players in weathering and pedogenesis, a quantitative evaluation of their impact remains elusive. The reason is our inability to disentangle abiotic from biotic processes in field observations (Amundson et al., 2007). Almost all mass transfer in the weathering zone can have biotic and abiotic causes. An additional challenge is the difficulty in accounting for confounding effects. Environmental state variables shaping the Critical Zone (the zone of the Earth surface that extends from the top of unweathered bedrock to the top of the vegetation cover; the zone in which most biogeochemical reactions take place) can obscure or amplify the effects of biology, making the attribution of cause and effect challenging. Another reason for our inability to directly attribute

weathering to plant growth arises from the ability of ecosystems to recycle nutrients through microbial mineralization from plant litter and organic matter, rather than acquiring fresh nutrients from rock

(Chaudhuri et al., 2007; Lang et al., 2016; Lucas, 2001; Spohn and Sierra, 2018; Wilcke et al., 2002). Given the ability of ecosystems to buffer changes in nutrient fluxes (Spohn and Sierra, 2018) the dependence of weathering on plant growth and biomass distribution can be expected to be a highly non-linear one.

A classical strategy in field studies that aim to decipher how ecosystem functioning and weathering shape
the Critical Zone relies on exploring the interactions along natural environmental gradients. Studies along a Hawaiian chronosequence (soils of variable discrete initial formation age) have evaluated the role of soil age in weathering and the distribution and cycling of cations through plants. These studies revealed the dependency of nutrient cycling on the degree of weathering (e.g. Bullen and Chadwick, 2016; Chadwick et al., 1999; Laliberte et al., 2013; Porder and Chadwick, 2009; Vitousek, 2004). Studies along
a climosequence (gradients in climate whilst minimizing other environmental differences) have evaluated the effect of climate on ecological and pedogenic processes (Bullen and Chadwick, 2016; Calmels et al., 2014; Dere et al., 2013; Egli et al., 2003; Ferrier et al., 2012). These studies generally show an increase in weathering rates with increasing mean annual temperature (MAT) and mean annual precipitation (MAP), while vegetation plays a significant role in pedogenesis. Studies across different rock substrates
have evaluated the availability of nutrients and the dissolution kinetics of minerals for ecosystem nutrient budgets (Hahm et al., 2014; Uhlig and von Blanckenburg, 2019) and indicate a 'bottom-up' lithological and mineralogical control on nutrient availability to ecosystems. Studies along gradients in erosion rates explored the supply of minerals to ecosystems and discovered an increase in nutrient supply through weathering with increasing erosion rates (Chadwick and Asner, 2016; Eger et al., 2018; Porder et al.,
2007; Schuessler et al., 2018). Studies that have tried to isolate just the role of vegetation cover show that the weathering fluxes in adjacent areas in which only vegetation differs showed higher fluxes with more vegetation (Moulton et al., 2000). All these studies differ widely in their methodology, time scale, spatial scale, conceptual framework and even discipline. We return to this topic below by comparing our conceptual perspective to other approaches.

In this study we explore weathering, nutrient uptake, and nutrient recycling along one of the Earth's most impressive climate and vegetation gradients, located in the Chilean Coastal Cordillera (Oeser et al., 2018) Along this gradient we quantify the degree of weathering, (using chemical analyses of rock and regolith, Oeser et al., 2018), rates of weathering (using cosmogenic $^{10}$Be, Schaller et al., 2018), and nutrient uptake (using net primary productivity, NPP and the chemical composition of the major plant species at each site). Sequential extraction protocols applied to bulk regolith were used to identify the stoichiometry of the main plant-available elements in the regolith. Radiogenic $^{87}$Sr/$^{86}$Sr isotope ratios in bulk rock, regolith, the bio-available fraction in regolith (where regolith is used here to mean the sum of consolidated and unconsolidated material above the weathering front, including soil), and plant biomass were used to identify the sources of mineral nutrients. We were thus able to identify gains and losses of mineral nutrients in and out of these ecosystems and to quantify the efficiency of nutrient recycling. We applied the conceptual framework and parameterization of Uhlig and von Blanckenburg (2019) to place quantitative constraints on the "organic nutrient cycle" and the "geogenic nutrient pathway" as detailed in the next section. In a companion paper we exploited $^{88}$Sr/$^{86}$Sr stable isotope fractionation in the materials studies here and established a stable isotope-based mass balance for Sr cycling in the critical zone including plants (Oeser and von Blanckenburg, 2020a).

Here, we specifically evaluated the following questions: (1) Do weathering rates increase along the north-south precipitation gradient because runoff, the main driver of weathering flux, increases? (2) Do the variations in NPP along the climate and vegetation gradient correlate with nutrient supply rates from weathering?

## 2. Conceptual perspectives

Two fundamentally different concepts describe the relationship between regolith formation and time, and their relationship to different geomorphic regimes (Lin, 2010; Smeck et al., 1983): the continuous evolution model and the steady-state model. The continuous evolution model describes regolith or soil evolution with time from an initial point and describes chronosequences, where soils evolve on stable (non-eroding) surfaces. These soils have a distinct age and undergo several phases of soil development (e.g. Chadwick et al., 1999; Vitousek and Chadwick, 2013). In contrast, the steady-state model assumes

all regolith state variables are independent of an initial point. In this concept, regolith is constantly rejuvenated by production at depth and its removal through erosion from above (e.g. Heimsath et al., 1997). In other words, the regolith is continuously turned over and has no distinct age, but rather a residence time. This concept applies to all sloping landscapes on Earth, on which typical regolith residence times ($\leq 10^4$ yrs) are often less than or equal to the timescales over which tectonics and climate vary ($\geq 10^4$ yrs). This suggests that much of the Earth surface operates in a manner that is consistent with the steady-state model of soil formation (Dixon et al., 2009; Ferrier et al., 2005; Riebe and Granger, 2013). The state variables do not necessarily vary linearly with age (in the continuous evolution model) or residence time (in the steady-state model). Thus, in the continuous evolution model, pedogenic thresholds have been deduced based on certain soil properties (Dixon et al., 2016; Vitousek and Chadwick, 2013). These have also been described to exist and strongly vary along the eroding surfaces in Chile explored in this study (Bernhard et al., 2018).

Although ecosystems respond over shorter time scales to environmental change, ranging from seasonal to decadal or longer climate cycles, their evolution can nevertheless be linked to the two regolith evolution models (Brantley et al., 2011). In the continuous evolution model, ecology and soil development are linked via progressive increases in soil stability and water retention capacity and a unidirectional decrease in mineral nutrient availability (Vitousek and Farrington, 1997). In contrast, in the steady-state model, regolith replenishment by uplift and erosion sets the upward advection of mineral nutrients (Buendía et al., 2010; Porder et al., 2007; Vitousek et al., 2003; Uhlig and von Blanckenburg, 2019) and availability of regolith moisture (Rempe and Dietrich, 2018). Thus, the combination of regolith residence time and mineral weathering rates determines whether supply by a specific mineral nutrient suffices to sustain an ecosystem over weathering time scales which in turn is thought to impact plant diversity and nutrient acquisition strategies. For example, ecosystems on strongly mineral nutrient-depleted soils seem to be characterized by high plant diversity (Laliberte et al., 2013; Lambers et al., 2008). Note that nitrogen (N), the most limiting nutrient in many ecosystems, is not an element addressed here. Although rocks have recently received attention as source of geogenic N (Houlton et al., 2018) this source is most prominent in sedimentary rock. This study explores ecosystems developed on granitoid rock where N is derived

from the inexhaustible atmospheric pool by nitrogen-fixing bacteria, and limitation mostly arises by the energy required for fixation (Chapin III et al., 2011).

The methods employed to explore these processes span a range of time scales that are discipline specific. Plant ecology typically works on (sub-)annual timescales for ecosystem fertilization or manipulation experiments (Tielbörger et al., 2014; Tipping et al., 1999), while instrumental monitoring of water, gas, and nutrient fluxes between Critical Zone compartments in hydrology, soil ecology, and biogeochemistry can reach decadal timescales (Joos et al., 2010; Kelly and Goulden, 2016; Lang et al., 2016; Sprenger et al., 2019; Sohrt et al., 2019; Wilcke et al., 2017). Geochemical estimates of rock weathering or evolution of plant-available nutrient inventories typically integrate over millennial timescales (Buendía et al., 2010; Porder et al., 2007; Riebe and Granger, 2013; Uhlig and von Blanckenburg, 2019; Vitousek et al., 2003). To integrate these different time scales, some soil ecological models account for the coupled weathering – recycling – uptake systems by linking the short-term, biological cycle with the long-term, largely geological and hydrological-driven cycle (Porder and Chadwick, 2009; Powers et al., 2015; Vitousek et al., 1998). Such models have recently been complemented by concepts and methods from geochemistry (Uhlig and von Blanckenburg, 2019) that we pursue in this study. In this conceptual framework, the so-called "organic nutrient cycle" comprises a set of strategies for efficient nutrient re-utilization through microbial mineralization from plant litter and organic matter and entails rapid nutrient turnover. The "geogenic nutrient pathway" compensates the loss of nutrients by erosion and in solution through the slow but steady supply of nutrients from chemical weathering of rock (Buendía et al., 2010; Cleveland et al., 2013; Uhlig and von Blanckenburg, 2019). This concept is particularly relevant where atmospheric wet and dry deposition (e.g. Boy and Wilcke, 2008; Chadwick et al., 1999; Dosseto et al., 2012) do not suffice to balance the losses. These geogenic input fluxes are often minor compared to those in the organic nutrient cycle and may even be undetectable over the annual to decadal scales of ecosystem monitoring experiments. However, they sustain ecosystem nutrition over longer (decadal to millennial) time scales because they prevent mineral nutrient deficiency that may otherwise develop (Hahm et al., 2014; Schuessler et al., 2018; Uhlig et al., 2017; Uhlig and von Blanckenburg, 2019). Whether the geogenic nutrient pathway is sufficient to prevent development of mineral nutrient limitation over the millennial scale depends on the rate of supply of fresh rock into the weathering zone, the bio-availability of the

nutrients released, and whether plant roots and the associated mycorrhizal fungi can access them. Thus, any exploration of these links must constrain where nutrients are released in regolith relative to where plants obtain them. The aim of this study is to illustrate how this approach from geochemistry can be 165 employed to assess the flux balances between the top of bedrock and the top of the vegetation canopy as integrated over millennia, and how plant growth affects these in comparison to the geologic drivers like uplift and erosion or climatic drivers like precipitation and runoff.

**3 Study area and previous results**

The study was conducted within the Critical Zone project "EarthShape: Earth Surface Shaping by Biota". 170 The four study sites are part of the EarthShape study area which is located along the Chilean Coastal Cordillera. Three sites are located in national parks and one in a nature reserve, so human impact is minimized. The sites are located on the plutonic rocks of the Chilean Coastal Cordillera and are close to the Pacific coast (less than 80 km; Oeser et al., 2018). Two previous studies introduced the field area, its pedogenic and weathering characteristics, and a set of new soil- and geochemical data (Oeser et al., 2018; 175 Bernhard et al., 2018).

The sites define a vegetation gradient controlled by climate, ranging over 1300 km. From north to south, they cover arid (Pan de Azúcar National Park, ~26°S), semi-arid (Santa Gracia Nature Reserve, ~30°S), mediterranean (La Campana National Park, ~33°S), and humid-temperate (Nahuelbuta National Park, ~38°S) climate conditions. The mean annual precipitation (MAP) increases from 10 mm $yr^{-1}$ in Pan de 180 Azúcar, 89 mm $yr^{-1}$ in Santa Gracia, 440 mm $yr^{-1}$ in La Campana, to 1100 mm $yr^{-1}$ in Nahuelbuta, respectively. The mean annual air temperature (MAT) ranges from 18.1°C in the northernmost site in Pan de Azúcar to 14.1°C in the southernmost site in Nahuelbuta (Fig. 1, Table 1; Ministerio de Obras Públicas, 2017).

Net primary productivity (NPP), derived from a dynamic vegetation model (LPJ-GUESS) that simulates 185 vegetation cover and composition during the Holocene (Werner et al., 2018), ranges from 30 g $m^{-2}$ $yr^{-1}$ C and 150 g $m^{-2}$ $yr^{-1}$ C in the arid shrubland of Pan de Azúcar and Santa Gracia, respectively, to 280 g $m^{-2}$ $yr^{-1}$ C in the sclerophyllous woodland of La Campana, and is highest (520 g $m^{-2}$ $yr^{-1}$ C) in the humid-temperate forests of Nahuelbuta (Fig. 1, Table 1). The vegetation cover (< 5%) in Pan de Azúcar

consists only of small shrubs, geophytes and annual plants (Armesto et al., 1993), which are mainly present in  small ravines. The vegetation in Santa Gracia belongs to the "Interior Mediterranean desert scrub of *Heliotropium stenophyllum* and *Flourensia thurifera*" formation (Luebert and Pliscoff, 2006). Plants are affected by livestock grazing (mostly goats; Bahre, 1979), and vegetation cover is generally sparse. In La Campana the vegetation (almost 100% ground cover) is part of the "Coastal Mediterranean sclerophyllous forest of *Lithraea caustica* and *Cryptocarya alba*" formation (Luebert and Pliscoff, 2006). The dominant vegetation in Nahuelbuta is associated with the "Coastal temperate forest of *Araucaria araucana*" formation (Luebert and Pliscoff, 2006) and covers 100% of ground area. Ecosystems at all sites are primarily nitrogen-limited (Stock et al., 2019).

The basement at those sites is mainly composed of granitoid intrusions of late Carboniferous to Cretaceous age. The compositional variation ranges from monzo- to syenogranites in Pan de Azúcar (199 Ma; Berg and Breitkreuz, 1983; Berg and Baumann, 1985; Parada et al., 2007), pyroxene and hornblende-bearing diorites and monzodiorites in Santa Gracia (98 – 89 Ma; Moscoso et al., 1982), as well as tonalites and granodiorites in Nahuelbuta (Nahuelbuta complex, 294 Ma; Parada et al., 2007) and in the Caleu Pluton in La Campana with an intrusion age of 130 Myr (Molina et al., 2015; Parada and Larrondo, 1999; Parada et al., 2002).

For the soil pits studied here, denudation rates inferred from cosmogenic nuclides (*in situ* [10]Be), interpreted as soil production rates, are $8 - 11$ t km$^{-2}$ yr$^{-1}$ in Pan de Azúcar, $16 - 22$ t km$^{-2}$ yr$^{-1}$ in Santa Gracia, $54 - 69$ t km$^{-2}$ yr$^{-1}$ in La Campana and $18 - 48$ t km$^{-2}$ yr$^{-1}$ in Nahuelbuta (Schaller et al., 2018b). Catchment-wide denudation rates broadly agree with these soil-scale rates, except in La Campana. Here, they are higher, attributed to debris flows in valley tops due to the higher channel steepness than elsewhere (e.g. mean slope 23° in La Campana, 9° in Nahuelbuta; van Dongen et al., 2019). The relative consistency of these rates along the climate gradient is ascribed to consistent tectonic forces acting along the whole gradient (e.g. Blanco-Chao et al., 2014; Melnik, 2016), with the moderate increase in denudation rates at the two southern sites explainable with the combined effect of higher precipitation and increasing shielding by vegetation (Schaller et al., 2018b).

The architecture of the regolith profiles, their chemistry, mineralogy, and the physical properties of soils, saprolite, and the rocks beneath have been extensively described for four soil pits at each site by Bernhard

et al. (2018), Dal Bo et al. (2019), Oeser et al. (2018) and Schaller et al. (2018b). The regolith profiles in Pan de Azúcar are located between 330 and 340 m above sea level (asl) on steep (25 – 40°; Table 1) hill slopes. The soils on the north- and south-facing slopes were classified by Bernhard et al. (2018) as Regosols with only shallow A and B horizons of ~20 – 30 cm thickness, lacking any kind of organic and litter layer. In this area, the processes disintegrating rock and developing regolith are mainly physical weathering, specifically a combination of insolation- and salt weathering (Oeser et al., 2018). The regolith profiles in Santa Gracia are situated at almost 700 m asl on gently sloping hills (15 – 25°; Table 1). The soils on the north- and on the south-facing slope are a Leptosol and a Cambisol, respectively (Bernhard et al., 2018). Distinct O-horizons and a litter layer are not apparent. The Ah horizons in both profiles reach depths of 10 cm and the transition from the mineral soil (Bw) into saprolite occurs at 25 – 30 cm depth. Oeser et al. (2018) attribute this sites' high degree of elemental depletion (50% loss relative to bedrock as quantified by the chemical depletion fraction CDF; Fig. A1; Data Table S2) despite low precipitation to the low abundance of quartz and the high abundance of readily weatherable plagioclase and mafic minerals. The regolith profiles in La Campana, located at 730 m asl and on gently sloping hills (12 – 23°), are classified as Cambisols. The O-horizon is ~5 cm thick and is followed by a Ah horizon, extending up to 40 cm depth (Bernhard et al., 2018). Here, the mineral-soil layer turns into saprolite at approximately 110 cm in both profiles (Table 1). The elemental depletion of Ca relative to bedrock increases from ~45% at the profiles' bottom towards ~70% at their top and can be classified as depletion (north-facing) or depletion and enrichment profiles (south-facing, Fig. A1; Table S2; Brantley and Lebedeva, 2011), respectively. The regolith profiles in Nahuelbuta are situated on gently sloping hills (~15°) at about 1200 m asl (Table 1). Bernhard et al. (2018) have classified the soils on the north- and south-facing slope as Umbric Podzols and Orthodystric Umbrisols, respectively. Here, the Ah horizons measure up to 50 cm (greater thickness on the south-facing slope) and are overlain by an organic layer of 5.5 cm thickness. In the two regolith profiles, the soil-saprolite transition is at 100 and 120 cm depth, respectively. The coarse-grained saprolite disaggregates readily. These two profiles are characterized by highly heterogeneous weathering patterns caused by the incorporation of the metamorphic basement at various parts (e.g. Oeser et al., 2018; Hervé, 1977). Along the EarthShape north-south transect, many of the soil properties indicate crossing of several distinct pedogenic thresholds (Bernhard et al., 2018). We

note that while the detailed geochemical work reported in this study is based on two profiles per site, the soil properties (Bernhard et al., 2018) and bulk geochemical data (Oeser et al., 2018) of these profiles are corroborated by two additional replicates per site as reported in these previous studies. A comprehensive summary of the characteristics of the eight regolith profiles and major plant types is given in Table 1.

## 4 Methods

### 4.1 Sampling


Regolith samples were collected in a continuous sequence of depth increments from bottom to top. Increments amount to a thickness of 5 cm for the uppermost two samples, 10 cm for the 3rd sample from top, and increase to 20 cm thickness for the 4th sample onwards. To account for the dependence on solar radiation, two regolith profiles on adjacent hillslopes (north- and south-facing) were sampled at each

study site (see Appendix B for further information on sample replication).

The underlying unweathered bedrock was not reached in any of the regolith profiles and the depth to bedrock remains unknown. Thus, bedrock samples were collected from nearby outcrops. This sample set comprises the 20 bedrock samples already reported in Oeser et al. (2018) and 15 additional bedrock samples (in total 12 in Pan de Azúcar, 8 in Santa Gracia, 10 in La Campana, and 5 in Nahuelbuta).

Vegetation samples from representative shrubs and trees (grasses have been excluded) of each study site were sampled in the austral summer to autumn 2016. The sample set comprises material from mature plants of the prevailing species: *Nolana mollis* (Pan de Azúcar), *Asterasia* sp., *Cordia decandra*, *Cumulopuntia sphaerica*, and *Proustia cuneifolia* (Santa Gracia), *Aristeguietia salvia*, *Colliguaja odorifera*, *Cryptocarya alba*, and *Lithraea caustica* (La Campana), *Araucaria araucana*, *Nothofagus*

*antarctica*, and *Chusquea coleu* (Nahuelbuta). From each sampled plant (n = 20), multiple samples of leaves, twigs and stem were collected, pooled together, and homogenized prior to analysis. These samples were either taken using an increment borer (stem samples) or plant scissors (leaf and twig samples) equipped with a telescopic arm to reach the higher parts of trees. Roots could not be sampled in a representative manner though we account for their influence on plant composition (see Appendix A). The

litter layer, comprising recently fallen leaves (from within the last two years) and small woody debris, was also sampled in La Campana and Nahuelbuta.

## 4.2 Analytical methods

### 4.2.1 Chemical composition of regolith and bedrock

The concentration of major and trace elements in bedrock and regolith samples were determined using a
X-Ray Fluorescence spectrometer (PANalytical AXIOS Advanced) at the section for "Inorganic and Isotope Geochemistry", GFZ German Research Centre for Geosciences. A detailed description of the analytical protocols and sample preparation is given in Oeser et al. (2018).

### 4.2.2 Chemical composition of vegetation

Major and trace element concentrations of vegetation samples were determined using a Varian 720-ES
axial ICP-OES at the Helmholtz Laboratory for the Geochemistry of the Earth Surface (HELGES), GFZ German Research Centre for Geosciences (von Blanckenburg et al., 2016) with relative uncertainties smaller than 10%. Prior to analysis, all samples were oven-dried at 120°C for 12 hrs. Subsequently, leaves were crushed and homogenized. About 0.5 g of leaf and 1 g of woody samples were digested in PFA vials using a microwave (MLS start) and ultra-pure concentrated acid mixtures comprising $H_2O_2$ and $HNO_3$,
HCl and $HNO_3$, and HF. In some plant samples Si-bearing precipitates formed upon evaporation after digestion. These sample cakes were redissolved in a mixture of concentrated HF and $HNO_3$ to ensure complete dissolution of Si prior to analysis. As some Si might have been lost by volatilization as $SiF_4$ in this process, we do not include these samples (indicated by a * in Data Table S5) for the compilation of the plants' Si budget. With each sample batch, the international reference material NIST SRM 1515 Apple
leaves and a procedural blank were processed (Data Table S5).

### 4.2.3 Extraction of the bio-available fraction and its chemical analyses

The bio-available fraction of regolith samples was extracted using a sequential extraction procedure adapted from Arunachalam et al. (1996), He et al. (1995), and Tessier et al. (1979). The sequential extraction was performed in parallel on two regolith aliquots, and the supernatants were pooled together

for analyses. About 2 g of dried and sieved (< 2 mm) sample material were immersed in 14 ml 18 MΩ deionised $H_2O$ (water-soluble fraction) and then in 1M $NH_4Oac$ (exchangeable fraction; maintaining a sample:reactant ratio of ca. 1:7), and gently agitated. After each extraction, the mixture was centrifuged for 30 min at 4200 rpm (3392 g) and the supernatant was pipetted off. The remaining sample was then rinsed with 10 ml deionised $H_2O$ and centrifuged again (4200 rpm, 3392 g, 30 min) and the rinse solution added to the supernatant. Subsequently, the supernatants were purified using a vacuum-driven filtration system (Millipore®; 0.2 μm acetate filter), evaporated to dryness, and redissolved with ultra-pure concentrated acid mixtures comprising $H_2O_2$, $HNO_3$, and $HCl$. With each sample batch, international reference materials (NIST SRM 2709a San Joaquin soil, CCRMP TILL-1) along with a procedural blank were processed.

The water-soluble fraction is comprised of elements contained in soil water in the form of free ions and ions which form complexes with soluble organic matter. It represents the most labile soil compartment and thus is most accessible to plants (e.g. He et al., 1995). This fraction was accessed by suspending the samples for 24 hrs in deionised $H_2O$ at room temperature. The exchangeable fraction comprises elements that form weak electrostatic bonds between the hydrated surfaces of phyllosilicates (i.e. clays and micas), oxyhydroxide minerals (e.g. boehmite, diaspore, goethite, lepidocrocite, ferrihydrite), and organic matter. This fraction was extracted by suspending the samples in a mechanical end over end shaker at room temperature in 1 M $NH_4OAc$ for 2 hrs at 60 rpm (0 g). Note that none of the further extraction steps described in Tessier et al. (1979) have been applied to the regolith samples as they are believed to make a negligible contribution to the bio-available fraction.

The element concentrations of the water-soluble and exchangeable fraction were determined using a Varian 720-ES axial ICP-OES at HELGES, following the analytical procedures described in Schuessler et al. (2016) with relative uncertainties estimated at smaller than 10%. Soil-P fractions were determined by Brucker and Spohn (2019). In this case, the bio-available fraction refers to the inorganic products of the modified Hedley sequential P fractionation method of Tiessen and Moir (1993), specifically the water-extractable $P_i$ and labile $P_i$ which was extracted by using 0.5 M $NaHCO_3$.

### 4.2.4 $^{87}$Sr/$^{86}$Sr isotope ratios

The radiogenic Sr isotope ratio was determined on bulk bedrock and regolith, the bio-available fractions of saprolite and soil, and on the different plant organs at each study site.

After sample digestion (bulk samples) or sequential extraction (bio-available fraction), Sr was separated from matrix elements using 200 µl Sr-Spec resin. Matrix elements were removed by elution with 2.5 ml 3 M and 2 ml 7.5 M HNO$_3$. Subsequently, Sr was eluted with 4 ml of 18 Ω deionised H$_2$O. Any organic crown-ether which has been released from the Sr-spec resin was removed after evaporation and subsequent redissolution of the Sr fraction in 1 ml of a 1:1 mixture of concentrated H$_2$O$_2$ and HNO$_3$. This mixture was cooked in a tightly closed beaker at 150°C for at least 12 hrs. Within each sample batch, a minimum of one standard reference material and a procedural blank were processed.

$^{87}$Sr/$^{86}$Sr was measured in a 50-ng g$^{-1}$ pure Sr solution in 0.3 M HNO$_3$ using a multi collector inductively coupled plasma mass spectrometer (MC-ICP-MS, Thermo Neptune) in medium mass resolution. The MC-ICP-MS was equipped with an APEX-Q (ESI) desolvater and a nebulizer with an uptake rate of 70 µl min$^{-1}$ and a nickel sampler cone. Radiogenic Sr isotope ratios were determined over one block of 20 cycles with an integration time of 16 seconds each. The sequence of a sample run was comprised of 10 to 12 blocks, where each block comprised a blank, four samples, and five SRM 987 which were not processed through chemistry. Blank correction of samples and reference material during the sequence was less than 0.4% of the sample signal. The intensities of the ion beams on the masses $^{82}$Kr (L4), $^{83}$Kr (L3), $^{84}$Sr (L2), $^{85}$Rb (L1), $^{86}$Sr (central Cup), $^{87}$Sr (H1) and $^{88}$Sr (H2) were monitored using Faraday collectors equipped with 10$^{11}$ Ω and one 10$^{12}$ Ω (connected to L4 cup) resistors. Isobaric interference on the masses 84, 86, and 87 were corrected for with the Kr and Rb isotope ratios measured prior to the sequence run. To correct for any natural and instrumental isotope fractionation, the measured $^{87}$Sr/$^{86}$Sr ratio was normalized to a $^{88}$Sr/$^{86}$Sr ratio of 8.375209 (Nier, 1938's value) by using an exponential law. Finally, the $^{87}$Sr/$^{86}$Sr ratios were corrected for a session offset that account for the differences between the certified and measured $^{87}$Sr/$^{86}$Sr ratio of the SRM 987 reference material, which in any case where smaller than ± 0.00006 (2SD).

**4.3 Parameterizing geogenic and biogenic element fluxes in a terrestrial ecosystem**

The parameterization of the "geogenic nutrient pathway" and the "organic nutrient cycle" (Fig. 2) to characterize element fluxes into, within, and from the Critical Zone and its ecosystem components is thoroughly described in Uhlig and von Blanckenburg (2019). Here, we only briefly summarize the metrics, which are shown in Table 2. Calculations and parameters used for these metrics are presented in Appendix A, including the propagation of uncertainties. A statistical analysis (i.e. ANOVA, Pearson correlation coefficients) of the weathering parameters is presented in Appendix B.

**4.4 Data reporting**

The original data and that pertaining to the companion paper (Oeser and von Blanckenburg, 2020a) can be found in a separate open access data publication (Oeser and von Blanckenburg, 2020b). These tables are referred to "Data Tables S1 to S5" from here on.

**5 Results**

We structure the presentation of our results in the following sequence: (1) the element fluxes of the geogenic nutrient pathway; (2) the availability of elements in regolith to plants; and (3) the plant chemical composition along with the element fluxes that couple the geogenic nutrient pathway to the organic nutrient cycle. The fluxes are presented as study-site averages, with the full dataset available in an associated open access data publication (Oeser and von Blanckenburg, 2020b).

We focus the detailed presentation of these results on P and K, the two most important rock-derived mineral nutrients to plants. Further data is provided for Al, Ca, Fe, Mg, Mn, Na, Si, and Sr. All metrics are defined in Table 2.

**5.1 Element fluxes contributing to the geogenic nutrient pathway**

**5.1.1 Degree of weathering and elemental gains and losses**

The chemical depletion fraction (CDF; Table 2, Eq. 5 and Appendix A) and elemental mass transfer coefficient ($\tau$; Table 2, Eq. 6 and Appendix A) disclose the total and the element-specific loss, respectively, of soluble elements relative to bedrock. Thus, both metrics quantify the degree of weathering. The average CDF of the shallowest subsoil (combined analysis of north- and south-facing

profiles) in Pan de Azúcar, Santa Gracia, La Campana, and Nahuelbuta amounts to 0.03, 0.54, 0.50, and 0.25, respectively (Fig. 3; Data Table S2). At all four sites, the elemental losses (Fig. A1; Data Table S2) can be attributed to a "kinetically limited weathering regime" (Brantley and Lebedeva, 2011). This means that the erosion rate is at a sufficient level to continuously replenish the weatherable primary minerals that transit vertically through the weathering profile.

Systematic differences in chemical depletion (i.e. CDF and $\tau$) are not discernible between north- and south-facing slopes. Anomalously high Zr concentrations throughout the entire north-facing profile at La Campana cause one exception to this rule. Moreover, we found that neither CDF nor $\tau^X$ differ significantly between Santa Gracia, La Campana, and Nahuelbuta, despite both increasing precipitation and increasing biomass growth.

A comprehensive presentation of these data can be found as Appendix in Fig. A1 and in the supplementary Data Table S2.

**5.1.2 Elemental chemical weathering fluxes**

The soil weathering rate W quantifies the bulk weathering flux from rock and regolith. This flux is lowest in Pan de Azúcar (0 – 0.9 t km$^{-2}$ yr$^{-1}$) and highest in La Campana (53.7 – 69.2 t km$^{-2}$ yr$^{-1}$). In Santa Gracia

(7.2 – 11.9 t km$^{-2}$ yr$^{-1}$) and Nahuelbuta (3.5 – 7.5 t km-2 yr-1, Table 1; Oeser et al., 2018; Schaller et al., 2018b), these fluxes are at a similarly intermediate level.

$W^X_{regolith}$ (Table 2, Eq. 3 and Appendix A) quantifies element-specific release fluxes from rock and regolith by weathering. It thus assesses the maximum possible weathering supply of nutrients to plants by the "geogenic pathway", as some of this flux is potentially lost into groundwater before being

accessible to roots. The weathering-release fluxes for phosphorus ($W_{regolith}^P$) amount to $1.3 \pm 0.4$, $12 \pm 3$,

$19 \pm 6$, and $11 \pm 4$ mg m$^{-2}$ yr$^{-1}$ and of potassium ($W_{regolith}^K$) to $30 \pm 30$, $80 \pm 50$, $840 \pm 220$, and $100 \pm 120$

mg m$^{-2}$ yr$^{-1}$ (Fig. 4, Table 3) in Pan de Azúcar, Santa Gracia, La Campana, and Nahuelbuta, respectively.

Similar trends are seen for Al, Na, and Si along with Fe and Sr. The rates of supply of P, K, and the

aforementioned elements are thus similar at both Santa Gracia and Nahuelbuta despite the differences in

MAP, NPP, and vegetation cover. $W_{regolith}^{Ca}$ and $W_{regolith}^{Mg}$ deviate from this general pattern: the highest Ca

and Mg weathering-release fluxes occur in Santa Gracia followed by La Campana, Nahuelbuta, and Pan

de Azúcar. These elevated fluxes in Santa Gracia are attributed to the initial bedrock mineralogy, with

their high Ca and Mg concentration (Data Table S1).

**5.2 Availability of mineral nutrients to plants**

The maximum amount of mineral nutrients present can be assessed by determining their inventory in bulk

regolith ($I_{bulk}^X$; Table 2, Eq. 8 and Appendix A). For most elements $I_{bulk}^X$ is by far greatest in Santa Gracia

(apart from K and Si; Table A1). $I_{bulk}^X$ at the other three study sites are at similar levels. Element

concentrations in the bio-available fraction are orders of magnitude lower than in the bulk regolith

(Fig. A2 & A3, Data Table S3). Bio-available P in saprolite ($I_{bio-av, sap}^P$) is virtually absent in Pan de

Azúcar and amounts to 21, 39, and 23 g m$^{-2}$ in Santa Gracia, La Campana, and Nahuelbuta, respectively

(Table A1). $I_{bio-av, sap}^K$ equals 253 in the northernmost, and 23, 70, and 19 g m$^{-2}$ at the sites progressively

southwards. The inventory of the remaining mineral nutrients in saprolite generally decreases from north

to south. Accordingly, the total inventory (i.e. the sum of all determined inventories) is highest in Pan de

Azúcar (5100 g m$^{-2}$), intermediate in Santa Gracia (2100 g m$^{-2}$) and La Campana (1600 g m$^{-2}$), and lowest

in Nahuelbuta (140 g m$^{-2}$; Table A1). Note that $I_{bio-av, sap}^X$ was calculated over the uppermost 1 m of

saprolite, whereas in fact the zone of mineral nutrient extraction might extend deeper (Uhlig et al., 2020).

Bio-availability in soil features a similar trend. The total inventory is highest in Pan de Azúcar (2100 g m$^{-2}$), on par in Santa Gracia (960 g m$^{-2}$) and La Campana (1000 g m$^{-2}$), and despite featuring the thickest

soils, lowest in Nahuelbuta (200 g m$^{-2}$). P deviates from this general trend: $I_{bio-av, soil}^P$ amounts to 3.3 g m$^{-2}$

in Pan de Azúcar, 22 g m$^{-2}$ in Santa Gracia, 28 g m$^{-2}$ in La Campana, and 31 g m$^{-2}$ and Nahuelbuta

(Table A1). $I_{bio-av, soil}^K$ behaves differently, and amounts to 53, 38, 90, and 38 g m$^{-2}$ in Pan de Azúcar, Santa Gracia, La Campana, and Nahuelbuta, respectively. Thus, K is almost equally available to plants in all four study sites.

**5.3 Plant element composition and nutrient-uptake fluxes**

Average elemental concentrations in bulk plants generally decrease from Pan de Azúcar towards Nahuelbuta. For example, the Al and Na concentrations in the plants of Pan de Azúcar reach 2700 and 34600 µg g$^{-1}$, respectively, compared with minima of 70 and 80 µg g$^{-1}$ in Nahuelbuta. However, element specific deviations from this pattern exist (Table 4). The most prominent exceptions are those of P and K. Average P concentration increases from 290 µg g$^{-1}$ in Pan de Azúcar to 1400 µg g$^{-1}$ in Nahuelbuta.

The average K concentration amounts to 6900, 6400, 12000, and 5400 µg g$^{-1}$ along the north-south gradient. Thus, in Pan de Azúcar, Santa Gracia, and Nahuelbuta, average K concentrations are in a similar range, whereas in La Campana, K concentration in plants is almost 2x higher than in the other sites (Table 4). In Pan de Azúcar and Santa Gracia some elemental concentrations in plants are exceptionally high. This elevated mineral-nutrient storage is typical for plants growing in infertile habitats (Chapin III

et al., 2011). Accumulation of such an internal nutrient pool allows for plant growth when conditions improve, e.g. during rare rain events (e.g. Chapin III, 1980; Chapin III et al., 2011; Vitousek et al., 1998). For example, high amounts of Al and Na are incorporated into plants tissues, though they may hinder plant growth at high concentrations (e.g. Delhaize and Ryan, 1995; Kronzucker and Britto, 2011). However, Al-toxicity is prevented in these plants by accumulation of correspondingly high amounts of

Si that compensates the effects of Al (Liang et al. (2007). The exceptional high Na concentration in *N. mollis* in Pan de Azúcar is typical of the metabolism of *N. mollis* which is known to be covered with salt glands on their leaves, aiding to retrieve water by directly condensing moisture from unsaturated air (Rundel et al., 1980; Mooney et al., 1980).

The nutrient-uptake fluxes ($U_{total}^X$; Table 2, Eq. 4 and Appendix A) of P and K increase from north to

south, such that $U_{total}^P$ amounts to $5 \pm 2$, $70 \pm 20$, $170 \pm 90$, and $350 \pm 100$ mg m$^{-2}$ yr$^{-1}$ and $U_{total}^K$ to $110 \pm 40$, $500 \pm 200$, $2000 \pm 1000$, and $1400 \pm 400$ mg m$^{-2}$ yr$^{-1}$ in Pan de Azúcar, Santa Gracia, La Campana, and Nahuelbuta, respectively (Table 3). $U_{total}^X$ of the elements Ca, K, Mg, Mn, P, and Sr exceed

$W^X_{regolith}$ up to several times. $U^X_{total}$ and $W^X_{regolith}$ are similar for Mg, Mn, and Sr in La Campana (Fig. 4; Table 3). $U^X_{total}$ of the remaining elements are, with the exception of Fe and Na in Pan de Azúcar, always

lower than their release by weathering.

### 5.4 $^{87}$Sr/$^{86}$Sr isotope ratios

Radiogenic Sr isotope ratios on bulk bedrock and regolith samples disclose mineral-weathering reactions and the incorporation of external sources into the regolith profiles. Moreover, $^{87}$Sr/$^{86}$Sr in the bio-available fraction and plants reveal the plants' mineral nutrient sources.

In Pan de Azúcar, the $^{87}$Sr/$^{86}$Sr ratio of average bedrock is $0.726 \pm 0.002$ (Fig. 5, Table A2). In regolith, $^{87}$Sr/$^{86}$Sr differs significantly between the two profiles ($0.728 \pm 0.003$ and $0.733 \pm 0.003$ on the north- and south-facing regolith profile, respectively) which can be attributed to varying degrees of atmospheric deposition ($^{87}$Sr/$^{86}$Sr$_{seaspray} = 0.7092$; Pearce et al., 2015). The $^{87}$Sr/$^{86}$Sr ratios in the bio-available fraction of saprolite and soil deviate by 0.02 from those of bulk bedrock and regolith but do not vary considerably

between saprolite and soil, or between the north- and south-facing slopes. Bulk plant samples yield $^{87}$Sr/$^{86}$Sr ratios of 0.710 and are thus indistinguishable from the $^{87}$Sr/$^{86}$Sr ratio in the bio-available fraction ($0.710 \pm 0.001$; Fig. 5, Table A2).

In Santa Gracia, the $^{87}$Sr/$^{86}$Sr ratios in both bedrock and the regolith profiles do not differ significantly ($^{87}$Sr/$^{86}$Sr$_{rock} = 0.7039 \pm 0.0004$, $^{87}$Sr/$^{86}$Sr$_{regolith} = 0.7043 \pm 0.0003$; Fig. 5, Table A2). The radiogenic Sr

composition of the bio-available fractions in saprolite and soil are identical within uncertainty, and no differences in $^{87}$Sr/$^{86}$Sr between the north- and south-facing regolith profile are apparent. Plants yield an average $^{87}$Sr/$^{86}$Sr ratio of $0.7062 \pm 0.0001$ and are thus indistinguishable from the bio-available fractions in saprolite and soil (Fig. 5, Table A2).

The bulk regolith $^{87}$Sr/$^{86}$Sr ratio in La Campana ranges from 0.7051 in the north-facing to 0.7055 in the

south-facing regolith profile. These ratios are lower than bedrock ($0.7063 \pm 0.0003$; Fig. 5, Table A2) which can be attributed to the loss of a mineral with a high $^{87}$Sr/$^{86}$Sr isotope ratio (e.g. biotite) beneath the sampled regolith profiles. The radiogenic Sr composition of the bio-available fraction in saprolite and soil amounts to 0.7051 and 0.7053, in the north- and south-facing slopes, respectively, and is within the range of bulk regolith. The average $^{87}$Sr/$^{86}$Sr ratio in plants is 0.7059 and can be as high as 0.7063 in

*Cryptocaria alba* (Data Table S7) and is thus higher than the soil and saprolite bio-available fractions. All these ratios are lower than in bulk bedrock.

In Nahuelbuta the radiogenic Sr isotope ratio in bedrock (0.716 ± 0.007) is in good agreement to those reported by Hervé et al. (1976) for the granitoid basement (0.717). However, the large spread among the bedrock samples implies petrological and geochemical heterogeneity of the Nahuelbuta mountain range (e.g. Hervé, 1977). Thus, $^{87}Sr/^{86}Sr$ in regolith is also variable (Fig. 5, Table A2 & S2). The $^{87}Sr/^{86}Sr$ ratios in both bio-available fractions in Nahuelbuta are restricted to a relatively narrow range in both regolith profiles, equal to 0.711 ± 0.002 and are indistinguishable from the mean ratio in plants (Fig. 5, Table A2). Individual plants' radiogenic Sr signature are distinct from each other and reflect the slope's bio-available fraction they grow on.

## 6 Discussion

### 6.1 The source of mineral nutrients

Comparing the radiogenic Sr composition of the bio-available fractions in saprolite and soil with that of bulk plant serves as a proxy for the nutrient sources of plants. At all four sites, the $^{87}Sr/^{86}Sr$ ratio in plants is largely indistinguishable within uncertainty to the bio-available fraction they grow on (Table A2), and no differences in $^{87}Sr/^{86}Sr$ between leaves, to twig, or stem are apparent (Data Table S5). Neither the plant $^{87}Sr/^{86}Sr$ ratio nor the $^{87}Sr/^{86}Sr$ ratio of the bio-available fraction is identical to that of bedrock or of bulk regolith. We conclude that plants obtain their Sr from the bio-available fraction rather than directly from primary minerals or from the atmosphere through leaves. Only La Campana showed evidence for a deep nutrient source (i.e. somewhere between the bottom of the regolith profile and unweathered rock) in the elemental-depletion pattern (Fig. A1). Here, deep-rooting plants (e.g. *Lithraea caustica*; Canadell et al., 1996) bypass the bio-available fraction of saprolite and soil and take up Sr with a higher proportion of radiogenic $^{87}Sr$ which has been released through biotite weathering beneath the regolith profiles. We can also use the $^{87}Sr/^{86}Sr$ ratio to identify the ultimate source of bio-available Sr. In the southernmost mediterranean and humid-temperate sites of La Campana and Nahuelbuta, the bio-available Sr is supplied by release from rock and regolith through weathering. In arid Pan de Azúcar the Sr pool in the bio-

available fraction is formed by deposition from atmospheric sources (up to 93% seaspray contribution; Table A2). In semi-arid Santa Gracia, we found a possible combination of both sources (up to 43% seaspray contribution; Table A2).

Expanding our analysis of the source of mineral nutrients, we normalized both the mineral nutrient concentrations in plants (Table 4) and those in the bio-available fraction in saprolite and soil (Data Table S3) by the most-demanded rock-derived mineral nutrient P (Fig. 6). This normalization removes differences in concentrations induced by the very different matrices of regolith and plant. In this analysis, an element X that plots on the 1:1 line would have the same X:P ratio in plants and in the bio-available fraction. In turn, any deviation from that line would indicate positive or negative discrimination of an element contained in the regolith bio-available fraction by plants relative to P. We find a good correlation in the X:P ratios for all elements, and the ratios found in plants reflect those in the bio-available regolith fraction to within one order of magnitude. We interpret this correlation to confirm nutrient uptake mainly from the bio-available fraction. We also note that the X:P ratios increasingly approach the 1:1 line with increasing NPP from Pan de Azúcar to Nahuelbuta and the agreement is more pronounced in soil than in saprolite. We interpret these shifts to denote the increasing significance of recycling, a topic we return to in the next section.

## 6.2 An increase in nutrient recycling with NPP

In Section 5.1.2 we established that neither total weathering rate W, nor elemental weathering rates $W^X_{regolith}$, correlate with NPP. Only at La Campana weathering rates are elevated, as expected from the higher denudation rate. Santa Gracia and Nahuelbuta, have similar denudation rates and element release rates by weathering $W^X_{regolith}$, yet elemental uptake rates $U^X_{total}$ of P, K, and Ca increase between a factor of two and five (Figure 4). We examine these correlations in more detail in Section 6.5. Here we first focus on the question: How is mineral nutrient demand satisfied at the more vegetated sites?

Recycling of mineral nutrients from organic material is the key mechanism enabling differences in NPP. We quantified recycling by the nutrient recycling factor $Rec^X$ (Table 2, Eq. 7 and Appendix A; Table 5; note that in this discussion we use the $Rec^X$ calculated for $W^X_{regolith}$ from rock weathering, whereas in Table 5 and Fig. A4 we also show $Rec^X$ including atmospheric inputs in Pan de Azúcar). The amplitude

of recycling varies from nutrient to nutrient and site to site. In the arid Pan de Azúcar, nutrients are primarily recycled via photodegradation of shrubs (e.g. Gallo et al., 2006; Day et al., 2015). In the remaining sites $Rec^X$ occurs through all organic-bearing soil horizons and increases from Santa Gracia to Nahuelbuta and is highest for Ca (increasing from 1 to 6), K (increasing from 6 to 15), and P (increasing from 5 to 30; Table 5). Thus, despite having the smallest nutrient inventory of bio-available nutrients (Table A1) but the highest NPP, Nahuelbuta can at least partially satisfy its nutrient requirements through efficient nutrient recycling. In the mediterranean site La Campana, nutrient requirements are satisfied through recycling and uptake from depth – a mechanism which has just recently been shown to balance losses by erosion and contributing to ecosystem nutrition (Uhlig et al., 2020). In contrast, in the (semi-) arid sites, where the bio-available pool is larger, plants forage nutrients and water by deep rooting from depth (McCulley et al., 2004).

The $Rec^X$ metric reflects a mass balance between the total weathering zone and the total vegetation cover but does not yield insight to the mechanisms of recycling. The elemental stochiometric considerations presented above show that recycling is indeed fed from plant material accumulated in soil (Lang et al., 2017). With increasing recycling the nutrient pools in the soil bio-available fraction are increasingly dominated by the pool of recycled nutrients, thus shifting the X:P ratio in the bio-available fraction successively towards the X:P ratio in vegetation (Fig. 6). In other words, over the course of several recycling loops, the chemical composition of the bio-available fraction and biota eventually approaches a ratio close to the relative requirement of the ecosystem for the different nutrients (Vitousek et al., 1998).

### 6.3 Processes that set the size of the bio-available pool

In none of our sites is the bio-available nutrient pool entirely depleted (Table 3), but its elemental concentrations strongly shift along the gradient. The concentrations of the mineral nutrients K, Ca, and Mg in saprolite are highest in the arid site, lower in the semi-arid and mediterranean site, and lowest in the humid-temperate site. The element concentration in the bio-available fraction translates into the size of the inventory, quantifying the pool size. Note, however, that the true inventory can in fact be larger than the 1 m inventory that we have used for its calculation. This is suggested by the elevated $^{87}Sr/^{86}Sr$ ratios in plants at La Campana suggesting extraction of a pool beneath the bio-available upper saprolite.

Nutrient uptake from greater depths have recently been demonstrated by Uhlig et al. (2020). The bio-available pool represents the link between the organic and the geogenic pathway. That is because weathering in the geogenic pathway supplies elements that plants take up and recycle in the organic pathway (Uhlig and von Blanckenburg, 2019). We thus briefly review the potential processes that may set the pool size.

If a bio-available pool is in conceptual steady state, input fluxes and loss fluxes balance. Over millennial time scales or longer, we consider that such a balance must exist, as otherwise a pool might become depleted. In this case the inventory of the pool is set by the input fluxes of an element and a first-order rate constant that describes the relationship between the loss flux as a function of element inventory and thus the retention capacity. Essentially it is the inverse of the turnover time of an element. Biotic processes likely contribute towards setting this retention capacity directly or indirectly, a topic we return to below. Given that elemental weathering fluxes $W^X_{regolith}$ do not correlate with pool size we assume that retention capacity sets the pool size.

A first potential control over element retention capacity are pedogenic properties. The decrease of soil pH from 8 at the arid site to 4 at humid-temperate site (Bernhard et al., 2018) might cause the decrease in the bio-available divalent base cations Mg, Ca, and Sr. Conversely, the decrease in pH could be the result of the loss of these elements and thus their pH buffering capacity. Another possibility is the degree of complexing of elements to organic molecules. Such complexing might lead to either higher retention, or higher loss, depending on the element. Organic complexing is likely more pronounced in the mediterranean and humid-temperate sites where soil organic carbon concentrations are higher compared to the (semi-) arid sites (Bernhard et al., 2018). However, elements like Al and P, which are readily complexed, are abundant in higher concentration in the humid-temperate and mediterranean sites than in the other two sites. Differences in water flow is the third cause we discuss. Where fluid residence times are long, concentrations of solutes are more likely to be at equilibrium with secondary minerals (Maher and Chamberlain, 2014) and the bio-available fraction, formed by precipitation and sorption from pore fluids, can build up. We consider this to be the case in the low-precipitation sites. At sites with high MAP regolith fluids may be diluted, and thus desorb elements from the bio-available pool (i.e. leaching). Such dilution effect might be in effect at Nahuelbuta for elements like Mg and Ca. At Nahuelbuta these are

also the elements with the lowest bio-available inventory. We consider the pH and water flow to be the main factor governing the size of the bio-available pool.

### 6.4 Concepts for biotas role in setting fluxes in the geogenic and the organic nutrient cycle

Even if negligible on ecological timescales, ecosystems experience losses of nutrients through erosion (e.g. Heartsill Scalley et al., 2012) and as solutes (e.g. Chaudhuri et al., 2007). To prevent bio-available nutrients becoming depleted over longer timescales, the pool must be replenished (Uhlig and von Blanckenburg, 2019). Biological mechanisms comprise two means to regulate this delicate balance between nutrient replenishment by weathering and plant uptake. The first is by adjusting the recycling of nutrients, as shown in Section 6.2. At Nahuelbuta, where the bio-available pool is smallest, nutrient recycling rates are the highest. If the bio-available pool is small, plants may invest energy into re-using P and other elements from leaf litter, rather than foraging P at depth which is associated with higher energy expenditure (Andrino et al., 2019). This is a component of the organic nutrient cycle. The biochemical mechanisms of nutrient-recycling are beyond the scope of this paper, but are thought to be related to leaf litter quality (Hattenschwiler et al., 2011), soil fungal and microbial communities (Fabian et al., 2017; Lambers et al., 2008), and plant diversity (Lambers et al., 2011; Oelmann et al., 2011; van der Heijden et al., 1998).

The second means for biota to influence the bio-available pool is via the geogenic pathway. Nutrient replenishment may take place either by exogenous inputs (e.g. Boy and Wilcke, 2008; Porder et al., 2007; Vitousek, 2004; Vitousek et al., 2010), or by weathering of primary minerals (Uhlig et al., 2017; Uhlig and von Blanckenburg, 2019). In arid Pan de Azúcar, where weathering-release fluxes are low, these pools are being replenished by the deposition of atmospheric sources (up to 93%; Table A2). In the other study sites the bio-available pools are replenished by weathering of rock and regolith. The timescales $T^X_{bio-av,W}$ of replenishment from weathering are long, and typically orders of magnitude longer than their turnover times with respect to plant uptake $T^X_{bio-av,U}$. For example, the inventory of K in the bio-available soil pool at Nahuelbuta is turned over every 30 years between soil and plants, but it takes 400 years to be replenished in its entirety by weathering (Table A3). Previous models in ecosystem science (e.g. Bormann et al., 1969; Vitousek and Reiners, 1975; Vitousek et al., 1998) suggest that increasing mineral-nutrient

610 demand will eventually lead to tightly coupled recycling loops such that nutrient losses will be minimized, and plant nutrition is sustained. Our data is also consistent with a relationship between demand (i.e. NPP) and recycling efficiency.

If recycling indeed exerts the dominant role in the supply of mineral nutrients, then we need to revisit the significance of biogenic weathering towards the nutrition of plants. The direct and indirect impacts of plants and their associated microbiota on weathering is well-documented and can be categorized into four suites of processes: (*A) Direct primary mineral dissolution by ectomycorrhizal fungi.* Ectomycorrhizal fungi can directly extract nutrients such as P, K, Ca, Mg, and Fe from minerals distant from the root, even under dry conditions, and thereby actively increase mineral dissolution kinetics. Laboratory dissolution experiments (Balogh-Brunstad et al., 2008b; Gerrits et al., 2020; Kalinowski et al., 2000), plant growth mesocosms (Bonneville et al., 2011; Smits et al., 2012), and deployment of minerals within the soil of natural ecosystems (Balogh-Brunstad et al., 2008a) all show either evidence for mineral dissolution by mycorrhiza, or quantify an increase in mineral dissolution over abiotic controls. Whether these short-term experiments can be extrapolated to the millennial time scales of the geogenic nutrient pathway is not obvious (review by Finlay et al., 2020). Over these time scales, mineral dissolution is often slowed by the development of nanoscale layers at the interface (Gerrits et al., 2020) or coatings by secondary precipitates (Oelkers et al., 2015). Slowing of mineral dissolution with time, known from weathering zone studies, has also been attributed to coating by secondary precipitates (White and Brantley, 2003), or to chemical saturation of pore fluids (Maher, 2010). *(B) Roots deepening regolith thickness*. Tree roots can physically penetrate and biogeochemically alter the immobile regolith underlying mobile soil (Brantley et al., 2017). They can take water up from depth, recycle water to depth for storage, or provide pathways in which water bypasses rather than infiltrating the shallow regolith (Fan et al., 2017). Deep roots aid nutrient transfer from the subsoil to shallow levels (Jobbágy and Jackson, 2004). *(C) Canopy and roots converting precipitation into evapotranspiration* (Drever and Zobrist, 1992). In sites with higher vegetation cover, water vapor is recycled and does not immediately enter runoff. By providing canopy, trees both modulate infiltration while turning water back into transpiration (Ibarra et al., 2019). For example Ibarra et al. (2019) have shown that total runoff can decrease by up to 23% as vegetation cover increases from barely vegetated to highly vegetated sites. Water recycling hence decreases total runoff

and potentially reduces weathering-release fluxes in the highly vegetated sites. (D) *Increasing mineral solubility by release of soil CO$_2$ and organic complexing agents.* Through the respiratory release of soil CO$_2$ and excretion of organic complexing agents, plants, hyphae, and their associated microbiota can increase the solubility limits of primary and secondary minerals by a factor of up to 10 (Perez-Fodich and Derry, 2019; Winnick and Maher, 2018). If dissolution is not kinetically limited, we would indeed expect higher solute concentrations with higher soil CO$_2$, and hence higher dissolution rates of primary minerals (Winnick and Maher, 2018).

Studies of biogenic weathering in natural Critical Zone systems struggle to disentangle expressions of these biogenic drivers of weathering rates from various competing drivers of weathering. Although the sites were selected to minimize potential confounding effects, this study also faces this challenge. We turn to a statistical approach in isolating any potential biogenic weathering signal.

**6.5 Is weathering modulated by biota? A statistical analysis**

To single out the possible biogenic weathering driver from the confounding factors at the EarthShape sites we used correlational statistics between indicators of weathering and metrics for its potential drivers along the EarthShape gradient. We determined Pearson correlation coefficients to determine how the degree of weathering (CDF, $\tau^X$) and the flux of weathering (W, $W_{regolith}^X$) depend on denudation rate D, water availability (approximated by mean annual precipitation, MAP), and biomass growth as quantified by net primary productivity (NPP). See Appendix B for a detailed description on statistical analysis and Table A4, A5, and A6 for the results. We used these statistics to evaluate three starting hypotheses that reflect the basic confounding factors: (1) Where denudation rate D is high bulk weathering fluxes are high, since minerals with fast dissolution kinetics, such as plagioclase and P-bearing apatite, are supply-limited (Dixon et al., 2012; Porder et al., 2007). Where D is high, regolith residence times are low such that $\tau^X$ for elements not mostly contained in rapidly dissolving minerals are not depleted. (2) At sites at which MAP and hence runoff is high, weathering fluxes are high. This is because weathering rate is proportional to runoff for the chemostatic elements that comprise the bulk of the weathering flux, amongst them Si that contributes roughly half of the flux (e.g. Godsey et al., 2019; Maher and Chamberlain, 2014). As a result, CDF and $\tau^X$ will also be high. $\tau^X$ of soluble elements (e.g. Na) will be higher at higher runoff

than $\tau^X$ of elements that strongly partition into secondary phases. (3) If NPP is high the degree (CDF, $\tau^X$) and rate of weathering (W, $W_{regolith}^X$) will be high (e.g. Berner et al., 2003; Brantley et al., 2011; Buss et al., 2005; Kelly et al., 1998; Porder, 2019; Schwartzmann, 2015), for the reasons predicted in Section 6.4. In support of hypothesis (1) we find that total and elemental weathering rates correlate well with D (Table A4) and only a weak correlation relates denudation rate with the degree of weathering and elemental depletion. Thus, denudation rate is the predominant driver of weathering rate. However, D itself is also correlated with MAP and NPP. To evaluate whether D is nevertheless the main driver we exclude the La Campana site of unusually high D. The correlations between W, $W_{regolith}^X$, and D are still significant (Table A5) confirming that D is the main driver of weathering rate. Concerning hypothesis (2), neither the degree nor rates of weathering correlate with MAP. Only the soluble element Na becomes more depleted (Table A5) at higher MAP. Thus, a competing effect seem to counteract the expected increase in weathering rate with precipitation. As NPP is an output of the LPJ-GUESS model for which MAP is the basis, it is no surprise that both parameters are strongly correlated (Table A4 & A5). We would thus expect the same strong relationship between the degree and rates of weathering and NPP as with MAP. This is indeed the case. However, weathering release rates $W_{regolith}^X$ for elements like Na, P, and Si correlate slightly more strongly with NPP than with MAP (Table A5). This is the only indication that biomass growth exerts any control over weathering at all. In summary, neither MAP nor NPP seem to have a major impact on the degree and rates of weathering, and D is the main driver of total and elemental weathering rate at the EarthShape sites.

In this analysis we have not evaluated the potential confounding effects of differences in bedrock mineral composition. Because of the lack of an unequivocal metric allowing a statistical evaluation of the resulting differences in rock weatherability we focus on a comparison between the two study sites in semi-arid (Santa Gracia) and humid-temperate climate (Nahuelbuta). At these two sites, denudation rates $(15 - 48$ t $km^{-2}\,yr^{-1})$ and soil residence times $(22 - 28$ kyr; Schaller et al., 2018b) are similar. Although both granitoid, bedrock between the two sites differs. Santa Gracia is underlain by diorite, a mafic rock, while Nahuelbuta is underlain by granodiorite (Oeser et al., 2018). Thus, the suite of primary minerals in Santa Gracia is more prone to weathering than in Nahuelbuta. Specifically, this means a higher amount of plagioclase and amphibole, and less unreactive quartz, at Santa Gracia. These differences in

predominantly Ca- and Mg-bearing minerals are reflected in higher Ca and Mg inventories in bulk regolith in Santa Gracia (Table A1) , that also translate into higher Ca and Mg weathering fluxes (Table 3). Total soil weathering rates ($5 - 10$ t km$^{-2}$ yr$^{-1}$; Table 1), and differences in weathering properties are not statistically significant (Table A6). The weathering-release fluxes (Fig. 4, Table 3) for K, Na, P, and Si are similar despite massive differences in vegetation cover, NPP, and even MAP (Table 1 & A6). These similarities, and the higher weathering fluxes of Ca and Mg at Santa Gracia can be explained with the confounding effects of higher rock weatherability at Santa Gracia and the higher precipitation at Nahuelbuta. A comparison of concentration-discharge relationships between catchments underlain by mafic (basaltic) and granitoid rock (Ibarra et al., 2016) shows higher solute concentrations for all major elements in the basaltic catchments at a given runoff, and the preservation of chemostatic solute concentrations to higher runoff than in granitoid catchments. As a result, weathering fluxes in mafic catchments at low runoff are similar to fluxes from granitoid rock subjected to high runoff, as we observe at Santa Gracia and Nahuelbuta. Regardless, an increase in either weathering rate or degree of weathering at Nahuelbuta resulting from the 3.5 times higher NPP at Nahuelbuta is not discernible.

### 6.6 Do negative feedbacks decouple biomass growth from weathering rate and degree?

Why do neither the degree nor the rate of weathering increase with NPP or MAP, nor does higher biomass growth overwhelm differences in rock mineralogy? Nutrient recycling may be the mechanism that decouples weathering from NPP, as shown in Section 6.2. Even so, the higher runoff results in a greater loss of nutrients from the bio-available pool and thus requires higher weathering rate to balance the loss. We thus speculate that the increased vegetation cover might even counteract a potential increase in weathering that would be caused by the increase in MAP, essentially damping the geogenic pathway. We return to the four suites of processes as outlined in Section 6.4 on the direct and indirect impacts of plants and their associated microbiota on weathering and discuss their potential operation at the EarthShape sites.

*(A) Direct primary mineral dissolution by ectomycorrhizal fungi.* As yet we have no direct observations on nutrient foraging by fungi and other microbes in regolith from the EarthShape sites as obtained on other mountain sites in Chile (Godoy and Mayr, 1989). Proxies for total microbial biomass in saprolite

do not increase along the gradient: total gene copy numbers have similar ranges from Santa Gracia to Nahuelbuta, and DNA amounts even decrease slightly (Oeser et al., 2018). Common strategies of microbial symbionts with tree roots suggest that energy investment into nutrient recycling from leaf litter is more advantageous than dissolving primary minerals (Andrino et al., 2019). Thus, we would expect that mycorrhiza predominantly aid recycling in Nahuelbuta. In Santa Gracia, however, the absence of a

litter layer may prompt the subsurface fungal network to invest in primary mineral dissolution, adding microbial weathering to total weathering at that that site.

*(B) Roots deepening regolith thickness.* A detailed survey of rooting depth along the gradient has not been completed, but deep roots were not observed in Santa Gracia whereas in Nahuelbuta and La Campana, individual roots reach several meters into the saprolite. A and B horizons in Santa Gracia are shallow (20

730    – 40 cm), whereas they are deep in Nahuelbuta (80 – 100 cm; Bernhard et al., 2018; Oeser et al., 2018). We do not know the depth of the weathering front which appears to be at least a dozen of meters depth or more at both sites. Thus, deep rooting can benefit plant growth by increasing the size of the bio-available pool.

*(C) Canopy and roots converting precipitation into evapotranspiration.* Along the EarthShape transect

the potential 23% reduction in runoff predicted by Ibarra et al. (2019) is minor considering the 100-fold increase in precipitation over the entire gradient. A larger effect may occur if roots provide preferential flow paths such that infiltrating water bypasses the regolith matrix available for weathering (Brantley et al., 2017). However, given the deep weathering fronts - likely beneath rooting depth - we consider this effect to be minor, or even acting to increase deep weathering. Thus, we consider the hydrological impact

of plants on weathering to be minor along the gradient.

*(D) Increasing solubility by release of soil $CO_2$ and organic complexing agents.* Although with increasing NPP soil respiration of $CO_2$ should lead to increased primary mineral dissolution, plants potentially impose a negative feedback onto this dependence by influencing the silicon cycle. Because silicon is the most abundant element in felsic rock and regolith (besides oxygen), it exerts a major control on the total

weathering fluxes. The Si concentration in the bio-available pool is key in setting the saturation with respect to the various dissolving and precipitating minerals in regolith. Plants can impact this pool in both directions. Some plant species accumulate Si by active transporter-mediated uptake or through passive

uptake within the transpiration stream, whole others exclude Si and avoid accumulation (Ma and Yamaji, 2008; Schaller et al., 2018a). Enhanced Si uptake from soil solution by Si accumulating plants would result in Si undersaturation of solutions with respect to secondary minerals and would thus result in an increase in weathering rates. However, this increase may be damped. That is because these plants would also convert silicon into biosilica (e.g. phytoliths). If returned to soil in plant debris this biosilica becomes a key factor in the stability of secondary minerals (e.g. kaolinite; Lucas, 2001). However, neither factor seems to be the case: In the EarthShape sites, the average Si concentration in the above-ground living ecosystems ranges from 110 µg g$^{-1}$ in Nahuelbuta to 2500 µg g$^{-1}$ in Pan de Azúcar (Table 4). Thus, the Si weathering flux $W_{regolith}^{Si}$ exceeds the Si uptake flux $U_{total}^{Si}$ throughout (Table 3) and uptake from soil solution by plants equates to only 5%, 0.2%, and 2% of the Si release flux in Santa Gracia, La Campana, and Nahuelbuta, respectively. Only in Pan de Azúcar, relative uptake of Si is higher (25%).The ecosystems at our sites can thus be regarded to be below the threshold considered for Si accumulators (Schaller et al., 2018a). We can therefore exclude plant Si uptake and recycling of Si as a factor that increases weathering rates substantially. Rather, if plants in these ecosystems are discriminating against Si uptake whilst taking up water, the residual pore waters will get oversaturated with respect to secondary minerals. In this regard a key observation is provided by the analysis of pedogenic oxides (i.e. dithionite-extractable Al, Si extracted by oxalate, dithionite, and pyrophosphate; Oeser et al., 2018) and cation exchange capacity (Bernhard et al., 2018). These analyses suggest high amounts of amorphous precipitates and secondary minerals in the regolith of Nahuelbuta. We thus argue that Si is effectively captured in these barely soluble secondary minerals after initial dissolution from rock and regolith. In turn, $W_{regolith}^{Si}$ in Nahuelbuta is subdued despite elevated solubility of primary minerals due to increased $CO_2$ respiration by roots.

Ecosystems thus exert substantial control over weathering by both directly and indirectly modulating processes. These processes can either enhance or reduce weathering fluxes and result, in combination with effective recycling loops of plant-litter material, in well-balanced nutrient cycles. From our field data, we did not find evidence for coupling of silicate weathering fluxes with the mineral-nutrient demands of biota to an extent that exceeds other controlling factors of weathering. Our data suggests that the combination of recycling and negative feedbacks on weathering by secondary mineral formation

within the regolith decrease weathering rates in areas of high vegetation cover and net primary productivity from what they would be in the absence of high biomass density.

## 7 Conclusions

Even though the EarthShape study sites define a north-south gradient in precipitation and biomass production, no such gradient is apparent for weathering rates and weathering intensity between the study sites situated in semi-arid, mediterranean, and humid-temperate climate.

At all four sites we locate the primary mineral nutrient source to plants in the bio-available fraction. This pool of mineral nutrients is initially fed by geogenic sources, which comprise the weathering of primary minerals. It is further fed from organic sources, which involves recycling of nutrients from leaf litter. The size of the bio-available nutrient pool decreases from north to south and while pedogenic properties (e.g. pH) likely contribute to set its size, we attribute its decrease mainly to an increase in the below-ground water flow. To fulfill their mineral-nutrient demand at increasing NPP but decreasing pool size, ecosystems increase nutrient recycling rather than enhancing biogenic weathering. We consequently find that the organic nutrient cycle intensifies, whereas the geogenic nutrient pathway is steady despite increasing MAP and NPP.

In fact, the presence of plants might even counteract a potential weathering increase along the gradient by inducing secondary mineral formation rather than nutrient-acquisition through weathering. Due to nutrient buffering by recycling and a potential biological dampening of weathering, any additional contribution to weathering by NPP is unresolvable in our data and is certainly smaller than abiotic controls like denudation, rainfall, or bedrock mineralogy. The global silicate-weathering cycle may thus not be as sensitive to plant growth as commonly thought and cannot be simulated in a straightforward manner in weathering models. This non-linear behavior is of relevance for models of the global weathering and the linked carbon cycle, of which accelerated weathering by land plants since the Ordovician is a common component.

# 8 Appendices

## Appendix A: Calculation of fluxes and inventories in terrestrial ecosystems

### Weathering indices (CDF & $\tau$)

Zr, Ti, and Nb are commonly used to estimate mass losses to the dissolved form during weathering (Eqs. 5 & 6) as they are presumed to be the least mobile elements during weathering (Chadwick et al., 1990; White et al., 1998). The suitability of these elements for the EarthShape study sites has been evaluated and thoroughly discussed on a site to site basis in Oeser et al. (2018). Based on possible Ti-mobility in some samples and the fact that Zr is used as a reference element in the majority of weathering and soil production studies worldwide (e.g. Fisher et al., 2017; Green et al., 2006; Hewawasam et al., 2013; Riebe and Granger, 2013; Riebe et al., 2001; Schuessler et al., 2018; Uhlig et al., 2017), Zr was taken as immobile reference element in this study.

The calculations of these weathering indices rely on a good approximation of the chemical composition of the initial bedrock from which regolith formed. To this end, any regolith sample with a Zr concentration that was lower than the mean of unweathered bedrock by more than one standard deviation (1SD) was excluded from further consideration. Because a lower Zr concentration cannot be due to weathering, such regolith samples likely originate from chemically distinct bedrock or small-scale bedrock heterogeneities (e.g. a pegmatitic vein). Saprolite samples were also excluded from our data set if Cr and Ti concentrations were twice those of unweathered bedrock (+ 1SD). Elevated concentrations of these elements imply the presence of mafic precursor rock such as commonly present in bedrocks' mafic enclaves. All such excluded samples are marked in grey color in Figs. 3 & A1, and mainly affect only the lower section of the south-facing Nahuelbuta profile.

The concentration of K throughout the entire regolith profiles in Santa Gracia is three-fold higher than K contained in local bedrock samples (Oeser et al., 2018). We thus assume that the K concentration in the bedrock samples of Santa Gracia as determined by Oeser et al. (2018) underestimates the actual occurring K concentration of local bedrock. Thus, $\tau^K$ has been calculated using published values for K and Zr concentration from a study nearby (Miralles González, 2013).

**Weathering fluxes**

To estimate elemental release fluxes from regolith (Eq. 3) for each study site, the most negative τ-values from the shallowest mineral-soil sample of each regolith profile were used (red-circled symbols in Fig. A1). This practice is common in eroding regolith, where the loss indicators τ and CDF represent the integrated mass loss over the time and depth interval that a given sample moved from bedrock reference level to its present position (Brantley and Lebedeva, 2011; Ferrier et al., 2010; Hewawasam et al., 2013; Uhlig and von Blanckenburg, 2019). The elemental chemical weathering flux ($W_{regolith}^X$) at each study site has been averaged. Because not all of this flux might be within reach of plant roots (e.g. if a fraction is lost into deep groundwater), this is an upper estimate of the nutrient supply from rock into vegetation. $W_{regolith}^X$ is reported in Table 3.

**Ecosystem nutrient uptake fluxes**

Total ecosystem nutrient uptake fluxes ($U_{total}^X$) have been evaluated using Eq. 4 and are reported in Table 3. Because we compare these to the weathering fluxes that integrate over several millennia, we estimate uptake fluxes that are representative for the Holocene. Net primary productivity (NPP), has been derived from a dynamic vegetation model (LPJ-GUESS) simulating vegetation cover and composition during the Holocene (Werner et al., 2018) and is reported in Table 1. Biomass production was estimated from NPP(C) by assuming that dry biomass consists of 50 wt% carbon. To obtain the element-specific uptake rate $U_{total}^X$, NPP is multiplied with the bulk concentration of X in the plants $[X]_{Plant}$.

The sampling and analyses of roots was not done in this study, because of the difficulties in obtaining entire roots or representative root segments from a specific tree or shrub including fine roots. For elemental analysis this difficulty is compounded by the need to remove any remaining soil particles or attached precipitates that might bias measured concentrations. To nevertheless estimate bulk plant elemental composition, we applied the dimensionless organ growth quotients GL/GS (leaf growth relative to stem growth) and GL/GR (leaf growth relative to root growth) in accordance with Niklas and Enquist (2002). This estimation invokes several assumptions: (1) Roots biomass growth contributes little to total plant growth, namely 9% in angiosperms and 17% in gymnosperms (Niklas and Enquist, 2002). We thus

treat roots and stem/ twig as one plant compartment. In total, the pooled growth of root, stem, and twig amounts to 68% and 52% of relative growth in angiosperms and gymnosperms, respectively. (2) Differences in biomass allocation are relevant only between angiosperms and gymnosperms and not between single plant species of a given class. (3) The pattern of relative growth and standing biomass allocation holds true across a minimum of eight orders of magnitude of species size (Niklas and Enquist, 2002). We thus adapted the organ growth quotients from the work of Niklas and Enquist (2002), such that we only differentiate between the growth rate of leaves and stem, respectively, and the adapt these quotients between angiosperms and gymnosperms. The bulk elemental ecosystem composition (Table 4) has been determined by weighting the averaged elemental composition for each sampled plant for their relative abundance in the respective ecosystem.

**Inventories**

The inventories for the bio-available fraction ($I^X_{bio-av.}$) and in bulk regolith ($I^X_{bulk}$) have been calculated using Eq. 8 and are reported in Table A1. $I^X_{bio-av.}$ was determined for both the bio-available fraction in soil (comprised of the A and B horizon; $I^X_{bio-av, soil}$) and saprolite of 1m thickness ($I^X_{bio-av, sap}$). For the calculation of all inventories we used the soils' bulk density determined by Bernhard et al. (2018). $I^X_{bulk}$ is comprised of elements contained in fine-earth material and in fragmented rocks and coarse material (e.g. core stones). We derive the relative amount of coarse material of each depth increment from Bernhard et al. (2018) and allocate them the bedrocks' chemical composition (Data Table S1). If information on either bulk density or the relative amount of coarse material was unavailable, the respective horizons' average has been used for the calculation of $I^X_{i,j}$. In none of the eight regolith profiles is the depth to unweathered bedrock known. Thus, for comparison purposes, we calculated the inventories of the bio-available fraction in saprolite ($I^X_{bio-av, sap}$) and in bulk regolith ($I^X_{bulk}$) to the depth of the respective regolith profile and normalized this value to the arbitrary value of 1 m.

**Nutrient recycling factor**

We call the ratio of nutrient uptake to nutrient supply by weathering the "nutrient recycling factor" $Rec^X$ which was calculated using Eq. 7 and is reported in Table 5. Importantly, as defined, this factor ratios fluxes between entire regolith and total uptake into the entire vegetation cover (the same rationale as used by Cleveland et al., 2013 for the inverse; the "new" fraction of P). $Rec^X$ represents a minimum estimate as some fraction of $W^X_{regolith}$ will bypass nutrient uptake by plants if it is drained directly via groundwater into streams. $Rec^X$ might represent an underestimate for some elements that are returned to soil by stem-flow or throughfall. According to e.g. Wilcke et al. (2017), these fluxes are generally highest for K compared to other elements. $Rec^X$ might also be an overestimate, if a substantial fraction of nutrient is eroded by leaf litter and other plant debris after uptake, rendering it unavailable for recycling (Oeser and von Blanckenburg, 2020a).

In Pan de Azúcar, where atmospheric deposition ($Dep^X_{dry}$ and $Dep^X_{wet}$) is known to be an important component of ecosystem element budgets (e.g. increasing $\tau$-values towards the profiles top in absence of bio-lifting of elements and field observation; Oeser et al., 2018) we need to consider these inputs in addition to the weathering release fluxes ($W^X_{regolith}$). Thus, to account for all potential sources of elements available for plant uptake, the nutrient recycling factor in Pan de Azúcar is given as:

$$Rec^X = \frac{U^X_{total}}{W^X_{regolith} + Dep^X_{wet} + Dep^X_{dry}}$$

Atmospheric deposition fluxes have been estimated by determining the absolute difference between the lowest $\tau$-value in the shallowest mineral-soil sample and the highest $\tau$-value in the soil profile above it. Further, we assume that elemental gains (i.e. increasing $\tau$-values) in the regolith profiles are attributed solely to atmospheric deposition. We test these estimates for atmospheric depositional fluxes by placing the elemental gains in proportion to the initially determined weathering release fluxes ($W^X_{regolith}$, Eq. 3; Table 3).

**Uncertainty estimation of nutrient fluxes**

The analytical uncertainty of measured samples and certified international reference materials are reported in section "Analytical methods" and in the data publication Oeser and von Blanckenburg ( 2020b).

The uncertainties on the nutrient fluxes of $W^X_{regolith}$ and $U^X_{total}$ were estimated by Monte Carlo simulations in which 20 000 random data sets were sampled within the standard deviation of all input parameters using a Box-Muller transformation (Box and Muller, 1958). The simulation of each regolith profiles'

$W^X_{regolith}$ incorporates the SD of the average soil denudation rate D (Table 1), the SD of the concentration of the element of interest in bedrock (Data Table S1), and 3% relative uncertainty on the element concentration in regolith samples. In the case of $U^X_{total}$ the SD of the respective study sites' NPP and the SD of the chemical composition of the weighted plants (Table 4) were used. The resultant uncertainties on both fluxes are reported in Table 3.

**Appendix B: Data presentation and Statistical analyses**

**Replication**

We present our results on nutrient fluxes, inventories, and turnover times as study-site averages for synthesis reasons only. Indeed, at each study site four replicate regolith profiles have been analyzed in previous studies. Within a given site, these profiles show no significant differences in chemistry and

915 pedogenic properties (Bernhard et al., 2018; Oeser et al., 2018). In this study we focused on two regolith profiles situated on opposing slopes (north- and south facing midslope profiles) to account for the variations in substrate and/ or the effects of insolation and microclimate on weathering and nutrient uptake by plants. However, these profiles are natural replicates and are considered independent from each other.

**Statistical analysis**

An analysis of variance (ANOVA) was performed to evaluate how denudation rate (D), the chemical depletion fraction (CDF), soil weathering rate (W), and the elemental weathering rates for Ca, K, Na, P, and Si ($W^X_{regolith}$) vary among sites. Variance homogeneity was tested using Levene's Test before applying

ANOVAs and pair-wise differences were assessed using Tukey's HSD test. In these, p values ≤ 0.1 were considered as significant. The correlations between D, MAP, NPP, and the degree (CDF, $\tau^X$) and rate (W, $W^X_{regolith}$) of weathering were evaluated using Pearson's correlation coefficients. To test for the significance of D on these weathering parameters, Pearson's correlation coefficients were evaluated twice: with (Table A4) and without La Campana (Table A5). This test is possible because of the high denudation rate of this site which originates from the steepest relief of all sites (Oeser et al., 2018; Schaller et al., 2018b; van Dongen et al., 2019). The sample set comprises the tested parameters for each regolith profile and each regolith profile is considered independent from each other (i.e. n = 8). Statistical analyses were conducted using the statistics packages included in the software OriginPro (Version 2020).

Given the small sample size (n = 8), test for equal variances failed. Still, overall ANOVA showed all weathering patterns (except for $W^K_{regolith}$) differed among sites on the total population (Table A6). However, post-hoc comparisons indicated that sites did not always differ, and that differences between sites varied for the different weathering parameters (Table A6). Particularly, few statistically significant differences exist between the semi-arid Santa Gracia, the mediterranean La Campana and humid-temperate Nahuelbuta. In these three sites the weathering release fluxes do not differ significantly (Table A6) despite massive differences in D, MAP, and NPP (Table 1).

**9 Sample availability**

All sample metadata are already available on a public server using unique sample identifiers in form of the "International Geo Sample Number" (IGSN).

**10 Author contributions**

R.A. Oeser conducted field sampling, analyzed samples, interpreted data, and wrote text. F. von Blanckenburg designed the study, selected the study sites, interpreted data, and wrote text.

**11 Competing financial interests**

The authors declare no competing financial interests.

**12 Additional information**

Supplementary data tables are available at GFZ data services (Oeser and von Blanckenburg, 2020b).

**13 Acknowledgements**

The study was conducted within the framework of the priority program of the Deutsche Forschungsgemeinschaft "EarthShape: Earth Surface Shaping by Biota" (DFG-SPP 1803; http://www.earthshape.net). R.A. Oeser and F. von Blanckenburg are grateful for funding. We thank Leandro Paulino (Departamento de Suelos y Recursos Naturales, Universitad de Concepción, Chile) and Kirstin Übernickel for managing the priority program and Todd Ehlers (both Institute for Geosciences, Universität Tübingen, Germany) for its co-coordination. We acknowledge CONAF in Chile for providing us with the opportunity to work in the national parks of Pan de Azúcar, La Campana, and Nahuelbuta. We also thank CEAZA for facilitating access to the Reserva Natural Santa Gracia. We are grateful to J. Boy (Soil Sciences, Leibniz Universität Hannover, Germany) for discussions, and D. Uhlig (Institute of Bio- and Geosciences, Forschungszentrum Jülich, Germany), Michaela Dippold (Department of Crop Sciences, Georg-August University Goettingen, Germany), Matthew Winnick (Department of Geosciences, University of Massachusetts, USA), and Patrick Frings (Section "Earth Surface Geochemistry", GFZ German Research Centre for Geosciences, Germany) for informal reviews of the text. We thank the three anonymous referees and Marijn van de Broeck and his MSc students for their detailed critique of our work which led us to revise the manuscript with the aim of attempting to avoiding the pitfalls emerging when working across disciplines.

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

**15 Figures**

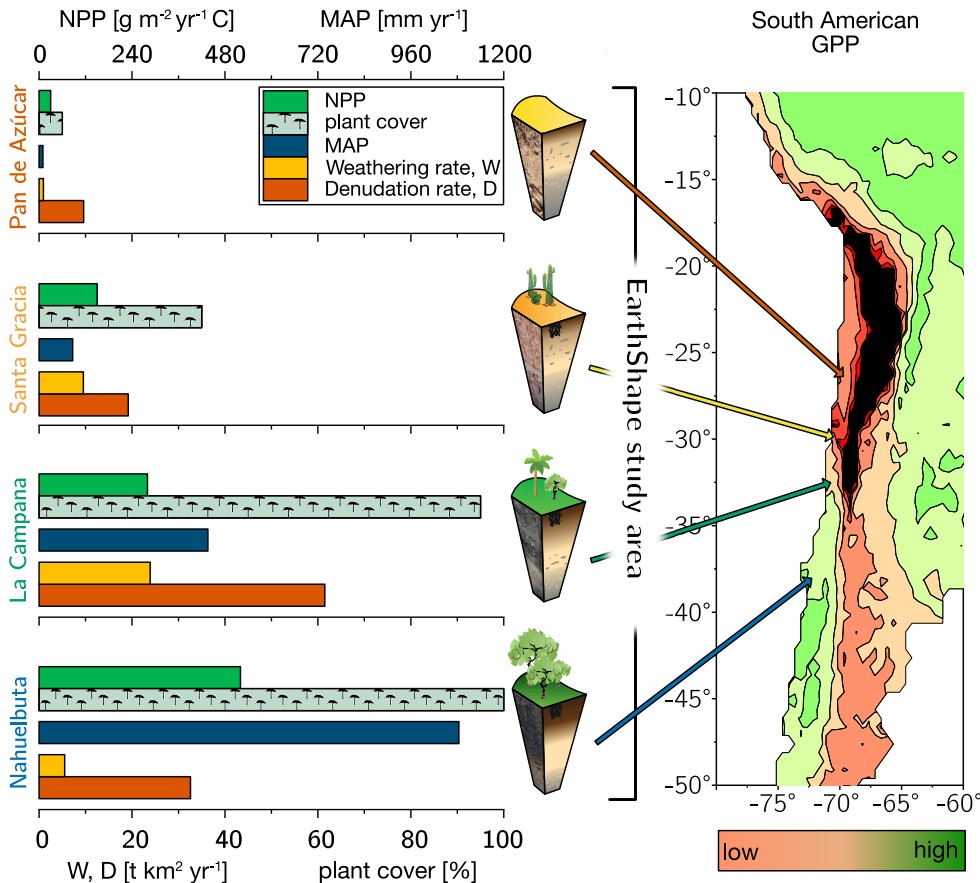

**Figure 1 The climate and vegetation gradient of the four EarthShape study sites (from arid to humid: Pan de Azúcar, Santa Gracia, La Campana, and Nahuelbuta). Left: Net primary productivity (NPP), plant cover, annual precipitation (MAP). Denudation rate (D) and weathering rate (W) were determined with cosmogenic [10]Be. Right: Position of the four study sites in South America and their respective gross primary productivity (GPP) derived from the FLUXNET data base (Jung et al., 2011). Black colour refers to very low GPP in the Atacama Desert. The uncertainties are not shown for clarity. They are provided in Table 1.**

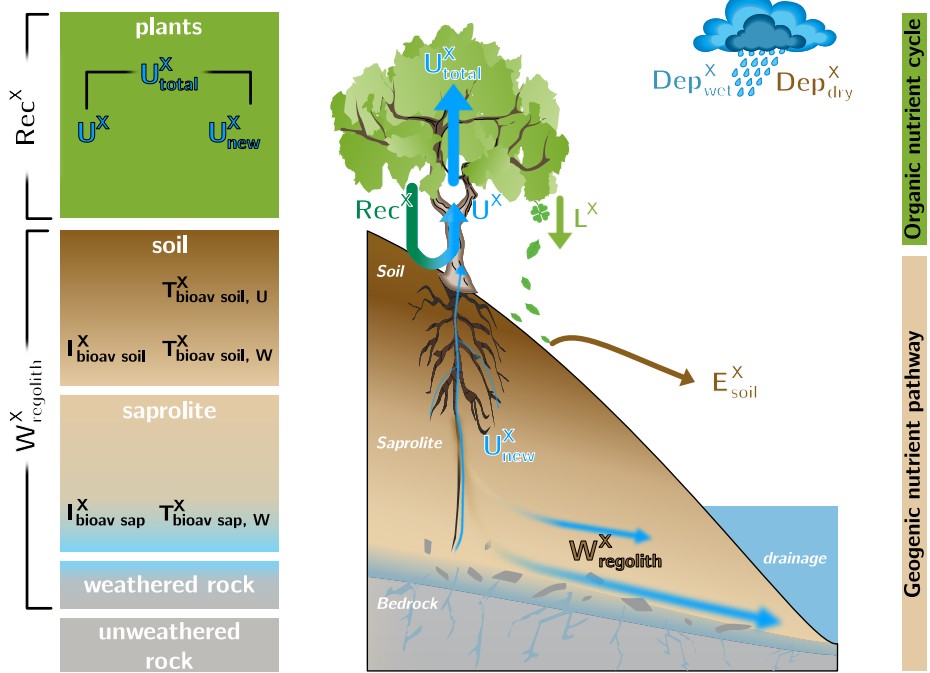

**Figure 2 Conceptual framework of an ecosystem comprising the "geogenic nutrient pathway" and the "organic nutrient cycle" (modified after Uhlig and von Blanckenburg, 2019). Whereas the former is mainly set by mineral nutrient release by weathering ($W_{regolith}^X$) and to a minor extent by atmospheric wet- ($Dep_{wet}^X$) and dry deposition ($Dep_{dry}^X$), the organic nutrient cycle is mainly affected by nutrient re-utilization (i.e. recycling; $Rec^X$) from organic matter. Left: The different compartments (i.e. rock, saprolite, soil, and plants) are shown as boxes. They include the metrics used to quantify their properties such as the inventory $I_{bulk}^X$ and turnover time $T_{i,j}^X$ of element X in compartment j. Right: The compartments are linked by fluxes (arrows) with the thickness of them denoting their relative proportions. $E_{soil}^X$ denotes to erosion of soil.**


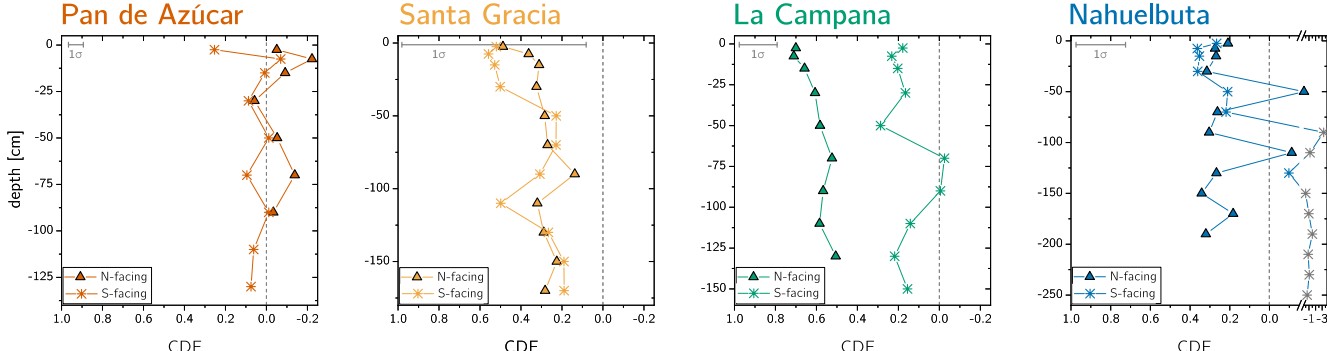

**Figure 3 Chemical depletion fraction (CDF) for each study sites' north- and south- facing profile. The accuracy of the absolute CDF values is limited by the variability in the bedrocks' Zr concentration in the respective study sites and are indicated as grey 1 σ bar (Data Table S1). The grey symbols correspond mainly to saprolite samples in the south-facing regolith profile in Nahuelbuta and are excluded from further consideration. Note that in Nahuelbuta a different scaling compared to the other study sites applies after the axis break.**


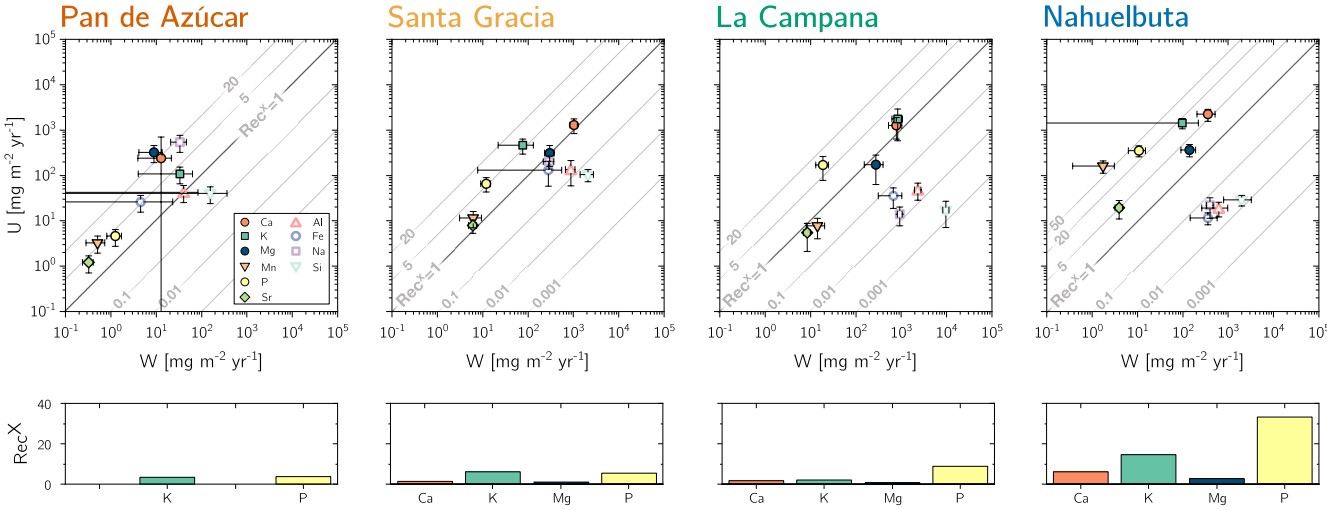

**Figure 4 Chemical weathering flux ($W^X_{regolith}$) and plant nutrient-uptake fluxes ($U^X_{total}$) for Pan de Azúcar, Santa Gracia, La Campana, and Nahuelbuta (from left to right) for mineral nutrients. Grey contour lines emphasize the nutrient recycling factor ($Rec^X$), which is the ratio of $U^X_{total}$ to $W^X_{regolith}$. Uncertainty bars show 1SD. Differences in nutrient recycling factors for Ca, K, Mg, and P among the four study sites are highlighted in the lower panels. Note that here we use the $Rec^X$ calculated for $W^X_{regolith}$ from silicate weathering only. In Table 5 and Fig. A4 we also show $Rec^X$ including atmospheric inputs. Because Pan de Azúcar Ca and Mg inputs are exclusively atmospheric their $Rec^X$ are overestimated and thus not plotted on the lower left panel.**



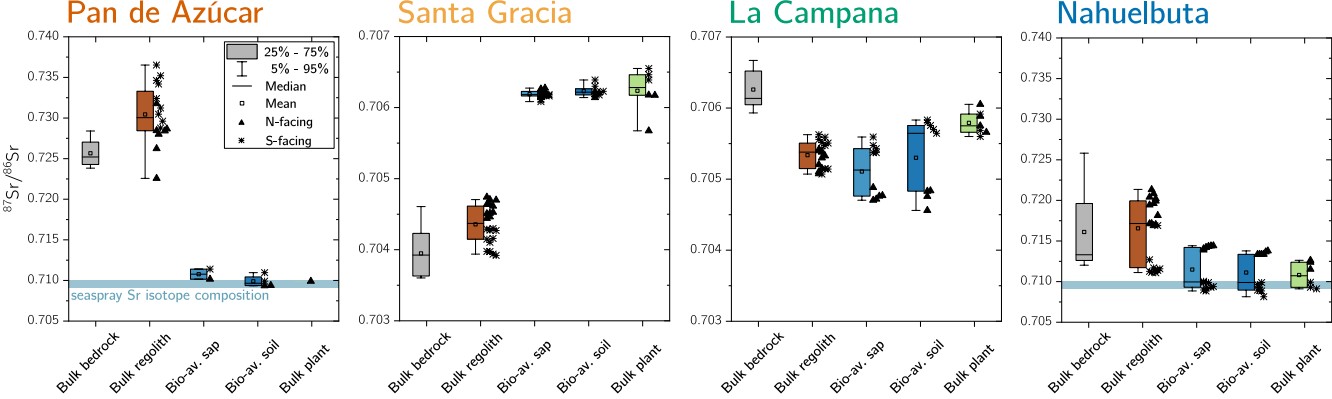

**Figure 5** Average $^{87}Sr/^{86}Sr$ isotope composition of bedrock, bulk regolith, and the bio-available fraction in saprolite, soil, and plants in Pan de Azúcar, Santa Gracia, La Campana, and Nahuelbuta. The $^{87}Sr/^{86}Sr$ isotope ratios of bulk plant (green) are weighted according to the single species' organs relative growth rate (see Table 4 for weighting parameters). Whiskers span 90% of the respective data set. On the boxes' right-hand site, the differences between north- and south-facing regolith profiles are depicted. Note that bulk regolith samples in Nahuelbuta with anomalously low Zr concentrations have been excluded from this analysis as they are suspected to comprise a different parent rock. Y-axis covers broader range in Pan de Azúcar and Nahuelbuta than in Santa Gracia and La Campana.

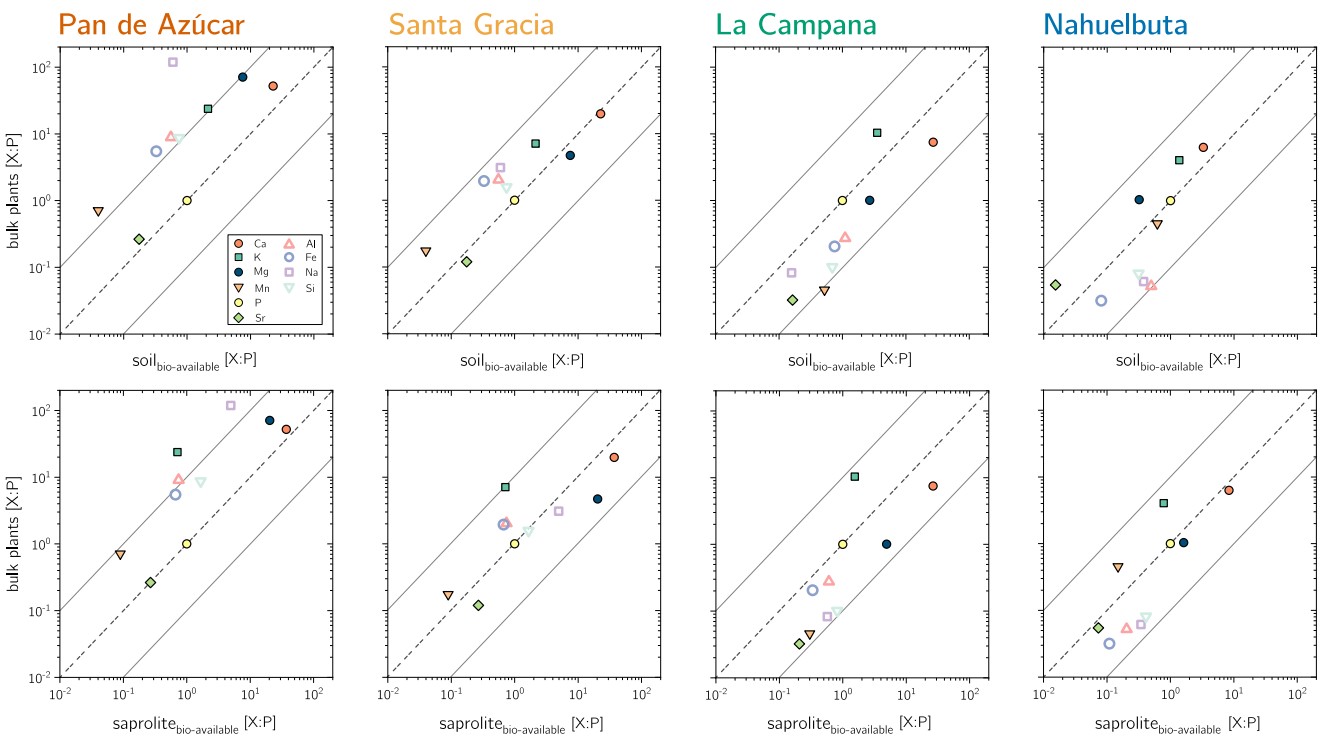

**Figure 6** P-normalized element composition for bulk plants and the bio-available fraction in soil and saprolite. Solid grey lines reflect the 10 x P and 0.1 x P concentration, respectively. Note that with increasing recycling from Santa Gracia to Nahuelbuta, the bio-available fractions' X:P successively approaches X:P in vegetation.

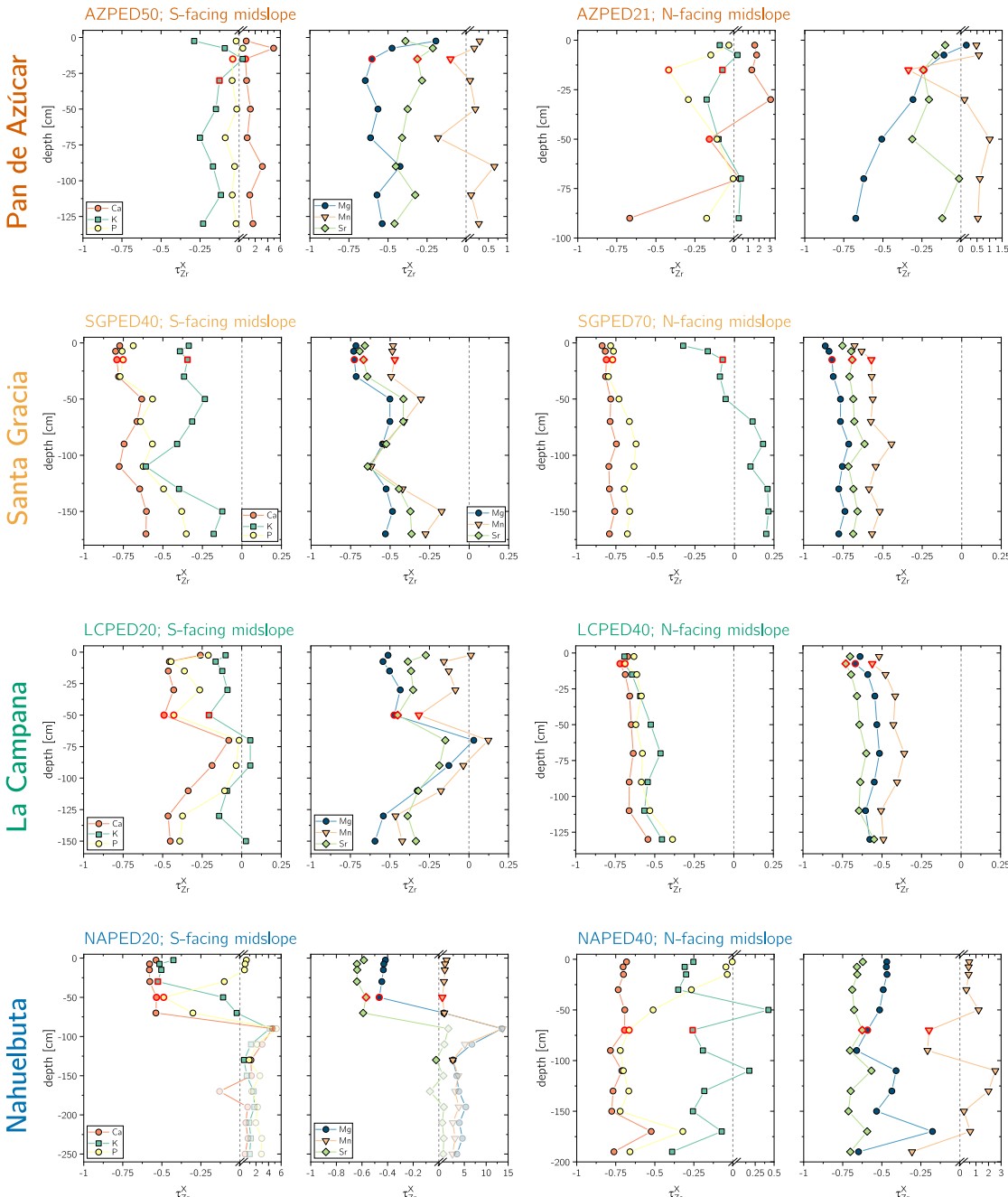

**Figure A1. Depth distribution of the elemental loss and gain fractions (i.e. elemental mass transfer coefficient, τ). The vertical dashed line indicates $\tau_{Zr}^{X} = 0$ and represents unweathered parent bedrock. τ-values corresponding to the shallowest mineral soil samples are highlighted with a red rim. Grey symbols in Nahuelbuta are discarded due to the samples' anomalous low Zr concentration. Note that these τ-values differ from those reported in Oeser et al., 2018, because in this work they have been calculated relative to the initial bedrocks' chemical composition.**


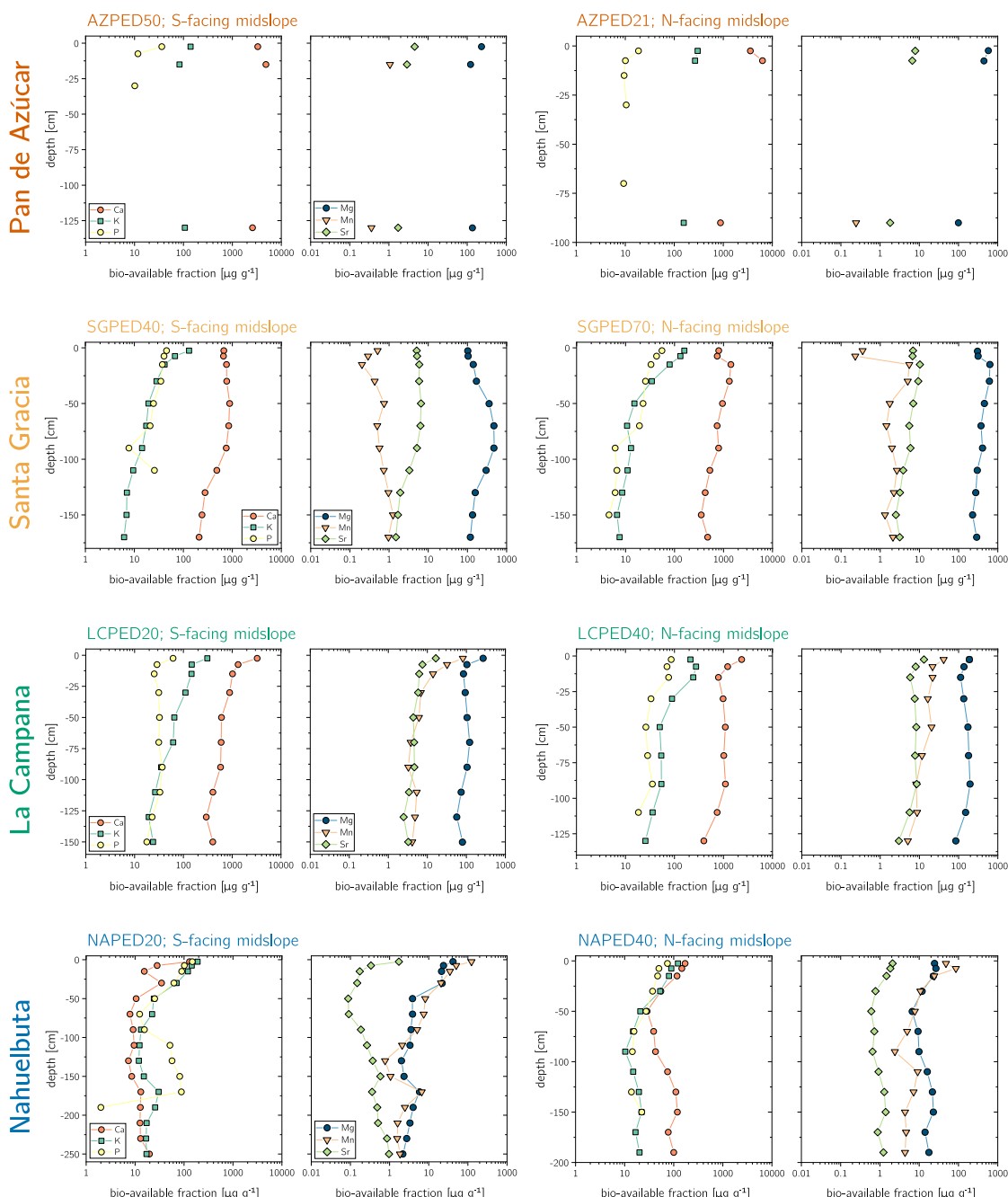

**Figure A2. Depth distribution of the concentration of sequentially extracted bio-available fraction of mineral nutrients including Sr, comprised of the water soluble (18 MΩ Deionized H$_2$O) and the exchangeable (1 M NH$_4$OAc) fraction. P-accessibility in the bio-available fraction has been determined by Brucker and Spohn (2019) using a modified Hedley sequential P fractionation method. Note that in Pan de Azúcar the acquisition of the bio-available fraction was only possible on three samples per site. Data gaps do**
**occur if both extractions of one sample were below limit of detection.**

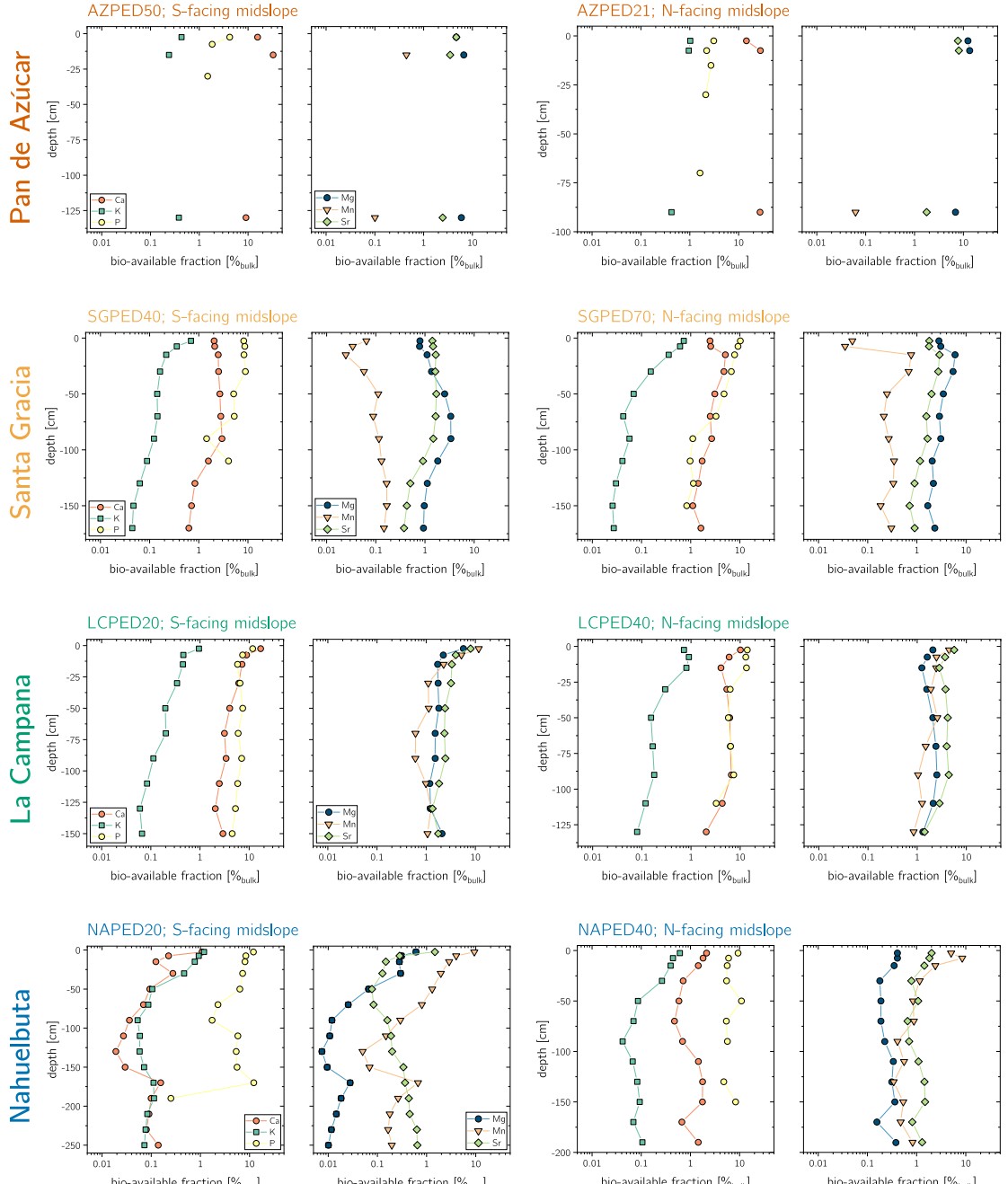

**Figure A3. Depth distribution of the sequentially extracted bio-available fraction of mineral nutrients including Sr relative to their respective amount contained in bulk regolith, comprised of the water soluble (18 MΩ Deionized H₂O) and the exchangeable (1 M NH₄OAc) fraction. P-accessibility in the bio-available fraction has been determined by Brucker and Spohn (2019) using a modified Hedley sequential P fractionation method. Note that in Pan de Azúcar the acquisition of the bio-available fraction was only possible on three samples per site. Data gaps do occur if both extractions of one sample were below limit of detection.**


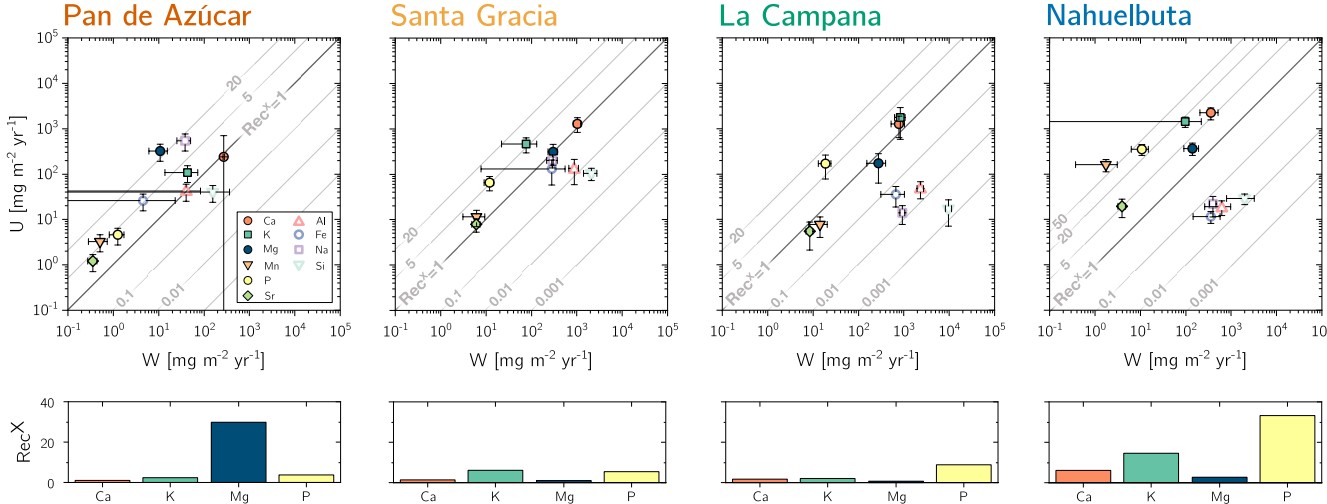

**Figure A4.** Chemical weathering flux ($W^X_{regolith}$) and ecosystem nutrient-uptake fluxes ($U^X_{total}$) for Pan de Azúcar, Santa Gracia, La Campana, and Nahuelbuta (from left to right). Weathering-release fluxes for Ca, K, Mg, Na, and Sr in Pan de Azúcar have been complemented by atmospheric depositional fluxes such that the total amount of available nutrients increase by 95, 22, 18, 12, and 10%, respectively. Grey contour lines emphasize the nutrient recycling factor ($Rec^X$), which is the ratio of $U^X_{total}$ to $W^X_{regolith}$. Uncertainty bars show 1SD. Differences in nutrient recycling factors for Ca, K, Mg, and P among the four study sites are highlighted in the lower panels.


**16 Tables**

**Table 1 Characteristics of the four EarthShape study sites and soil profile names in Pan de Azúcar, Santa Gracia, La Campana, and Nahuelbuta.**

| | | Pan de Azúcar | | Santa Gracia | | La Campana | | Nahuelbua | | Reference |
|---|---|---|---|---|---|---|---|---|---|---|
| | | AZPED21 | AZPED50 | SGPED70 | SGPED40 | LCPED40 | LCPED20 | NAPED40 | NAPED20 | |
| **Latitude** | | 26.1093 S | 26.1102 S | 29.7612 S | 29.7574 S | 32.9573 S | 32.9559 S | 37.8090 S | 37.8077 S | † |
| **Longitude** | | 70.5491 W | 70.5493 W | 71.1656 W | 71.1664 W | 71.0643 W | 71.0635 W | 73.0138 W | 73.0135 W | † |
| **Altitude** | [m a.s.l.] | 343 | 330 | 690 | 682 | 734 | 730 | 1219 | 1239 | * |
| **Slope** | [°] | 25 | 40 | 15 | 25 | 12 | 23 | 13 | 15 | ‡ |
| **Aspect** | | N-facing | S-facing | N-facing | S-facing | N-facing | S-facing | N-facing | S-facing | †, ‡ |
| **Mean annual temperature (MAT)** | [°C] | 18.1 | | 16.1 | | 14.9 | | 14.1 | | § |
| **Mean annual precipitation (MAP)** | [mm yr$^{-1}$] | 10 | | 87 | | 436 | | 1084 | | § |
| **Lithology** | | granite | | diorite | | granodiorite | | granodiorite | | † |
| **Mineralogy*** | [Vol%] | Quartz xxx, Plagioclase x, Pyroxene -, K-feldspar xxx, Biotite x, Amphibole - | | Quartz x, Plagioclase xx, Pyroxene xx, K-feldspar xxx, Biotite -, Amphibole x | | Quartz xx, Plagioclase x, Pyroxene -, K-feldspar xxx, Biotite x, Amphibole - | | Quartz xx, Plagioclase xx, Pyroxene x, K-feldspar xxx, Biotite x, Amphibole - | | † |
| **Soil type** | | Regosol | Regosol | Cambisol | Leptosol | Cambisol | Cambisol | umbric Podzol | orthodystric Umbrisol | ‡ |
| **Soil thickness** | [cm] | 20 | 20 | 35 | 45 | 35 | 60 | 60 | 70 | † |
| **Soil pH**** | | 8.1 ± 0.1 | 8.1 ± 0.1 | 6.0 ± 0.3 | 6.1 ± 0.3 | 5.2 ± 0.3 | 5.0 ± 0.3 | 4.7 ± 0.1 | 4.3 ± 0.2 | ‡ |
| **Cation exchange capacity (CEC)**** | [µmolc g$^{-1}$] | - | - | 108.5 ± 50.2 | 64.6 ± 23.4 | 86.4 ± 43.1 | 72.7 ± 62.1 | 21.0 ± 15.4 | 38.2 ± 24.7 | ‡ |
| **Catchment-wide denudation rate (D)** | [t km$^{-2}$ yr$^{-1}$] | 7.7 ± 0.7 | | 9.2 ± 0.8 | | 200 ± 22 | | 27.2 ± 2.4 | | \|\| |
| **Soil denudation rate (D$_{soil}$)** | [t km$^{-2}$ yr$^{-1}$] | 8.2 ± 0.5 | 11.0 ± 0.7 | 15.9 ± 0.9 | 22.4 ± 1.5 | 69.2 ± 4.6 | 53.7 ± 3.4 | 17.7 ± 1.1 | 47.5 ± 3.0 | †, # |
| **Soil weathering rate (W)** | [t km$^{-2}$ yr$^{-1}$] | -1.0 ± 0.1*** | 0.9 ± 0.2 | 7.2 ± 4.7 | 11.9 ± 7.6 | 45.9 ± 8.0 | 20.0 ± 3.1 | 3.5 ± 0.9 | 7.5 ± 3.1 | †, # |
| **Soil erosion rate (E)** | [t km$^{-2}$ yr$^{-1}$] | 9.1 ± 0.5 | 10.1 ± 0.7 | 8.7 ± 4.8 | 10.5 ± 7.7 | 23.4 ± 9.2 | 33.8 ± 4.6 | 14.2 ± 1.4 | 40.0 ± 4.3 | †, # |
| **Chemical depletion fraction (CDF)** | | -0.1 ± 0.0 | 0.1 ± 0.0 | 0.5 ± 0.3 | 0.5 ± 0.3 | 0.7 ± 0.1 | 0.4 ± 0.1 | 0.2 ± 0.1 | 0.2 ± 0.1 | †, # |
| **Vegetation cover** | [%] | <5 | | 30 – 40 | | 95 | | 100 | | ‡ |
| **Vegetation types** | | Cristaria integerrima, Nolana mollis, Perityle sp., Stipa plumosa, Tetragonia maritima | | Adesmia sp., Cordia decandra, Cumulopuntia sphaerica, Eulychnia acida, Proustia cuneifolia, Senna cumingii | | Aristeguietia salvia, Colliguaja odorifera, Cryptocarya alba, Jubaea chilensis, Lithraea caustica | | Araucaria araucana, Chusquea culeon, Nothofagus antarctica | | ‡ |
| **Net primary production (NPP)** | [gc m$^{-2}$ yr$^{-1}$] | 30 ± 10 | | 150 ± 40 | | 280 ± 50 | | 520 ± 130 | | ¶ |

* Estimation of mineral abundances based on thin section microscopy: -: absence, x: presence (<10 Vol%), xx: abundant (10-35 Vol%), xxx: very abundant (>35 Vol%)

** Denoting to regolith-profile averages

*** N-facing slope in Pan de Azúcar yield negative CDF and hence weathering rates due to the input of seaspray

† Oeser et al. (2018); ‡ Bernhard et al. (2018); § Ministerio de Obras Públicas (2017); || van Dongen et al. (2019); # Schaller et al. (2018); ¶ Werner et al. (2018)

**Table 2 Glossary of metrics for the parameterization of the geogenic nutrient pathway and organic nutrient cycle in terrestrial ecosystems after Uhlig and von Blanckenburg (2019).**

### *Total mass fluxes (in e.g. t km$^{-2}$ yr$^{-1}$)*

| | | |
|---|---|---|
| Eq. (1) | $D = E + W$ | denudation rate; the sum of chemical weathering and physical erosion |
| | $E$ | physical erosion; physical removal of primary and secondary minerals along with biogenic material |
| Eq. (2) | $W = D \times CDF$ | chemical weathering rate; net-chemical release flux from minerals as some fraction of which is being incorporated into secondary minerals and pedogenic (hydr-)oxides |
| | GPP | gross primary production; gross carbon input into biomass |
| | NPP | net primary productivity; net-carbon fixation by biomass |

### *Elemental fluxes (in e.g. mg m$^2$ yr$^{-1}$)*

| | | |
|---|---|---|
| Eq. (3) | $W^X_{regolith} = D \times [X]_{parent} \times (-\tau^X_{Xi})$ | Chemical weathering flux of element X; release flux of X from minerals minus the flux of incorporation of X into secondary minerals and oxides |
| Eq. (4) | $U^X_{total} = \dfrac{NPP \times [X]_{plant}}{[C]_{plant}}$ | Total nutrient uptake flux of element X; uptake of X by trees at the ecosystem scale, where $[C]_{plant}$ denotes the carbon concentration in dry mass, typically 50 weight% |
| | $Dep^X_{dry}$ | Atmospheric dry deposition of element X |
| | $Dep^X_{wet}$ | Atmospheric wet deposition of element X as rainfall |

### *Elemental mass fractions and flux ratios (dimensionless)*

| | | |
|---|---|---|
| Eq. (5) | $CDF = 1 - \dfrac{[X_i]_{parent}}{[X_i]_{weathered}}$ | chemical depletion fraction; fractional mass loss by dissolution of elements from the regolith |
| Eq. (6) | $\tau^X = \dfrac{[X]_{weathered}}{[X]_{parent}} \times \dfrac{[X_i]_{parent}}{[X_i]_{weathered}} - 1$ | elemental mass transfer coefficient; elemental loss or gain relative to unweathered bedrock |
| Eq. (7) | $Rec^X = \dfrac{U^X_{total}}{W^X_{regolith}}$ | nutrient recycling factor; number of times, element X is re-utilized from plant litter after its initial release from rock weathering |

### *Elemental inventories (in e.g. g m$^{-2}$ or kg m$^{-2}$)*

| | | |
|---|---|---|
| Eq. (8) | $I_j = \displaystyle\int_{z=a}^{z=b} [X_j] \times \rho \, dz$ | Inventory of element X in compartment j |
| | $I^X_{bio-av.\,soil}$ | Inventory of element X in the bio-available fraction in soil |
| | $I^X_{bio-av.\,sap}$ | Inventory of element X in the bio-available fraction in saprolite |
| | $I^X_{bulk}$ | Inventory of element X in bulk regolith |

### *Elemental turnover times (in e.g. yr)*

| | | |
|---|---|---|
| Eq. (9) | $T^X_{i,j} = \dfrac{I^X_i}{j}$ | Turnover time of element X in compartment i with respect to input or output flux j; the ratio of total stock of element X in i to input or output flux j |
| | $T^X_{bio-av,\,U}$ | Turnover time of element X in the forest floor with respect to uptake into trees; mean time a nutrient rest in the forest floor before re-utilization by forest trees |
| | $T^X_{bio-av,\,W}$ | Turnover time of element X in the bio-available fraction in regolith with respect to adsorption onto clay minerals; mean time over which the inventory of the bio-available fraction is replenished by chemical silicate weathering in the absence of other gains or losses |

**Table 3** Elemental weathering fluxes ($W^X_{regolith}$) and ecosystem nutrient uptake fluxes ($U^X_{total}$) in Pan de Azúcar, Santa Gracia, La Campana, and Nahuelbuta along with the respective study site's average soil denudation rate (D) and net primary productivity (NPP).

| Study site | D [t km⁻² yr⁻¹] | NPP [g_C m⁻² yr⁻¹] | Al | Ca | Fe | K | Mg [mg m⁻² yr⁻¹] | Mn | Na | P | Si | Sr |
|---|---|---|---|---|---|---|---|---|---|---|---|---|
| **Pan de Azúcar** | | | | | | | | | | | | |
| $W^X_{regolith}$ | 9.6 | | 40 | 13* | 5 | 30 | 9 | 0.5 | 33 | 1.3 | 160 | 0.3 |
| SD | 0.6 | | 43 | 9 | 18 | 30 | 5 | 0.2 | 13 | 0.4 | 210 | 0.1 |
| $U^X_{total}$ | - | 30 | 40 | 200 | 30 | 110 | 300 | 3 | 500 | 5 | 40 | 1.2 |
| SD | - | 10 | 20 | 500 | 10 | 40 | 100 | 1 | 200 | 2 | 20 | 0.5 |
| **Santa Gracia** | | | | | | | | | | | | |
| $W^X_{regolith}$ | 19.2 | | 870 | 1030 | 280 | 80 | 300 | 6 | 290 | 12 | 2100 | 6.1 |
| SD | 1.2 | | 200 | 200 | 270 | 50 | 70 | 3 | 80 | 3 | 680 | 1.3 |
| $U^X_{total}$ | - | 150 | 140 | 1300 | 130 | 500 | 300 | 12 | 200 | 70 | 100 | 8 |
| SD | - | 40 | 80 | 500 | 70 | 200 | 100 | 5 | 60 | 20 | 30 | 3 |
| **La Campana** | | | | | | | | | | | | |
| $W^X_{regolith}$ | 61.5 | | 2330 | 770 | 670 | 840 | 280 | 14 | 930 | 19 | 9700 | 8.5 |
| SD | 4.0 | | 370 | 250 | 350 | 220 | 120 | 6 | 110 | 6 | 1500 | 1.5 |
| $U^X_{total}$ | - | 280 | 50 | 1300 | 40 | 2000 | 200 | 8 | 14 | 170 | 17 | 6 |
| SD | - | 50 | 20 | 600 | 20 | 1000 | 100 | 4 | 6 | 90 | 10 | 3 |
| **Nahuelbuta** | | | | | | | | | | | | |
| $W^X_{regolith}$ | 32.6 | | 620 | 360 | 360 | 100 | 140 | 1 | 400 | 11 | 2000 | 4.0 |
| SD | 2.1 | | 360 | 150 | 210 | 120 | 50 | 3 | 70 | 4 | 1200 | 0.7 |
| $U^X_{total}$ | - | 520 | 19 | 2200 | 12 | 1400 | 400 | 160 | 22 | 350 | 30 | 19 |
| SD | - | 130 | 7 | 700 | 3 | 400 | 100 | 50 | 11 | 100 | 10 | 9 |

* $W^X_{regolith}$ only includes information from AZPED21 (N-facing slope regolith profile) as atmospheric deposition of Ca in the S-facing slope led to (theoretically) negative weathering fluxes.

Uncertainties on weathering fluxes are estimated by Monte-Carlo simulations, where the SD of the respective profile's denudation rate, the SD of the bedrocks' element concentration of interest, and 3% relative uncertainty on the element concentration in regolith samples have been used.

Uncertainties on nutrient uptake fluxes are estimated by Monte-Carlo simulations, where the SD of the respective study site's net primary productivity (NPP) and the SD of the chemical composition of the weighted above-ground living ecosystem have been used (Table 4)

**Table 4 Chemical composition of the above ground living plants. Plant organs have been weighted according to Niklas and Enquist (2002), using the plant organs' relative growth rate (see Appendix A). Relative growth rates and relative abundance of the different plant species can be found in this table's footnotes. The unweighted chemical composition of each plant organ is listed in Data Table S5.**

| Study site | Al | Ca | Fe | K | Mg | Mn | Na | P | Si | Sr |
|---|---|---|---|---|---|---|---|---|---|---|
| | | | | | [µg g$^{-1}$] | | | | | |
| *Pan de Azúcar*[†] | | | | | | | | | | |
| mean | 2700 | 15200 | 1700 | 6900 | 20700 | 210 | 34600 | 290 | 2500 | 80 |
| *SD* | *300* | *1500* | *200* | *700* | *2100* | *20* | *3500* | *30* | *300* | *10* |
| *Santa Gracia*[‡] | | | | | | | | | | |
| mean | 1880 | 17800 | 1800 | 6400 | 4200 | 160 | 2800 | 900 | 1400 | 110 |
| *SD* | *920* | *4400* | *900* | *1600* | *1700* | *50* | *500* | *220* | *300* | *20* |
| *SE (n=15)* | *650* | *2900* | *600* | *1100* | *1000* | *30* | *400* | *140* | *200* | *20* |
| *La Campana*[§] | | | | | | | | | | |
| mean | 340 | 8900 | 250 | 12300 | 1200 | 50 | 100 | 1200 | 120 | 40 |
| *SD* | *120* | *4100* | *110* | *8000* | *700* | *20* | *40* | *600* | *70* | *20* |
| *SE (n=16)* | *70* | *2300* | *70* | *5300* | *400* | *20* | *20* | *400* | *40* | *10* |
| *Nahuelbuta*[¶] | | | | | | | | | | |
| mean | 70 | 8500 | 40 | 5400 | 1400 | 610 | 80 | 1300 | 110 | 70 |
| *SD* | *20* | *1400* | *10* | *500* | *250* | *110* | *30* | *200* | *10* | *30* |
| *SE (n=10)* | *10* | *1000* | *10* | *300* | *180* | *80* | *20* | *100* | *10* | *20* |

Standard deviation and standard error relate to the variability within the data set of each ecosystem. Where natural replicates were not available (i.e. in Pan de Azúcar), 10% relative uncertainty has been assumed.

[†] Pan de Azúcar ecosystem composition: 100% Nolona mollis; 32% and 68% relative leaf and stem growth, respectively, accounting for 5% leaf and 95% stem standing biomass

[‡] Santa Gracia ecosystem composition: 25% each of Asterasia sp., Cordia decandra, Cumulopuntia sphaerica, Proustia cuneifolia; 32% and 68% relative leaf and stem growth assumed for all species, respectively, accounting for 5% leaf and 95% stem standing biomass

[§] La Campana ecosystem composition: 5% each for Aristeguieta salvia and Colliguaja odorifera and 45% each for Cryptocaria alba and Lithraea caustica; 32% and 68% relative leaf and stem growth assumed for all species, respectively, accounting for 5% leaf and 95% stem standing biomass

[¶] Nahuelbuta ecosystem composition: 60% Araucaria araucana, 10% Chusquea culeou, and 30% Nothofagus antarctica; 48% and 52% relative leaf and stem growth assumed for Araucaria araucana, respectively, accounting for 16% leaf and 84% stem standing biomass, 32% and 68% relative leaf and stem growth assumed for Chusquea culeou and Nothofagus antarctica, respectively, accounting for 5% leaf and 95% stem standing biomass.

**Table 5 Nutrient recycling factors in Pan de Azúcar, Santa Gracia, La Campana, and Nahuelbuta. Shown in brackets are the $Rec^X$ prior correction for atmospheric deposition.**

| | $Rec^{Al}$ | $Rec^{Ca}$ | $Rec^{Fe}$ | $Rec^{K}$ | $Rec^{Mg}$ | $Rec^{Mn}$ | $Rec^{Na}$ | $Rec^{P}$ | $Rec^{Si}$ | $Rec^{Sr}$ |
|---|---|---|---|---|---|---|---|---|---|---|
| *Pan de Azúcar* | 1.1 | 1 (19)[*] | 5.8 | 3 (3)[*] | 30 (36)[*] | 6 | 15 (16)[*] | 4 | 0.26 | 3 (4)[*] |
| *SD* | *0.4* | *2* | *0.6* | *1* | *20* | *6* | *15* | *4* | *0.08* | *5* |
| *Santa Gracia* | 0.1 | 1 | 0.4 | 6 | 1 | 1 | 1 | 5 | 0.04 | 1 |
| *SD* | *0.5* | *4* | *0.5* | *3* | *3* | *3* | *1* | *13* | *0.07* | *3* |
| *La Campana* | 0 | 2 | 0.1 | 2 | 1 | 0.5 | 0 | 9 | 0 | 1 |
| *SD* | *0.1* | *2* | *0* | *5* | *1* | *0.6* | *0.1* | *15* | *0.01* | *2* |
| *Nahuelbuta* | 0 | 6 | 0 | 15 | 3 | 190[†] | 0.1 | 30 | 0.01 | 5 |
| *SD* | *0* | *4* | *0* | *3* | *2* | *70* | *0.2* | *20* | *0.01* | *12* |

[*] $Rec^X$ in Pan de Azúcar has been corrected for atmospheric deposition of seaspray, ultimately decreases the recycling rates of weathering-derived nutrients by 95, 22, 18, 12, and 10% for Ca, K, Mg, Na, and Sr, respectively (see supporting information for further explanation and Fig. A6).

[†] values not being considered in the discussion as $W_{regolith}^{Mn}$ is potentially biased by high bedrock heterogeneities

**Table A1 Inventories of elements in bulk regolith and the bio-available fraction in soil and saprolite. Apart from phosphorus, the accessibility of these elements was determined using a sequential extraction method described by Arunachalam et al. (1996); Tessier et al. (1979); He et al. (1995). P-accessibility in the bio-available fraction has been determined by Brucker and Spohn (2019) using a modified Hedley sequential P fractionation method. Data Tables S3 & S4 include depth-dependent concentration of the bio-available fraction (pooled) and the Deionized and NH$_4$OAc extractions used for calculation of the inventories.**

| Study site | | Extent* [m] | Al | Ca | Fe | K | Mg | Mn | Na | P | Si | Sr | Σ |
|---|---|---|---|---|---|---|---|---|---|---|---|---|---|
| **Pan de Azúcar** | | | | | | | | | | | | | |
| $I^X_{bio\text{-}av.\,soil}$ | [g m$^{-2}$] | 0.2 | 0.3 | 1440 | n.c. | 53 | 92 | 0.1 | 493 | 3.3 | 19 | 1.5 | 2100 |
| $I^X_{bio\text{-}av.\,sap}$ | [g m$^{-2}$] | 1.0 | 1.7 | 3833 | n.c. | 253 | 244 | 0.6 | 682 | 0.0 | 75 | 3.5 | 5100 |
| $I^X_{bulk}$ | [kg m$^{-2}$] | 1.0 | 136 | 21 | 44 | 65 | 8.6 | 0.5 | 39 | 1.3 | 636 | 0.2 | 950 |
| **Santa Gracia** | | | | | | | | | | | | | |
| $I^X_{bio\text{-}av.\,soil}$ | [g m$^{-2}$] | 0.4 | 12 | 616 | 7.2 | 38 | 221 | 1.4 | 18 | 22 | 19 | 4.6 | 960 |
| $I^X_{bio\text{-}av.\,sap}$ | [g m$^{-2}$] | 1.0 | 23 | 1179 | 21 | 23 | 651 | 2.9 | 159 | 21 | 53 | 8.5 | 2100 |
| $I^X_{bulk}$ | [kg m$^{-2}$] | 1.0 | 183 | 130 | 75 | 29 | 42 | 1.5 | 61 | 1.6 | 532 | 1.0 | 1100 |
| **La Campana** | | | | | | | | | | | | | |
| $I^X_{bio\text{-}av.\,soil}$ | [g m$^{-2}$] | 0.5 | 37 | 673 | 24 | 90 | 79 | 11 | 6.7 | 28 | 34 | 4.5 | 1000 |
| $I^X_{bio\text{-}av.\,sap}$ | [g m$^{-2}$] | 1.0 | 51 | 1026 | 23 | 70 | 191 | 12 | 31 | 39 | 142 | 8.0 | 1600 |
| $I^X_{bulk}$ | [kg m$^{-2}$] | 1.0 | 118 | 26 | 49 | 46 | 10 | 0.9 | 31 | 0.7 | 456 | 0.3 | 740 |
| **Nahuelbuta** | | | | | | | | | | | | | |
| $I^X_{bio\text{-}av.\,soil}$ | [g m$^{-2}$] | 0.9 | 14 | 60 | 1.8 | 39 | 9.9 | 15 | 17 | 31 | 14 | 0.5 | 200 |
| $I^X_{bio\text{-}av.\,sap}$ | [g m$^{-2}$] | 1.0 | 1.5 | 52 | < 0.5 | 19 | 11 | 3.9 | 13 | 23 | 12 | 0.8 | 140 |
| $I^X_{bulk}$ | [kg m$^{-2}$] | 1.0 | 95 | 15 | 47 | 22 | 13 | 1.0 | 10 | 0.7 | 309 | 0.1 | 510 |

$I^X_{bio\text{-}av,\,soil}$ = inventory of element X in the soil bio-available fraction; extent amounts to maximum soil depth

$I^X_{bio\text{-}av,\,sap}$ = inventory of element X in the saprolite bio-available fraction;

$I^X_{bulk}$ = inventory of element X in bulk regolith

* the extent of the saprolite and regolith inventory have been scaled to 1.0 m for purposes of comparisons between the four study sites and the lack of an absolute measure of the depth of saprolite.

n.c. = not calculated as the respective bio-available fraction (Table S4) was below the limit of calibration of ICP-OES measurements

**Table A2 Average $^{87}Sr/^{86}Sr$ ratio for bulk bedrock, bulk regolith, and the bio-available fraction in saprolite and soil. $^{87}Sr/^{86}Sr$ in bulk plants are weighted by the plant organs' relative growth rate and relative species abundance in the respective ecosystem (see Table 4). Radiogenic Sr composition for each single specimen are reported in Data Tables S2 (bulk regolith samples), S3 (bio-available fraction of saprolite and soil), and S5 (plant samples), respectively.**

| | bulk samples | | bio-available samples | | bulk living plants[†] | Seaspray input[‡] |
|---|---|---|---|---|---|---|
| | $^{87}Sr/^{86}Sr_{rock}$ | $^{87}Sr/^{86}Sr_{regolith}$ | $^{87}Sr/^{86}Sr_{sap}$ | $^{87}Sr/^{86}Sr_{soil}$ | $^{87}Sr/^{86}Sr_{plant}$ | |
| ***Pan de Azúcar*** | 0.7257 | 0.7305 | 0.7108 | 0.7099 | 0.7099 | 93% |
| *SD* | *0.0020* | *0.0036* | *0.0009* | *0.0007* | | |
| ***Santa Gracia*** | 0.7039 | 0.7044 | 0.7062 | 0.7062 | 0.7062 | 43% |
| *SD* | *0.0004* | *0.0003* | *0.0001* | *0.0001* | *0.0003* | |
| ***La Campana*** | 0.7063 | 0.7053 | 0.7051 | 0.7053 | 0.7059 | |
| *SD* | *0.0003* | *0.0002* | *0.0004* | *0.0005* | *0.0002* | |
| ***Nahuelbuta*** | 0.7161 | 0.7162 | 0.7115 | 0.7111 | 0.7111 | |
| *SD* | *0.0065* | *0.0036* | *0.0025* | *0.0023* | *0.0016* | |
| ***Seaspray**** | | | | 0.7092 | | |

[*] Seaspray composition from Pearce et al. (2015)

[†] Standard deviation corresponds to species-to-species differences in $^{87}Sr/^{86}Sr$

[‡] Potential seaspray input into the bio-available fraction derived from a simple two-component mixing equation using bulk bedrock and seaspray as end-members. Substantial seaspray incorporation into the bio-available fraction in La Campana and Nahuelbuta is very unlikely (see text for discussion), therefore not shown.

**Table A3 Turnover times for the soil and saprolite bio-available fraction with respect to the release by weathering and turnover times for bio-available fraction in soil with respect to uptake into plants.**

| study site | Al | Ca | Fe | K | Mg | Mn | Na | P | Si | Sr |
|---|---|---|---|---|---|---|---|---|---|---|
| | | | | | | **[yr]** | | | | |
| *Pan de Azúcar* | | | | | | | | | | |
| $T^X_{bio-av.soil,U}$ | 10 | 6040 | 0 | 490 | 280 | 40 | 910 | 710 | 480 | 1250 |
| $T^X_{bio-av.soil,W}$ | 10 | n.d. | 0 | 1590 | 10300 | 280 | 14800 | 2570 | 120 | 4670 |
| $T^X_{bio-av.sap,W}$ | 40 | n.d. | 0 | 7570 | 27400 | 1240 | 20400 | n.d. | 490 | 10870 |
| *Santa Gracia* | | | | | | | | | | |
| $T^X_{bio-av.soil,U}$ | 90 | 480 | 50 | 80 | 710 | 120 | 90 | 330 | 180 | 590 |
| $T^X_{bio-av.soil,W}$ | 10 | 600 | 30 | 510 | 730 | 230 | 60 | 1850 | 10 | 760 |
| $T^X_{bio-av.sap,W}$ | 30 | 1150 | 80 | 300 | 2160 | 470 | 540 | 1760 | 30 | 1400 |
| *La Campana* | | | | | | | | | | |
| $T^X_{bio-av.soil,U}$ | 780 | 530 | 660 | 50 | 460 | 1420 | 480 | 160 | 1970 | 820 |
| $T^X_{bio-av.soil,W}$ | 20 | 870 | 40 | 110 | 290 | 770 | 10 | 1470 | 3 | 530 |
| $T^X_{bio-av.sap,W}$ | 20 | 1330 | 30 | 80 | 690 | 830 | 30 | 2050 | 10 | 950 |
| *Nahuelbuta* | | | | | | | | | | |
| $T^X_{bio-av.soil,U}$ | 760 | 30 | 160 | 30 | 30 | 90 | 790 | 90 | 490 | 20 |
| $T^X_{bio-av.soil,W}$ | 20 | 170 | 10 | 400 | 70 | 17400 | 40 | 2900 | 10 | 120 |
| $T^X_{bio-av.sap,W}$ | 0 | 150 | 0 | 190 | 80 | 4750 | 30 | 2130 | 10 | 210 |

$T^X_{bio-av.soil,U}$ = turnover time of element X in the soil bio-available fraction with respect to uptake into the ecosystem

$T^X_{bio-av.soil,W}$ = turnover time of element X in the soil bio-available fraction with respect to supply from dissolution of primary minerals and secondary precipitates

$T^X_{bio-av.sap,W}$ = turnover time of element X in the saprolite bio-available fraction with respect to supply from dissolution of primary minerals and secondary precipitates

n.d. = not determined; not determined turnover times because the respective inventory (Table A1) could not be determined


**Table A4 Correlation matrix with Pearson's correlation coefficients and significance (\*\*: p < 0.01, \*: p < 0.05) for net primary productivity (NPP), denudation rate (D), mean annual precipitation (MAP), and indices of total and elemental degree and rate of weathering. Here we treat each regolith profile as independent, i.e. n = 8. Correlation coefficients involve the entire EarthShape study area.**

| | MAP | NPP | CDF | W | $W_{regolith}^{Ca}$ | $W_{regolith}^{K}$ | $W_{regolith}^{Na}$ | $W_{regolith}^{P}$ | $W_{regolith}^{Si}$ | $\tau^{Ca}$ | $\tau^{K}$ | $\tau^{Na}$ | $\tau^{P}$ | $\tau^{Si}$ |
|---|---|---|---|---|---|---|---|---|---|---|---|---|---|---|
| D | 0.39 | 0.47 | 0.54 | **0.86\*\*** | 0.46 | **0.80\*** | **0.94\*\*** | **0.87\*\*** | **0.86\*\*** | 0.33 | **0.66\*** | 0.57 | 0.21 | 0.57 |
| MAP | | **0.98\*\*** | -0.15 | 0.06 | -0.09 | 0.08 | 0.30 | 0.29 | 0.12 | 0.29 | 0.10 | **0.63\*** | 0.20 | -0.05 |
| NPP | | | 0.01 | 0.15 | 0.08 | 0.14 | 0.38 | 0.41 | 0.20 | 0.45 | 0.17 | **0.74\*** | 0.34 | 0.10 |

CDF = chemical depletion fraction; W = soil weathering rate; $W_{regolith}^{X}$ = elemental weathering flux; $\tau^{X}$ = elemental mass transfer coefficient

Note that because CDF and $\tau^{X}$ are per definition different in sign (i.e. a CDF of +1 denote entire depletion, whereas a $\tau^{X}$ of -1 denote entire depletion), $\tau^{X}$ was multiplied by -1 for presentation purposes. Bold numbers denote significant correlation.

**Table A5 Correlation matrix with Pearson's correlation coefficients and significance (\*\*: p < 0.01, \*: p < 0.05) for net primary productivity (NPP), denudation rate (D), mean annual precipitation (MAP), and indices of total and elemental degree and rate of weathering. Correlation coefficients involve the study sites Pan de Azúcar, Santa Gracia, and Nahuelbuta. Here we treat each regolith profile as independent, i.e. n = 6. La Campana has been excluded because it features the steepest relief of all sites which causes elevated denudation rates.**

| | MAP | NPP | CDF | W | $W_{regolith}^{Ca}$ | $W_{regolith}^{K}$ | $W_{regolith}^{Na}$ | $W_{regolith}^{P}$ | $W_{regolith}^{Si}$ | $\tau^{Ca}$ | $\tau^{K}$ | $\tau^{Na}$ | $\tau^{P}$ | $\tau^{Si}$ |
|---|---|---|---|---|---|---|---|---|---|---|---|---|---|---|
| D | **0.68\*** | **0.72\*** | 0.02 | **0.54\*** | 0.31 | 0.53 | **0.94\*\*** | **0.76\*** | **0.81\*** | 0.33 | 0.05 | 0.57 | 0.20 | 0.16 |
| MAP | | **0.99\*\*** | -0.24 | 0.09 | -0.11 | 0.41 | **0.64\*** | 0.41 | 0.41 | 0.29 | 0.16 | **0.72\*** | 0.21 | -0.11 |
| NPP | | | -0.07 | 0.24 | 0.05 | 0.46 | **0.72\*** | 0.54 | 0.52 | 0.45 | 0.23 | **0.82\*** | 0.35 | 0.06 |

CDF = chemical depletion fraction; W = soil weathering rate; $W_{regolith}^{X}$ = elemental weathering flux; $\tau^{X}$ = elemental mass transfer coefficient

Note that because CDF and $\tau^{X}$ are per definition different in sign (i.e. a CDF of +1 denote entire depletion, whereas a $\tau^{X}$ of -1 denote entire depletion), $\tau^{X}$ was multiplied by -1 for presentation purposes. Bold numbers denote significant correlation.

**Table A6** ANOVAs evaluating variations in denudation rate (D), the chemical depletion fraction (CDF), soil weathering rate (W), and the elemental weathering rates for Ca, K, Na, P, and Si ($W^{Ca}_{regolith}$, $W^{K}_{regolith}$, $W^{Na}_{regolith}$, $W^{P}_{regolith}$, $W^{Si}_{regolith}$) among sites (single regolith profiles treated as independent from each other, n = 8). The Tukey HSD test for site comparison is shown below. Sig = 1 indicates significant differences between the sites. The comparison between Santa Gracia and Nahuelbuta is highlighted in bold because of their importance in the discussion.

| | | D | | | CDF | | | W | | | $W^{Ca}_{regolith}$ | | | $W^{K}_{regolith}$ | | | $W^{Na}_{regolith}$ | | | $W^{P}_{regolith}$ | | | $W^{Si}_{regolith}$ | | |
|---|---|---|---|---|---|---|---|---|---|---|---|---|---|---|---|---|---|---|---|---|---|---|---|---|---|---|
| | | *F ratio* | *p > F* | *Sig* | *F ratio* | *p > F* | *Sig* | *F ratio* | *p > F* | *Sig* | *F ratio* | *p > F* | *Sig* | *F ratio* | *p > F* | *Sig* | *F ratio* | *p > F* | *Sig* | *F ratio* | *p > F* | *Sig* | *F ratio* | *p > F* | *Sig* |
| **Overall ANOVA** | | 6.91 | 0.05 | | 8.07 | 0.04 | | 4.58 | 0.09 | | 7.86 | 0.04 | | 2.22 | 0.23 | | 3.63 | 0.10 | | 3.52 | 0.00 | | 7.96 | 0.01 | |
| **Site 1** | **Site 2** | *F value* | *p value* | *Sig* | *F value* | *p value* | *Sig* | *F value* | *p value* | *Sig* | *F value* | *p value* | *Sig* | *F value* | *p value* | *Sig* | *F value* | *p value* | *Sig* | *F value* | *p value* | *Sig* | *F value* | *p value* | *Sig* |
| Santa Gracia | Pan de Azúcar | | 0.86 | 0 | | 0.07 | 1 | | 0.80 | 0 | | 0.04 | 1 | | 1.00 | 0 | | 0.80 | 0 | | 0.34 | 0 | | 0.80 | 0 |
| La Campana | Pan de Azúcar | | 0.04 | 1 | | 0.05 | 1 | | 0.09 | 1 | | 0.09 | 1 | | 0.27 | 0 | | 0.10 | 1 | | 0.10 | 1 | | 0.01 | 1 |
| La Campana | Santa Gracia | | 0.08 | 1 | | 0.99 | 0 | | 0.20 | 0 | | 0.70 | 0 | | 0.30 | 0 | | 0.25 | 0 | | 0.62 | 0 | | 0.30 | 0 |
| Nahuelbuta | Pan de Azúcar | | 0.36 | 0 | | 0.83 | 0 | | 0.96 | 0 | | 0.50 | 0 | | 1.00 | 0 | | 0.61 | 0 | | 0.42 | 0 | | 0.95 | 0 |
| **Nahuelbuta** | **Santa Gracia** | | **0.70** | **0** | | **0.14** | **0** | | **0.97** | **0** | | **0.04** | **1** | | **1.00** | **0** | | **0.98** | **0** | | **1.00** | **0** | | **1.00** | **0** |
| Nahuelbuta | La Campana | | 0.22 | 0 | | 0.11 | 0 | | 0.13 | 0 | | 0.38 | 0 | | 0.31 | 0 | | 0.35 | 0 | | 0.51 | 0 | | 0.29 | 0 |

F ratio = ratio of two mean square values.
P value ≤ 0.1 = Populations have significant different mean values
Sig = significance; 0: not significant, 1: significant