# Peer review of "Do degree and rate of silicate weathering depend on plant productivity?"

_Biogeosciences, 2020_

## Referee Comment (RC1) · Anonymous Referee #1 · 4 Apr 2020

This paper is ambitious in scope, and attempts to tease apart the effect of plants on weathering across a well-studied climate/productivity gradient in the Andes. I applaud the attempt, but am not convinced by the conclusion that plants have little effect on weathering. In the end, despite the ambition, there are so many confounding variables between these four sites that I think site to site variation makes larger conclusions impossible. Four sites, with so much variation both within and among them, are not likely to be sufficient to see the signal through the noise. For example: 1) An alternative to the idea that plants retard weathering is that some of these soils are in the same "process

domain" as defined by Vitousek and Chadwick (2013). Given relatively short residence times, and relatively dry conditions (even at the wetter site), this is not surprising. For example, sites at these rainfalls do not differ after 10000 years of soil development on a Hawaiian lava flow, and barely differ after 150,000 years. That does not necessarily mean that plants have no effect on weathering. 2) Between the two wetter sites, plant cover as a percent doesn't differ, but NPP differs by a factor of 2 s, catchment denudation rates by a factor of 10, and soil denudation by a factor of 3. Yet the CDF is much higher at the drier of these two sites. One way to interpret this is that there is no effect of plants on weathering. Another is that there are so many differences between these sites that it would be hard to see the effects of plants, especially as these are both relatively dry sites, and weathering under dry conditions takes a long time.

I also provide some line by line comments below:

L45 – Porder et al 2007 evaluated mass loss and dust inputs on a climate x time matrix. I think it's relevant to cite here.

L59 – I'm not sure I follow the logic here. Nutrient recycling makes plants less dependent on inputs of nutrients via weathering, but it doesn't necessarily mean that plants don't drive weathering anyway. For example, as organic matter accumulates in soil over time this can help drive down pH. Plants may increasingly rely on the organic matter for nutrients, but the lower pH may drive increased weathering none the less. A classic example of biology driving weathering quasi independently of nutrient uptake is the role of nitrification (which provides nutrients to plants) in driving soil acidification via nitrate leaching.

L92 – river sand or soil profile cosmogenic 10Be?

L105 – It is worth thinking about these results in the context of the "pedogenic threshold" model of Vitousek and Chadwick. It strikes me that all of these sites may be in a pretty similar "process domain" and that given the mean residence time of the soils one might not expect big differences in the amount of observed weathering if the soils

are relatively well buffered.

L137 – It seems odd to state that erosion rates are similar between these sites when they vary by more than an order of magnitude. That seems a potential cofounding factor. It doesn't vary directly with precipitation, which is nice, but it will set up differences in soil residence time that could confound the results here (since climate by time interactions are common, see Porder et al 2007 as an example).

L145 – I agree these don't vary as much as erosion rates around the world, but I'm not sure that's the bar by which to judge whether these are "similar" sites.

L155 – "south" not "soul"

L165 – The "gently sloping hills" at Nahuelbuta would lead to longer soil residence times and thus more weathered soils. Again, I am skeptical of the "control" over erosion rates and residence times in this set up. Especially because the depth of the weathering zone is not known.

L180 – Not sampling roots will lead to an underestimation of both the plant pool and of NPP. In addition, some grasses and desert woody plants have an extremely high fraction of biomass below ground, so not sampling belowground will lead to bias (not just underestimates). Since there is very little detail on vegetation sampling, it is hard to evaluate how much a problem this is, but it could be substantial. In addition, the stoichiometry of NPP is not just NPP x chemistry, since woody plants and perennials in general may have a bulk chemistry that is very different form the chemistry of leaves that are forming and falling more frequently. Much more description of the vegetation and the assumptions about pools and fluxes is needed in order to evaluate this part of the paper.

L199 – Drying vegetation at 120C will lead to a substantial loss of carbon and nitrogen. Loss of P and cations will be smaller. Were plant standards dried at this temperature to ensure that this high temperature did not influence the results? It's hard to tell when

the NIST standards were included in the process.

L235 – Depending on the age of the parent material and the mineralogy, using Sr isotopes from granitic rocks as a tracer through plants can be problematic. This occurs particularly if there are high amounts of Kspar (which likely varies from site to site here and will be particularly sensitive to the occurrence of "metamorphic basement"(L170) at the Nahuelbuta site. See early work by Tom Bullen for a more complete description of the problem.

L275 –I find the lack of replication within site really troubling, especially given how sensitive CDF can be to variations in Zr (as the authors note). I appreciate that the authors used Monte Carlo to get at uncertainty, but there seem to be so few samples that I worry this will underestimate the uncertainty nonetheless. Another concern is that (in Appendix A) it seems many samples were excluded from the parent material if they had different chemistry (e.g. pegmatite, mafics). However those samples must contribute to the soil. Including them would make for much bigger error bars on CDF (I think) and thus make consideration about differences (or lack of differences) between sites all that much harder to justify. As for the potassium issue discussed here, couldn't the concentration of K be increased by a combination of plant uplift (e.g. Jobbagy and Jackson, 2004) and soil collapse (which is why you correct by Zr to get tau)? Overall, these uncertainties are very understandable, given heterogenous bedrock etc. But that speaks to the need for way more sampling in order to constrain that heterogeneity.

L275 – The idea of "kinetically limited weathering" seems more an interpretation than a result. Thus it seems more appropriate for the discussion.

L284 – If weathering is deep below the rooting zone weathering from rock does not necessarily mean availability from plants.

L290 – Equation three is a good example of why I think there needs to be a much more rigorous treatment of uncertainty. D, X parent and tau all have uncertainties associated with them, but that does not seem to be considered when thinking about the differences

between sites. The data are presented without any estimate of uncertainty, and thus it is impossible to tell whether there are any statistically significant differences between sites.

L320 – It is true in all ecosystems that uptake of nutrients is fed mostly by recycling and very little by the weathering flux. That is true even if 100% of the nutrients were originally supplied by weathering.

L325 – First, how is a range of 0.723-0.737 "distinct" from 0.726 which falls in that range? Second, given incongruent weathering, why would one expect the bulk bedrock value to match the regolith value?

L338 – Here, incongruent weathering is postulated. What not anywhere else?

L349 – Perhaps due to very few samples, and soils integrating lots of different minerals plus atmospheric inputs?

L364 – Al is often toxic to plants so I'm not sure I would call it "plant beneficial".

L365 – What does it mean to be "mostly N limited"? And why do you consider other elements to be "co-limiting"? These seem two key points for the following text, and should be explained more clearly so the reader can follow the argument.

L385 – I don't understand why you say the system is N limited (by which I presume you mean NPP is N limited), and then compare other elements to P?

L389 – I'm not sure I agree with this interpretation, since the available nutrients are coming out of recycled organic material.

L395 – You might have a look at Ben Turner's recent (2018) Nature paper, where they show relatively constant production across a very strong soil P gradient. Production is maintained by species turnover. Not all plants need the same amount of P (or other nutrients) to maintain the same NPP.

L475 – I'm surprised by this interpretation. There are probably more (in amount) atmospheric inputs at the wetter sites, but the relative balance between rock and atmospheric fluxes is a different thing (could be 50/50 at all sites, but still have much higher fluxes at one site than another). This point comes back to the uncertainty in the weathering fluxes, which themselves depend on three highly variable numbers: D, [Bedrock], and tau.

L525 – If you do not know the depth of the weathering front how you can tell the total amount of weathering, or assert that the total does not differ between sites?

L572 – I completely agree that NPP is maintained by recycling across your sites and indeed across all ecosystems. That does not mean that over long timescales the weathering flux is unimportant.

L574 – If the "geologic pathway" stays constant, one possible reason for that is that the soil residence time is very short for all these sites. The tau and CDF values you present are all pretty low relative to highly weathered soils. This doesn't mean that plants are accelerating weathering, it simply means that the crank is turning over more quickly. This comes back to the total denudation rates at the specific sites where the soil pits were dug.

L578 – I do no think it is appropriate to speculated on what nutrient might be in line to be "next" for limitation. This is not really how ecosystems work, and the high level of species turnover among these sites make stoichiometric interpretations such as these even more speculative.

L580 – I really don't see this conclusion as supported by the data.

L653 – Clarify if this is increasing towards the top or bottom. Also, tau values are negative, so increasing tau usually means less weathered. Some clarification in the text would help avoid confusion.

Figure 1 – It would be great to see error bars on these plots.

Figure 5 – It would be helpful to have the atmospheric input Sr value on these graphs

as well. That way we could see what fraction of the Sr flux is coming from rock vs atmosphere.

Figure A2 – I am not aware of a method that uses NH4OAc to extract "bioavailable" P.

Table 3 – Is D the catchment wide rate or the average of the two soil profiles at the site? Seems like the latter but it would be helpful to clarify.

Table 4 – If you don't include the whole weathering zone how can you know how much weathering is occurring?

Table 5 – Why would grasses and trees have the same leaf:stem biomass (5:95)?

---

## Short Comment (SC1) · 9 Apr 2020

*A note upfront from the submitting person: This review was prepared by four master students in geography at the University of Zurich. The review was part of an exercise during a second semester master level seminar on "the biogeochemistry of plant-soil systems in a changing world", which is organized by prof. Dr. Michael Schmidt and myself. We would like to highlight that the depth of scientific knowledge and technical understanding of these reviewers represents that of master students. We enjoyed discussing the manuscript in the seminar, and hope that the comments will be helpful for

the authors.*

1. Summary

Oeser and Blanckenburg (2020) have evaluated the quantitative impact of plants, their nutrient cycle and growth on weathering processes. Therefore, they have explored weathering, nutrient uptake, and nutrient recycling along the "EarthShape" climate and vegetation gradient in the Chilean Coastal Cordillera. There, the lithology (granitoid), tectonic uplift and erosion rates are generally similar. While the mean annual temperature (MAT) is similar, the mean annual precipitation (MAP) has a strong increase from north to south (precipitation differs from 10 mm yr$-1$ in the north to 1100 mm yr$-1$ in the south). Within their study they are covering arid (Pan de Azúcar National Park, $\sim$26°S), semi-arid (Santa Gracia Nature Reserve, $\sim$30°S), mediterranean (La Campana National Park, $\sim$33°S), and humid-temperate (Nahuelbuta National Park, $\sim$38°S) areas. Moreover, different vegetation types, controlled by climate, are taken into account. To identify the sources of minerals nutrients to plants, they have quantified the bio-available fraction of nutritive elements in regolith and have measured 87Sr/86Sr isotope ratios in bulk rock, regolith, and plant samples. Using these measurements, they determined inventories, gains and losses of nutritive elements in and out of these ecosystems, and quantified mineral nutrient recycling.

The following research questions were addressed: Does weathering increase from north to south along the EarthShape precipitation gradient? Can the increase of NPP from north to south be explained with the additional nutrient supply by weathering? They found that the weathering rates do not increase from north to south along the climate gradient even though the four EarthShape study sites feature a steep north-south gradient in MAP and NPP. Instead, the simultaneous increase in biomass growth rate is accommodated by faster nutrient recycling. Moreover, the presence of plants compensates a potential weathering increase along the gradient downward by regulating the hydrological cycle. Furthermore, the vegetation induces the formation of secondary minerals, which in turn promote a microbial community adapted to nutrient recycling

through weathering. Oeser and Blanckenburg conclude that higher NPP may not be an important driver in the global silicate-weathering cycle due to this nutrient buffering by recycling.

2. General Comments

First of all, it is a very interesting and relevant study with the goal to investigate processes that are not yet fully understood. By investigating four climatologically different locations and corresponding different vegetation, the authors have taken a good approach to their research question, where Chile seems to be an ideal research site. However, in the paper a lot of different aspects and factors were included. We were wondering if some topics go beyond the scope. Therefore, you might consider to narrow down some aspects and factors. The discussion part is really interesting but without a lot of notes and rereading the text, it is really hard to understand and follow the main messages.

Our general opinion is that some parts of the paper are not structured clearly, especially in the beginning, which makes it hard to keep track. By improving that, a lot of our following criticism will be redundant.

We do not comment on the applied measurements and analyses methods, since we do not have the appropriate expertise to evaluate these. However, what caught our eye was that no statistical method was mentioned, which could be included additionally.

Having said that, the structure improved towards the end. We appreciated that in the results and discussion subtitles are used which makes it easier to navigate and keep up with the text. We found that the usage of literature is very good on which the argumentation is based on. Furthermore, we really enjoyed the graphs, they are nicely visualized. We thought that the title may be a bit long but seems appropriate to introduce the study.

Language and definitions:

We often find that the authors have written in a complicated manner where the sentences are too long and interlaced. It is very hard to follow the statement and thoughts. - We would suggest coming up with shorter and more precise sentences

- Also you might try to include more topic sentences

Furthermore, we found some concepts, short forms etc. without sufficient references or definitions. This leads to some difficulties during reading, and we propose to be more consistent when introducing short forms and concepts. For example: - p. 3, line 86: for the reader it is not that clear what EarthShape is, maybe introduce it shortly with a few sentences as a long term research project; especially since concepts of that project are important for the presented study

- Wxregolith etc. is not comprehensible = weathering from regolith and bedrock. Would it not be better to call it Wxsupply?

- p. 11, line 316: define new introduced short form U as uptake more clearly similar to the previous paragraphs

Typos:

We found some typos which are listed below, therefore we suggest some revision grammar- and spelling-wise: - p.2, line 50: nutrients budgets

- p.2, line 53: an increase in nutrient supply through weathering with erosion rates → do you mean: with increasing erosion rates?

- p. 4, line 100: along the along the

- p. 5, line 155: soul-facing

- p. 9, line 263: we focus on

- p. 40, Fig 4/A4 emphazize

- Some numbers are different in the text than in then tables, is this intended?

- p.41, table 5: Value of mean Mg in La Campana is in the table 12'300 and in the text on page 10 line 313 it is only 12'000

- p.41, table 5: mean P in Nahuelbuta is 1300 in the table and 1400 on page 11, line 316

3. Comments on manuscript

Introduction

- In our opinion, the introduction seems too long and weakly structured. We guess not all the information is directly relevant for the specific topic of the article.

- p. 4, line 100-103: The research questions could be formulated more clearly. Maybe use 2-3 sentences to make it clearer. As we understand the question (1): "Is weathering increasing from north to south along the EarthShape precipitation gradient? We want to investigate the differences, although (or precisely because...?), there are similarities concerning mineral supply, dissolution kinetics of solids due to erosion rate and lithology."

- p. 4, line 103-104: Why are the research questions already answered in the introduction? We think this belongs to different text bodies with some links to these questions here.

- p. 4, line 104-106: In our opinion, this should be an outcome of the results, discussion and to conclude at the end. If you were able to identify something else than planned from the beginning, why not add or adjust the research question?

Study area and previous results

p.4, line 108: "Previous results" in the title is not clear for us. Is this meant as results from previous studies? If yes, we would recommend to put citations or the few relevant points for this continuative study. This could be the main part of the introduction.

Sampling

- We can't find any explanations why there were different amounts of samples (bedrock and regolith) taken depending on the locations. e.g. there are less samples for Nahuelbuta

- Descriptions of the sites, sampling and why at some sites specific samples could not be taken could be written in the method part and not later in the text (e.g. discussion) e.g. p.18, line 544: Santa Gracia - the absence of a litter layer

Results and discussion

- p.10, line 304: "Bio-availability of most elements in soil, bar a few exceptions, increase from Pan de Azúcar to La Campana and is lowest in Nahuelbuta." - We are not sure if you can generalize this like that, there are a lot of exceptions: Ca/ Na is rather decreasing, Mg is lower in La Campana, Si is the same for Pan Azucar and Santa Gracia and Sr the same for Santa Garcia and La Campana. The only nutrients which increase are Al, Mn, P and kind of Fe, but there the value for Pan de Azucar is missing

- p.10, line 310: "Average elemental concentrations in bulk plants decrease from Pan de Azúcar towards Nahuelbuta." → We think there are a lot of exceptions too: There are higher values in Santa Gracia (17'800) than in Pan Azucar (15'200) for Ca, higher values in La Campana for Fe,Sr, K and higher values in Nahuelbuta for Mn and Mg

- p.11, line 315: "The nutrient-uptake fluxes of the two most important rock-derived mineral nutrients to plants, P and K, increase steadily from north to south. . ." - we wondered if "steadily" is the right word since there is an exception for Nahuelbuta (1400) which is lower than for La Campana (2900) for K

- p.13, line 377: "As an evaluation of the hypothesis that the nutrient reservoir sets plant stoichiometry ..." → We would recommend that you formulate your hypotheses already in the introduction and not only in the discussion part (like on p.1, line 30)

- p.16, line 480: Why do you introduce the turnover time differences in the discussion part and not in the results part? You might want to consider introducing this term as

well because you use the term a lot in the discussion part.

- p.17, line 513: "We speculate that the effect of vegetation might even compensate for a potential increase in weathering that would be caused by the increase in MAP, essentially damping the geogenic pathway." - Would it not be better to include this part in the conclusion?

Conclusion

- We found that you did not clearly answered the research questions from the introduction

- Missing structure: The first two sentences are well understandable but the thoughts or statements are not properly formulated. To separate the sentences might help.

- p. 19, line 573-574: why still "geogenic nutrient pathway" and "organic nutrient cycle" in quotation marks?

- It seems that the functions of the pathways are one main outcome. Better links to the results, discussion and research questions would be desirable.

Figures and Tables

- We would suggest not to use red and green in the same figure for a gradient/ symbols because of color-blind people (e.g figure 1, 6)

- For us, an illustration summarizing the results would help the understanding of the paper (maybe like figure 1)

- Figure 1: The soil profiles are very nicely designed. We would suggest to rather cut out the "EarthShape study area" labeling and scale up the soil profiles for better visibility

- You refer a lot to tables in other papers which is very confusing (e.g p. 13, line 378) - You might put the most important ones in the appendix.

- For a better visual understanding, it would be nice if you take the same symbols for the same nutrients e.g. Figure 6 orange triangle for Mn, Figure A1 orange circles for Mn

---

## Author Comment (AC1) · 5 May 2020

**REPLY** We thank the reviewer for taking the time to provide us with his detailed critique of our manuscript. The reviewer has raised important points that lead us to think even harder on how to tackle the formidable challenge of unraveling the links between geochemical and biological processes in the Critical Zone, with the confounding factors that no doubt make the resolution of trends caused by an individual process so difficult. This objective lies at the heart of the German-Chilean "EarthShape" project (Earth Surface shaping by biota), and the aim of this project is exploring these interactions over

the most extreme climate and biological gradient on Earth. If we are not able to resolve these interactions, there we might never be able to do so – anywhere. Towards this aim we supply here an extensive dataset of elemental and isotopic weathering zone, soil, and plant compositions at four Critical Zones along this gradient. And we employ several (in part novel) metrics to quantify the geological and biological fluxes. Our aim is to use these quantitative metrics for hypothesis testing: namely to resolve the interactions between ecosystem productivity and silicate weathering. We reply point by point to the reviewer's comments.

**COMMENT** This paper is ambitious in scope and attempts to tease apart the effect of plants on weathering across a well-studied climate/productivity gradient in the Andes. I applaud the attempt. . .
**REPLY** We are grateful for this acknowledgement. Indeed, we have introduced and suggested a whole set of in part novel geochemical and ecological methods and metrics that we deem necessary to tackle this problem.

**COMMENT** . . .but am not convinced by the conclusion that plants have little effect on weathering. In the end, despite the ambition, there are so many confounding variables between these four sites that I think site to site variation makes larger conclusions impossible. Four sites, with so much variation both within and among them, are not likely to be sufficient to see the signal through the noise.
**REPLY:** We appreciate that the reviewer raises the point of the confounding variables. We are fully aware of this issue, in particular with respect to the role of denudation rate. For this very reason for addressing the core question, namely whether plants accelerate weathering, we have limited our comparison to the two sites where all things (except precipitation and biomass growth rate) are similar: Semi-arid Santa Gracia and humid-temperate Nahuelbuta. It is the comparison between these two sites that leads us to conclude that plants do not accelerate weathering. In a revised version we can emphasize that we pursue this strategy to minimize the effect of confounding variables.

**COMMENT** For example: 1) An alternative to the idea that plants retard weathering is

that some of these soils are in the same "process domain" as defined by Vitousek and Chadwick (2013).

**REPLY:** The "Process Domain" and "Pedogenic Threshold" are indeed interesting concepts that were developed for the chronosequence in Hawaii – sites of hugely differing age but featuring stability of their surfaces. In contrast, here we are dealing with sites that have no discrete age as they are permanently eroding. It is not obvious whether and how the threshold concept applies to these entirely different boundary conditions, and even if so, whether the application of these concepts adds to the quantification of linked geochemical and biogenic cycling, which is the objective of this paper. We get back to this point below.

**COMMENT:** Given relatively short residence times, . . .

**REPLY:** We do not understand what the reviewer means with "short". In contrast to My old soils in Hawaii our soil tunrover times can be seen as short. Assuming the cosmic ray attenuation depth of 600 mm and the measured soil denudation rates of 4 to 27 mm/ky we get cosmogenic nuclide integration times (that can be interpreted as soil residence times) of 20 – 140 ky. Compared to many soil denudation rates from eroding landscapes most of our profiles are in fact at the "long" end of residence times.

**COMMENT** . . . and relatively dry conditions (even at the wetter site), this is not surprising.

**REPLY:** Again, we do not understand why the wetter site is considered to be "dry". With 1100 mm/yr Nahuelbuta is way above terrestrial mean global annual precipitation of 700 mm/yr. It is a "humid" site.

**COMMENT** For example, sites at these rainfalls do not differ after 10000 years of soil development on a Hawaiian lava flow, and barely differ after 150,000 years.

**REPLY:** The detailed soil and weathering zone characterization of the EarthShape study sites show that these sites indeed differ in many, if not most properties (see below), and possibly transect a range of thresholds. Yet even though the idea sounds intriguing, we do not know how a strategy to apply the Hawaii concept is to the granitoid

rocks of the Chilean coastal range would look like, and how this would aid to answer the biogenic weathering question.

**COMMENT** That does not necessarily mean that plants have no effect on weathering. **REPLY:** As stated above: we never say this.

**COMMENT:** Between the two wetter sites, plant cover as a percent doesn't differ, but NPP differs by a factor of 2 s, catchment denudation rates by a factor of 10, and soil denudation by a factor of 3.
**REPLY:** The soil denudation rates are the relevant ones because this is where we study the ecosystems and weathering (in particular in La Campana catchment-wide rates that the reviewer cites exceed soil rates by landsliding due to the steep slopes (van Dongen et al., 2019)). Mean soil production rates only differ by a factor of 2 between the two wetter sites.

**COMMENT** Yet the CDF is much higher at the drier of these two sites. One way to interpret this is that there is no effect of plants on weathering. Another is that there are so many differences between these sites that it would be hard to see the effects of plants, especially as these are both relatively dry sites, and weathering under dry conditions takes a long time.
**REPLY:** While the reviewer is right to say that when comparing these two sites it is hard to see the effect of plants in light of these differences, we disagree that these sites are relatively dry. There is a large body of literature stating that for most solutes weathering rate scales with fluid flow, and thus it is difficult to understand why the CDF should be higher at the dryer site. However, because of the confounding effect of the high denudation rates at the second wettest site (La Campana) we based our interpretation on the comparison between to the two sites where all things (except precipitation and biomass growth rate) are similar: Semi-arid Santa Gracia and humid-temperate Nahuelbuta.

**L45 –** Porder et al 2007 evaluated mass loss and dust inputs on a climate x time matrix.

I think it's relevant to cite here.

**REPLY:** We have made extensive use of the papers by Porder, Chadwick, and colleagues that have shaped our thinking. We believe that the suggested reference does not add anything substantially different in addition to the three Porder et al papers we cite already.

**L59 –** I'm not sure I follow the logic here. Nutrient recycling makes plants less dependent on inputs of nutrients via weathering, but it doesn't necessarily mean that plants don't drive weathering anyway. For example, as organic matter accumulates in soil over time this can help drive down pH. Plants may increasingly rely on the organic matter for nutrients, but the lower pH may drive increased weathering none the less. A classic example of biology driving weathering quasi independently of nutrient uptake is the role of nitrification (which provides nutrients to plants) in driving soil acidification via nitrate leaching.

**REPLY:** We do never state that plants do not drive weathering. In fact, in the manuscript's first paragraph and lines 503-510 we explain in detail the mechanisms through which plants do drive weathering. In the line in question we state that the dependence of plant growth on weathering would be non-linear as some fraction of the nutrients needed to fulfill the plants' physiological needs will stem from recycling rather than "fresh" nutrients derived from weathering of primary minerals. This concept is essential to the entire manuscript, and when revising we will make sure that this point is not missed.

**L92 –** river sand or soil profile cosmogenic 10Be?

**REPLY:** Schaller et al. (2018) determined weathering rates using soil profile cosmogenic $^{10}$Be. We will clarify this in a future version of the text accordingly.

**L105 –** It is worth thinking about these results in the context of the "pedogenic threshold" model of Vitousek and Chadwick. It strikes me that all of these sites may be in a pretty similar "process domain" and that given the mean residence time of the soils one might not expect big differences in the amount of observed weathering if the soils

are relatively well buffered.

**REPLY:** As stated above we are concerned that the "pedogenic threshold" model may not be transferable in a straightforward manner from the Hawaiian chronosequences into the eroding Chilean Coastal Range. However, in a detailed pedogenic description we see that many (and more) of the properties used by Vitousek and Chadwick to define their pedogenic thresholds do indeed vary in Chile, and this work was cited (Bernhard et al. (2018). Moreover, it is found that, unlike in Hawaii, these relationships in the soils of the EarthShape study sites vary in a non-linear relationship. The most prominent thresholds were found between the arid Pan de Azúcar and the semiarid Santa Gracia and was attributed to MAP exceeding potential evapotranspiration (see also Slessarev et al., 2016). However, a threshold for base saturation was found to exist between the mediterranean site (La Campana) and the humid-temperate site (Nahuelbuta). Thus, different thresholds exist for different soil properties which do not feature at identical positions along the climate gradient. It is very obvious that our sites are not within the same process domain.

**L137 –** It seems odd to state that erosion rates are similar between these sites when they vary by more than an order of magnitude. That seems a potential cofounding factor. It doesn't vary directly with precipitation, which is nice, but it will set up differences in soil residence time that could confound the results here (since climate by time interactions are common, see Porder et al 2007 as an example).

**REPLY:** Catchment-wide denudation rates indeed vary by almost an order of magnitude and we do not conceal that information. On the soil pit scale (which count for this study), mean denudation rates vary from 10 to 40 t $\mathrm{km}^{-2}$ $\mathrm{yr}^{-1}$, and individual rates from 8 to 70 t $\mathrm{km}^{-2}$ $\mathrm{yr}^{-1}$. In the detailed parts of the discussion, we focus on the comparison between Santa Gracia and Nahuelbuta. In these two sites, soil residence times are similar (24 $\pm$ 1 and 28 $\pm$ 2 kyr in Santa Gracia vs. 22 $\pm$ 1 kyr in Nahuelbuta; Schaller et al., 2018). Yet our general conclusions on the impacts of plants on weathering also agree with the findings at the arid (Pan de Azúcar) and mediterranean study site (La Campana) even though these are subject to either atmospheric deposition of e.g. Ca

and Sr or increased denudation rates because of very steep hill slopes, respectively.

**L165 –** The "gently sloping hills" at Nahuelbuta would lead to longer soil residence times and thus more weathered soils. Again, I am skeptical of the "control" over erosion rates and residence times in this set up. Especially because the depth of the weathering zone is not known.

**REPLY:** We agree that gently sloping hills would lead to longer soil residence times and thus more weathered soils. However, the soil residence times in Nahuelbuta are shorter than in Santa Gracia despite lower slope angles (these are the two sites we base our principle comparison on).

**L180 –** Not sampling roots will lead to an underestimation of both the plant pool and of NPP. In addition, some grasses and desert woody plants have an extremely high fraction of biomass below ground, so not sampling belowground will lead to bias (not just underestimates). Since there is very little detail on vegetation sampling, it is hard to evaluate how much a problem this is, but it could be substantial. In addition, the stoichiometry of NPP is not just NPP x chemistry, since woody plants and perennials in general may have a bulk chemistry that is very different form the chemistry of leaves that are forming and falling more frequently. Much more description of the vegetation and the assumptions about pools and fluxes is needed in order to evaluate this part of the paper.

**REPLY:** Indeed, we acknowledge that the description of vegetation sampling should have been more detailed, and we will amend this deficit. We are also aware of the fact that the estimation of plants' representative chemical composition is an estimate, and that this estimate includes assumptions. This is due to the fact that a) preciously little information exists on belowground biomass, on its mass per se but even more so on its stoichiometry; b) it is very hard and mostly close to impossible to sample whole plants including roots, in particular when it comes to large trees. Nevertheless, to begin somewhere we have attempted to do whole-plant budgets, and this is how we have done it: Vegetation samples have been taken in the austral summer to autumn

2016 and was restricted to mature higher plants in the study sites (e.g. grasses have been excluded from sampling). From each sampled plant (n=20), multiple samples of leaves, twigs and stem have been taken and were pooled together and homogenized prior analysis. Our estimation of the plants' chemical composition invokes several assumptions: (1) Roots' biomass growth attribute only little to total plant growth, namely 9% in angiosperms and 17% in gymnosperms (Niklas and Enquist, 2002). We thus treat roots and stem/twig as one plant compartment and allocate 68% and 52% in angiosperms and gymnosperms, respectively, of relative growth to these compartments. (2) Differences do only occur between angiosperms and gymnosperms. (3) The pattern of relative growth and standing biomass allocation holds true across a minimum of eight orders of magnitude of species size (Niklas and Enquist, 2002). Thus, we assume that the growth rates of plant organs do not vary considerably between different plant species. We are aware that these assumptions result in only a rough estimation on plants' chemical composition. NPP was derived from a dynamic vegetation model simulating the vegetation cover and composition during the Holocene (Werner et al., 2018). It is thus independent on our sampling strategy.

**L199 –** Drying vegetation at 120C will lead to a substantial loss of carbon and nitrogen. Loss of P and cations will be smaller. Were plant standards dried at this temperature to ensure that this high temperature did not influence the results? It's hard to tell when the NIST standards were included in the process.
**REPLY:** We did not analyze C and N. We did dry plant samples at this high temperature to ensure that any $H_2O$ will disappear. Unfortunately, the SRM 1515 has not been dried at $120°$.

**L235 –** Depending on the age of the parent material and the mineralogy, using Sr isotopes from granitic rocks as a tracer through plants can be problematic. This occurs particularly if there are high amounts of Kspar (which likely varies from site to site here and will be particularly sensitive to the occurrence of "metamorphic basement"(L170) at the Nahuelbuta site. See early work by Tom Bullen for a more complete description

of the problem.

**REPLY:** Bullen et al. (1997) did investigate the behavior of Sr along a chronosequence and found that $^{87}Sr/^{86}Sr$ extracted using ammonium-acetate decreases from a value representing K-feldspar to those of plagioclase and hornblende with increasing soil age, suggesting that the exchangeable pool is dominated by Sr which is leached from K-feldspar after deposition. In contrast, the abundance of K-feldspar in the EarthShape bedrocks is similar (Oeser et al., 2018) and based on the calculation of $\tau$-values (i.e. K, Si, Na), no preferential or early dissolution of K-feldspar is recorded. We would further like to stress differences in the setup between our study and the study by Bullen et al. (1997). Their study is based on the concept of a chronosequence whereas our study follows the climosequence approach. Bullen et al. (1997) derived their conclusions based on soils of varying age whereas in the EarthShape sites, the soil residence times i.e. the average time a mineral grain remains in the mobile layer are broadly similar and range from $11 \pm 1$ to $30 \pm 2$ kyr (Schaller et al., 2018).

**L275 –** I find the lack of replication within site really troubling, especially given how sensitive CDF can be to variations in Zr (as the authors note). I appreciate that the authors used Monte Carlo to get at uncertainty, but there seem to be so few samples that I worry this will underestimate the uncertainty nonetheless. Another concern is that (in Appendix A) it seems many samples were excluded from the parent material if they had different chemistry (e.g. pegmatite, mafics). However those samples must contribute to the soil. Including them would make for much bigger error bars on CDF (I think) and thus make consideration about differences (or lack of differences) between sites all that much harder to justify. As for the potassium issue discussed here, couldn't the concentration of K be increased by a combination of plant uplift (e.g. Jobbagy and Jackson, 2004) and soil collapse (which is why you correct by Zr to get tau)? Overall, these uncertainties are very understandable, given heterogenous bedrock etc. But that speaks to the need for way more sampling in order to constrain that heterogeneity.

**REPLY:** The reviewer is very wrong with the statement that there is a "lack of within-site replication". At each site two regolith profiles situated on opposing slopes have

been studied to account for variations in substrate and/or effects of insolation and microclimate on weathering and nutrient uptake by plants. These two regolith profiles at each site are replicates. In fact, in the site description paper that this study is based on (Oeser et al. 2018, cited) we have measured four profiles at each site. Here, due to the extremely time-consuming extraction and isotope work, we have limited these replicates to two per site. Only for synthesis we present the elemental fluxes in terms of study site averages. Maybe the reviewer read these to infer lack of replications. We will clarify that have in fact made extensive replication. The reviewer rightly scrutinizes the exclusion of certain samples. However, the exclusion of regolith samples only involves samples from one of eight profiles: The S-facing regolith profile in Nahuelbuta. This exclusion does not influence the estimate of $W^X_{regolith}$. The excluded samples are exclusively situated within the saprolite and we parameterize $W^X_{regolith}$ using the most negative tau-value of the lowermost mineral-soil sample. $W^X_{regolith}$ thus integrates over the entire mass loss occurring in regolith. Note that these samples were not excluded in the determination of the inventories in saprolite and soil as their solutes released from weathering indeed contribute to the bio-available fraction. In terms of CDF, the exclusion of these samples from the Nahuelbuta S-facing profile remove samples with highly negative CDF only. Negative CDF cannot be explained with element loss through weathering in regolith with a single parent material. Instead, regolith chemical composition might reflect mixing of multiple parent materials with distinct rates of soil production each. However, we are not able to disentangle these rates. In the previous study (Oeser et al., 2018), four regolith profiles at each site were used to determine study-site representative values for CDF and no regolith samples have been excluded. The results of both studies show consistent results. Oeser et al. (2018) determined the volume loss or gain in the four sites using the volumetric strain $\epsilon$ (Brimhall and Dietrich, 1987). Dilatation (or soil collapse?) was only found in the A and B horizon in Santa Gracia. In the other sites, saprolite and soil was characterized by volume expansion.

**L275 –** The idea of "kinetically limited weathering" seems more an interpretation than

a result. Thus it seems more appropriate for the discussion.

**REPLY:** There are primary minerals left in the soil, thus erosion is sufficiently high such that weatherable minerals still exist in soil and the weathering rate is limited by mineral dissolution kinetics, based on the concepts of numerous publications (e.g. Dixon et al., 2012). In our opinion this is a factual observation, hence it is no discussion item.

**L284 –** If weathering is deep below the rooting zone weathering from rock does not necessarily mean availability from plants.

**REPLY:** Weathering occurs throughout the entire regolith and different weathering fronts do exist. They depend on the minerals' dissolution kinetics. We did refer to "most plant-essential rock-derived mineral nutrients" to emphasize the geologic origin. We will address this concern and rephrase the sentence accordingly.

**L290 –** Equation three is a good example of why I think there needs to be a much more rigorous treatment of uncertainty. D, X parent and tau all have uncertainties associated with them, but that does not seem to be considered when thinking about the differences between sites. The data are presented without any estimate of uncertainty, and thus it is impossible to tell whether there are any statistically significant differences between sites.

**REPLY:** The calculation of $W_{regolith}^{X}$ did involve a rigorous error propagation and the uncertainties on the weathering fluxes were estimated by Monte Carlo simulations. For this calculation we did use 1SD of the respective profiles Denudation rate, the 1SD of the bedrocks' element concentration of interest, and 3% relative uncertainty on the element concentration in regolith samples (see Table 3). We will address this concern and include the standard deviation in the text such that the reader can more easily decide whether a statistical difference exist or not.

**L320 –** It is true in all ecosystems that uptake of nutrients is fed mostly by recycling and very little by the weathering flux. That is true even if 100% of the nutrients were originally supplied by weathering.

**REPLY:** But over geological time scales the losses occurring through erosion and as

solutes need to be balanced by supply from weathering (Uhlig and von Blanckenburg, 2019), otherwise ecosystems would run into depletion.

**L325 –** First, how is a range of 0.723-0.737 "distinct" from 0.726 which falls in that range? Second, given incongruent weathering, why would one expect the bulk bedrock value to match the regolith value?
**REPLY:** Indeed, the wording of this sentence is misleading, and we will rephrase the sentence accordingly. In Pan de Azúcar, the degree of weathering is very low, mainly attributed to physical disintegration of rock. Given the low losses through weathering, we would assume that bedrock and regolith are identical in their $^{87}$Sr/$^{86}$Sr.

**L338 –** Here, incongruent weathering is postulated. What not anywhere else?
**REPLY:** The regolith profiles in La Campana were the only profiles where we were able to correlate changes in $^{87}$Sr/$^{86}$Sr to losses of certain elements by using $\tau^{Sr}$, $\tau^{Ca}$, and $\tau^{K}$. The other regolith profiles did not permit to resolve weathering-related trends in $^{87}$Sr/$^{86}$Sr or incongruent weathering.

**L349 –** Perhaps due to very few samples, and soils integrating lots of different minerals plus atmospheric inputs?
**REPLY:** We do not understand why the reviewer places so much emphasis on the "low" number of samples. In total we did sample 13 different plant species comprising 20 different specimens. Each leaf sample for example integrates over several leaves which have been homogenized prior dissolution. In terms of interpretation, it seems very unlikely that we did sample just by coincidence the plants with the same $^{87}$Sr/$^{86}$Sr ratio than the bio-available fraction they grow on. It is rather the proof that plants take up nutrients from the bio-available fraction.

**L364 –** Al is often toxic to plants so I'm not sure I would call it "plant beneficial".
**REPLY:** Plant-beneficial elements are those which compensate for the toxic effects of other elements or substitute for elements and cover some of their less-specific functions (e.g. maintaining osmotic pressure; Al, Na, Si). However, whether an element is

essential or beneficial to plants is species dependent (Marschner, 1993). According to Liang et al. (2007), the effects of Al-toxicity can be mediated by Si.

**L365 –** What does it mean to be "mostly N limited"? And why do you consider other elements to be "co-limiting"? These seem two key points for the following text, and should be explained more clearly so the reader can follow the argument.
**REPLY:** It is an observation by soil ecologists that the EarthShape sites are first and primarily N-limited (Stock et al., 2019). However, the role of additional nutrients on NPP is increasingly recognized- subsumed under the term "co-limitation". We will add explanatory text.

**L385 –** I don't understand why you say the system is N limited (by which I presume you mean NPP is N limited), and then compare other elements to P?
**REPLY:** For N-limitation, see previous comment. However, this paper is about the mineral nutrients of which P is the most likely element to limit NPP unless continuously supplied by weathering (over the timescale of the "geogenic pathway"). This is why we normalize to P, in a "Redfield Ratio" sense.

**L389 –** I'm not sure I agree with this interpretation, since the available nutrients are coming out of recycled organic material.
**REPLY:** We fully agree with the reviewer's statement. In fact, we make exactly this point in this paragraphs' point (4) and in section 5.3. In the first instance, we find that a first-order stoichiometry is set by the geogenic source (our point 1.). We then indeed find evidence that with increasing recycling efficiency (increases from Pan de Azúcar towards Nahuelbuta), the nutrient pools in the soil bio-available fraction are increasingly dominated by the pool of recycled (our point 4). These two findings lead to key concepts of this paper.

**L395 –** You might have a look at Ben Turner's recent (2018) Nature paper, where they show relatively constant production across a very strong soil P gradient. Production is maintained by species turnover. Not all plants need the same amount of P (or other

nutrients) to maintain the same NPP.

**REPLY:** Turner et al. (2018) did focus on a multitude of different plant species, their P concentration in leaf and in soil. This, however, is well beyond the scope of our study. The reviewer might be right however in as much as possible nutrient limitations on NPP are buffered by the entire ecosystem community by a shift in species community composition (i.e. species turnover).

**L475 –** I'm surprised by this interpretation. There are probably more (in amount) atmospheric inputs at the wetter sites, but the relative balance between rock and atmospheric fluxes is a different thing (could be 50/50 at all sites, but still have much higher fluxes at one site than another). This point comes back to the uncertainty in the weathering fluxes, which themselves depend on three highly variable numbers: D, [Bedrock], and tau.

**REPLY:** We do not understand this comment. We do not present absolute values on atmospheric fluxes. We rather estimate the relative (!) contribution based on radiogenic Sr isotope ratios. In Pan de Azúcar, the weathering release fluxes are very low but a constant supply of moisture (and aerosols) through the Atacama Desert fog (Camanchaca) is prevailing. Thus, our estimate of roughly 90% seems reasonable. Regarding the other sites their relative contribution to ecosystem nutrition is minor and negligible.

**L525 –** If you do not know the depth of the weathering front how you can tell the total amount of weathering, or assert that the total does not differ between sites? **REPLY:** Because the metric $W^X_{regolith}$ integrates over entire loss occurring in regolith, as the lowest $\tau^X$ is (by definition) the depth-integrated weathering loss. Measuring weathering flux with cosmogenic nuclides and chemical depletion does not require knowing the depth of the weathering front.

**L572 –** I completely agree that NPP is maintained by recycling across your sites and indeed across all ecosystems. That does not mean that over long timescales the weathering flux is unimportant.

**REPLY:** This is EXACTLY what we say. The "geogenic pathway" does the job over the

long-term.

**L574 –** If the "geologic pathway" stays constant, one possible reason for that is that the soil residence time is very short for all these sites. The tau and CDF values you present are all pretty low relative to highly weathered soils. This doesn't mean that plants are accelerating weathering, it simply means that the crank is turning over more quickly. This comes back to the total denudation rates at the specific sites where the soil pits were dug.

**REPLY:** We failed to understand what the reviewer wishes to say. However, we re-iterate that soil residence times are not what we consider to be low. Also, we do not find direct evidence that plants accelerate weathering along the gradient, and do not say they do so. However, plants likely do weathering work. Our point is that weathering isn't proportional to NPP.

**L578 –** I do not think it is appropriate to speculated on what nutrient might be in line to be "next" for limitation. This is not really how ecosystems work, and the high level of species turnover among these sites make stoichiometric interpretations such as these even more speculative.

**REPLY:** Agree this may be a simple geochemists' view on how processes in ecosystems work. We can tune this statement down, or eliminate it. But we would be curious to hear how one can explain our evidence for deep K uptake in La Campana and the higher K concentrations in plants relative to P when compared to the source (Figure 6).

**L580 –** I really don't see this conclusion as supported by the data.

**REPLY:** This point is indeed speculative. It results from the discussion (line 547ff on soil $CO_2$ and Si solubility) and explains the lower weathering rate in the wettest and highest NPP site.

**L653 –** Clarify if this is increasing towards the top or bottom. Also, tau values are negative, so increasing tau usually means less weathered. Some clarification in the text would help avoid confusion.

**REPLY:** The $\tau$-values do increase towards the profiles' top, thus indicating elemental gain.

**Figure 1 –** It would be great to see error bars on these plots.
**REPLY:** We decided not to plot the error bars in this figure to maintain a clear layout. However, this can be done.

**Figure 5 –** It would be helpful to have the atmospheric input Sr value on these graphs as well. That way we could see what fraction of the Sr flux is coming from rock vs atmosphere.
**REPLY:** We will add the atmospheric input Sr value on these graphs.

**Figure A2 –** I am not aware of a method that uses NH4OAc to extract "bioavailable" P.
**REPLY:** P-accessibility in the bio-available fraction has been determined by Brucker and Spohn (2019) using a modified Hedley sequential P fractionation method. We will correct the figure's caption accordingly.

**Table 3 –** Is D the catchment wide rate or the average of the two soil profiles at the site? Seems like the latter but it would be helpful to clarify.
**REPLY:** D is the average of the two soil profiles at the site. We will correct the table's caption accordingly.

**Table 4 –** If you don't include the whole weathering zone how can you know how much weathering is occurring?
**REPLY:** In this table we report on the size of inventories, not about the degree of weathering or the weathering rate. We did decide to scale the extent of the saprolite and the regolith inventory to 1.0m for purposes of comparisons. Once the actual extent of the weathering zone is known, one can extrapolate the size of the inventories to that depth.

**Table 5 –** Why would grasses and trees have the same leaf:stem biomass (5:95)?
**REPLY:** Please see our response to line your comment on L180: According to Niklas

and Enquist (2002), the pattern of relative growth and standing biomass allocation holds true across a minimum of eight orders of magnitude of species size.

**References**

Bernhard, N., Moskwa, L.-M., Schmidt, K., Oeser, R. A., Aburto, F., Bader, M. Y., Baumann, K., von Blanckenburg, F., Boy, J., van den Brink, L., Brucker, E., Canessa, R., Dippold, M. A., Ehlers, T. A., Fuentes, J. P., Godoy, R., Köster, M., Kuzyakov, Y., Leinweber, P., Neidhard, H., Matus, F., Mueller, C. W., Oelmann, Y., Oses, R., Osses, P., Paulino, L., Schaller, M., Schmid, M., Spielvogel, S., Spohn, M., Stock, S., Stroncik, N., Tielbörger, K., Übernickel, K., Scholten, T., Seguel, O., Wagner, D., and Kühn, P.: Pedogenic and microbial interrelations to regional climate and local topography: new insights from a climate gradient (arid to humid) along the Coastal Cordillera of Chile, Catena, 170, 10.1016/j.catena.2018.06.018, 2018.

Brimhall, G. H., and Dietrich, W. E.: Constitutive mass balance relations between chemical composition, volume, density, porosity, and strain in metasomatic hydrochemical systems: Results on weathering and pedogenesis, Geochim. Cosmochim. Acta, 51, 567-587, 10.1016/0016-7037(87)90070-6, 1987.

Brucker, E., and Spohn, M.: Formation of soil phosphorus fractions along a climate and vegetation gradient in the Coastal Cordillera of Chile, Catena, 180, 203-211, 10.1016/j.catena.2019.04.022, 2019.

Bullen, T. D., White, A., Blum, A., Harden, J., and Schulz, M.: Chemical weathering of a soil chronosequence on granitoid alluvium: II. Mineralogic and isotopic constraints on the behavior of strontium, Geochim. Cosmochim. Acta, 61, 291-306, 10.1016/s0016-7037(96)00344-4, 1997.

Dixon, J. L., Hartshorn, A. S., Heimsath, A. M., DiBiase, R. A., and Whipple, K. X.: Chemical weathering response to tectonic forcing: A soils perspective from the San Gabriel Mountains, California, Earth. Planet. Sci. Lett., 323-324, 40-49, 10.1016/j.epsl.2012.01.010, 2012.

Liang, Y., Sun, W., Zhu, Y. G., and Christie, P.: Mechanisms of silicon-mediated alleviation of abiotic stresses in higher plants: a review, Environ. Pollut., 147, 422-428, 10.1016/j.envpol.2006.06.008, 2007.

Marschner, H.: Marschner's mineral nutrition of higher plants, 2nd ed., 1993. Niklas, K. J., and Enquist, B. J.: Canonical rules for plant organ biomass partitioning and annual allocation., Am. J. Bot., 89, 812-819, 2002.

Oeser, R. A., Stroncik, N., Moskwa, L.-M., Bernhard, N., Schaller, M., Canessa, R., van den Brink, L., Köster, M., Brucker, E., Stock, S., Fuentes, J. P., Godoy, R., Matus, F. J., Oses Pedraza, R., Osses McIntyre, P., Paulino, L., Seguel, O., Bader, M. Y., Boy, J., Dippold, M. A., Ehlers, T. A., Kühn, P., Kuzyakov, Y., Leinweber, P., Scholten, T., Spielvogel, S., Spohn, M., Übernickel, K., Tielbörger, K., Wagner, D., and von Blanckenburg, F.: Chemistry and Microbiology of the Critical Zone along a steep climate and vegetation gradient in the Chilean Coastal Cordillera., Catena, 170, 183-203, 10.1016/j.catena.2018.06.002, 2018. Schaller, M., Ehlers, T. A., Lang, K. A. H., Schmid, M., and Fuentes-Espoz, J. P.: Addressing the contribution of climate and vegetation cover on hillslope denudation, Chilean Coastal Cordillera (26°–38°S), Earth. Planet. Sci. Lett., 489, 111-122, 10.1016/j.epsl.2018.02.026, 2018.

Slessarev, E. W., Lin, Y., Bingham, N. L., Johnson, J. E., Dai, Y., Schimel, J. P., and Chadwick, O. A.: Water balance creates a threshold in soil pH at the global scale, Nature, 540, 567, 10.1038/nature20139, 2016.

Stock, S. C., Köster, M., Dippold, M. A., Nájera, F., Matus, F., Merino, C., Boy, J., Spielvogel, S., Gorbushina, A., and Kuzyakov, Y.: Environmental drivers and stoichiometric constraints on enzyme activities in soils from rhizosphere to continental scale, Geoderma, 337, 973-982, 10.1016/j.geoderma.2018.10.030, 2019.

Turner, B. L., Brenes-Arguedas, T., and Condit, R.: Pervasive phosphorus limitation of tree species but not communities in tropical forests, Nature, 555, 367-370, 10.1038/nature25789, 2018.

Uhlig, D., and von Blanckenburg, F.: How Slow Rock Weathering Balances Nutrient Loss During Fast Forest Floor Turnover in Montane, Temperate Forest Ecosystems, Frontiers in Earth Science, 7, 10.3389/feart.2019.00159, 2019.

van Dongen, R., Scherler, D., Wittmann, H., and von Blanckenburg, F.: Cosmogenic 10Be in river sediment: where grain size matters and why, Earth Surface Dynamics, 7, 393-410, 10.5194/esurf-7-393-2019, 2019.

Werner, C., Schmid, M., Ehlers, T. A., Fuentes-Espoz, J. P., Steinkamp, J., Forrest, M., Liakka, J., Maldonado, A., and Hickler, T.: Effect of changing vegetation and precipitation on denudation – Part 1: Predicted vegetation composition and cover over the last 21 thousand years along the Coastal Cordillera of Chile, Earth Surface Dynamics, 6, 829-858, 10.5194/esurf-6-829-2018, 2018.

---

## Author Comment (AC2) · 8 May 2020

*A note upfront from the submitting person: This review was prepared by four master students in geography at the University of Zurich. The review was part of an exercise during a second semester master level seminar on "the biogeochemistry of plant-soil systems in a changing world", which is organized by prof. Dr. Michael Schmidt and myself. We would like to highlight that the depth of scientific knowledge and technical understanding of these reviewers represents that of master students. We enjoyed discussing the manuscript in the seminar, and hope that the comments will be helpful

for the authors.*

**REPLY** We thank Marjin Van de Broek and in particular his master students for their detailed comments that indeed will be very helpful to revise our manuscript. We highly appreciate these comments as they point to some structural weaknesses in the manuscript in a constructive manner. In particular you seem to realize the innovative way of tackling the bio-geo interactions in the Critical Zone using novel metrics. You seem to have very good master students at Zürich!
We will REPLY point by point to the comments.
2. General Comments

**COMMENT** First of all, it is a very interesting and relevant study with the goal to investigate processes that are not yet fully understood. By investigating four climatologically different locations and corresponding different vegetation, the authors have taken a good approach to their research question, where Chile seems to be an ideal research site. However, in the paper a lot of different aspects and factors were included. We were wondering if some topics go beyond the scope. Therefore, you might consider to narrow down some aspects and factors.
**REPLY**: We are grateful for the reviewers view on our study design and approach to resolve the interactions between ecosystem productivity and silicate weathering. We acknowledge the reviewer's comment on the multitude of (or too many) aspects of this manuscript. But please bear in mind that we employed a multitude of in part novel metrics to quantify the geological and biological fluxes in ecosystems. These itself and their context in the Critical Zone require a detailed description.
We will address the reviewers concern in a future version of the manuscript by restructuring certain parts of the text. In particular we will rephrase our research hypothesis at the end of the introduction and make better use of topic sentences to guide the reader through the manuscript. We will scrutinise the text for components that can be removed for now without loss of quality.

**COMMENT** The discussion part is really interesting but without a lot of notes and rereading the text, it is really hard to understand and follow the main messages. Our general opinion is that some parts of the paper are not structured clearly, especially in the beginning, which makes it hard to keep track. By improving that, a lot of our following criticism will be redundant.
**REPLY**: We will better structure the manuscript and have other colleagues read it before re-submission.

**COMMENT** We do not comment on the applied measurements and analyses methods, since we do not have the appropriate expertise to evaluate these. However, what caught our eye was that no statistical method was mentioned, which could be included additionally.
**REPLY**: Concerning the note that no statistical evaluation is lacking we note that rigorous within-site replication exists. At each site two regolith profiles situated on opposing slopes have been studied (L176) to account for variations in substrate and/ or effects of insolation and microclimate on weathering and nutrient uptake by plants. These two regolith profiles are natural replicates. The site description and description of the regolith profiles is based on Oeser et al. (2018) where four profiles at each site were measured. We can use this resource to develop an uncertainty estimation on the element fluxes.

**COMMENT** Having said that, the structure improved towards the end. We appreciated that in the results and discussion subtitles are used which makes it easier to navigate and keep up with the text. We found that the usage of literature is very good on which the argumentation is based on. Furthermore, we really enjoyed the graphs, they are nicely visualized. We thought that the title may be a bit long but seems appropriate to introduce the study.
**REPLY**: We are open to suggestions for a shorter title. Would it help if we removed the

first part "Decoupling silicate weathering from primary productivity – "?
Language and definitions:

**COMMENT** We often find that the authors have written in a complicated manner where the sentences are too long and interlaced. It is very hard to follow the statement and thoughts. - We would suggest coming up with shorter and more precise sentences
**REPLY**: We will adjust our manuscript accordingly, and will ask a native speaker for advice.

**COMMENT** Also you might try to include more topic sentences
**REPLY**: Good suggestion. We will make better use of topic sentences to guide the reader through the manuscript.

**COMMENT** Furthermore, we found some concepts, short forms etc. without sufficient references or definitions. This leads to some difficulties during reading, and we propose to be more consistent when introducing short forms and concepts. For example: - p. 3, line 86: for the reader it is not that clear what EarthShape is, maybe introduce it shortly with a few sentences as a long term research project; especially since concepts of that project are important for the presented study
**REPLY**: We regret that the introduction of the EarthShape project – a large interdisciplinary research network - was not sufficient in the manuscript's introduction. We will provide this information.

**COMMENT** Wxregolith etc. is not comprehensible = weathering from regolith and bedrock. Would it not be better to call it Wxsupply?
**REPLY**: Throughout our manuscript we applied the metrics to describe and quantify the element fluxes in the Critical Zone as introduced by Uhlig and von Blanckenburg (2019). In our manuscript we do only briefly introduce them to the reader and refer to Uhlig and von Blanckenburg (2019) for the detailed derivation and description of

these metrices. Therefore, we prefer to retain to the terminology of Uhlig and von Blanckenburg (2019).

**COMMENT** p. 11, line 316: define new introduced short form U as uptake more clearly similar to the previous paragraphs
**REPLY**: In section 3.3 (L253) we introduce the metrics we use to parameterize the geogenic nutrient pathway and the organic nutrient cycle. We briefly refer to Table 2 and the Appendix. In a future version of the manuscript we will make sure that first used abbreviations will refer to Table 2 and the Appendix as well.
Typos:

**COMMENT** We found some typos which are listed below, therefore we suggest some revision grammar- and spelling-wise:
**REPLY**: We thank the reviewers for their careful reading! We will correct the typos and grammar errors accordingly.
3. Comments on manuscript Introduction

**COMMENT** In our opinion, the introduction seems too long and weakly structured. We guess not all the information is directly relevant for the specific topic of the article.
**REPLY**: We will condense the introduction. In particular we will put our emphasis on a clear definition of our research questions and hypothesis (see also below).

**COMMENT** p. 4, line 100-103: The research questions could be formulated more clearly. Maybe use 2-3 sentences to make it clearer. As we understand the question (1): "Is weathering increasing from north to south along the EarthShape precipitation gradient? We want to investigate the differences, although (or precisely because. . .?), there are similarities concerning mineral supply, dissolution kinetics of solids due to erosion rate and lithology."

**REPLY**: We believe that our two principle research questions formulated at the end of the discussion are clear and concise and for clarity we would prefer not to extend them. They are: (1) Does weathering increase from north to south along the along the EarthShape precipitation gradient, because runoff increases while other factors like mineral supply and dissolution kinetics are similar due to the similarities in erosion rate and lithology? (2) Is the increase in net primary productivity (NPP) from north to south accommodated by additional nutrient supply from weathering? (3) Does the nutrient reservoir sets plant stoichiometry?

**COMMENT** p. 4, line 103-104: Why are the research questions already answered in the introduction? We think this belongs to different text bodies with some links to these questions here. - p. 4, line 104-106: In our opinion, this should be an outcome of the results, discussion and to conclude at the end. If you were able to identify something else than planned from the beginning, why not add or adjust the research question?

**REPLY**: We answered the research questions at the end of the introduction as a stylistic element and to trigger the readers' curiosity. We would prefer to retain this, but we do not insist.

Study area and previous results

**COMMENT** p.4, line 108: "Previous results" in the title is not clear for us. Is this meant as results from previous studies? If yes, we would recommend to put citations or the few relevant points for this continuative study. This could be the main part of the introduction.

**REPLY**: We will add: "This study is based on the findings of two previous studies that introduced the field area, its pedogenic and weathering characteristics, and a massive set of soil- and geochemical data Bernhard et al. 2018, Oeser et al. 2018)."

Sampling

**COMMENT** We can't find any explanations why there were different amounts of samples (bedrock and regolith) taken depending on the locations. e.g. there are less samples for Nahuelbuta

**REPLY**: The number of regolith samples depends on the depth of the dug regolith profiles (see Oeser et al., 2018 Fig. 3 to 6 for representative photographs of the regolith profiles at each site). Regarding the bedrock samples, we took as many representative samples as possible. However, access to these samples was highly variable and particularly difficult in the densely vegetated Nahuelbuta.

**COMMENT** Descriptions of the sites, sampling and why at some sites specific samples could not be taken could be written in the method part and not later in the text (e.g. discussion) e.g. p.18, line 544: Santa Gracia - the absence of a litter layer

**REPLY**: The architecture of the regolith profiles, minerology and chemistry has been extensively described by Bernhard et al. (2018); Oeser et al. (2018). However, we summarized these results in the section "Study area and previous results". We mention the absence of a distinct O-horizon in Santa Gracia (L155) and Pan de Azúcar (L153) but indeed do not mention the absence of a litter layer. We will add this information to the manuscript.

Results and discussion

**COMMENT** p.10, line 304: "Bio-availability of most elements in soil, bar a few exceptions, increase from Pan de Azúcar to La Campana and is lowest in Nahuelbuta." - We are not sure if you can generalize this like that, there are a lot of exceptions: Ca/Na is rather decreasing, Mg is lower in La Campana, Si is the same for Pan Azucar and Santa Gracia and Sr the same for Santa Garcia and La Campana. The only nutrients which increase are Al, Mn, P and kind of Fe, but there the value for Pan de Azucar is missing

**REPLY**: The reviewers are right. This generalization does not meet the data. We will rephrase this section starting from line 301 accordingly: Bio-availability of the remaining mineral nutrients in saprolite generally decreases from north to south. Accordingly, the total inventory is highest in Pan de Azúcar (5100 g m$^{-2}$), intermediate in Santa Gracia (2100 g m$^{-2}$) and La Campana (1600 g m$^{-2}$), and lowest in Nahuelbuta (140 g m$^{-2}$; Table 4). Note that I$_{bio-av,\,sap}^{X}$ was calculated over the uppermost 1 m of saprolite, whereas in fact the zone of mineral extraction might extent much deeper. Bio-availability in soil features a similar trend. Total availability is highest in Pan de Azúcar (2100 g m$^{-2}$), on par in Santa Gracia (960 g m$^{-2}$) and La Campana (1000 g m$^{-2}$), and lowest in Nahuelbuta (200 g m$^{-2}$). However, especially P and K deviate from this general trend. . . .

**COMMENT** p.10, line 310: "Average elemental concentrations in bulk plants decrease from Pan de Azúcar towards Nahuelbuta." → We think there are a lot of exceptions too: There are higher values in Santa Gracia (17'800) than in Pan Azucar (15'200) for Ca, higher values in La Campana for Fe,Sr, K and higher values in Nahuelbuta for Mn and Mg
**REPLY**: We will address this point by rephrasing this paragraph accordingly: (L310) Average elemental concentrations in bulk plants generally decrease from Pan de Azúcar towards Nahuelbuta. However, element specific deviations from this pattern do exist (Table 5). The most prominent exceptions are those of P and K. Average P concentration increases from. . .

**COMMENT** p.11, line 315: "The nutrient-uptake fluxes of the two most important rock-derived mineral nutrients to plants, P and K, increase steadily from north to south. . ." - we wondered if "steadily" is the right word since there is an exception for Nahuelbuta (1400) which is lower than for La Campana (2900) for K
**REPLY**: Taken the standard deviation into account our argument holds true. However, we will not use the word "steadily" in a future version of the text.

**COMMENT** p.13, line 377: "As an evaluation of the hypothesis that the nutrient reservoir sets plant stoichiometry ..." → We would recommend that you formulate your hypotheses already in the introduction and not only in the discussion part (like on p.1, line 30)
**REPLY**: In order to better structure this manuscript we will relocate our hypothesis to the end of our introduction.

**COMMENT** p.16, line 480: Why do you introduce the turnover time differences in the discussion part and not in the results part? You might want to consider introducing this term as well because you use the term a lot in the discussion part.
**REPLY**: There is a delicate balance between a result and their interpretation. Only the element composition and radiogenic Sr isotopes in bulk samples, the bio-available fraction, and in plants are results sensu stricto. Element fluxes, weathering rates, etc. are parameterized and an interpretation of this data. The same holds true for the turnover times. Also the turnover times are a derived parameters based on assumtions (i.e. they rely on the parameterized fluxes $U^X_{tot}$ and $W^X_{regolith}$). Thus, we prefer to retain in the discussions section. The same holds true for the nutrient recycling factor RecX.

**COMMENT** p.17, line 513: "We speculate that the effect of vegetation might even compensate for a potential increase in weathering that would be caused by the increase in MAP, essentially damping the geogenic pathway." - Would it not be better to include this part in the conclusion?
**REPLY**: There is a high level of speculation in this part- meant to trigger discussion within the community so as to appreciate the complexity of these feedbacks. The point is truly "discussion" rather than a firm "conclusion". Thus we would prefer to retain this in the discussion section.

Conclusion

**COMMENT** We found that you did not clearly answered the research questions from the introduction

**REPLY**: We think the research questions are answered, But as we will revise the research questions anyway we will also increase the recognition of their answers in the conclusion section.

Figures and Tables (only replies to substantial comments)

**COMMENT** For us, an illustration summarizing the results would help the under-standing of the paper (maybe like figure 1)

**REPLY**: We are limited to a certain amount of publication units and must condense the manuscript, figures and tables as much as possible. If BG offers to submit a graphical abstract, we will create one.

**References**

Bernhard, N., Moskwa, L.-M., Schmidt, K., Oeser, R. A., Aburto, F., Bader, M. Y., Baumann, K., von Blanckenburg, F., Boy, J., van den Brink, L., Brucker, E., Canessa, R., Dippold, M. A., Ehlers, T. A., Fuentes, J. P., Godoy, R., Köster, M., Kuzyakov, Y., Leinweber, P., Neidhard, H., Matus, F., Mueller, C. W., Oelmann, Y., Oses, R., Osses, P., Paulino, L., Schaller, M., Schmid, M., Spielvogel, S., Spohn, M., Stock, S., Stroncik, N., Tielbörger, K., Übernickel, K., Scholten, T., Seguel, O., Wagner, D., and Kühn, P.: Pedogenic and microbial interrelations to regional climate and local topography: new insights from a climate gradient (arid to humid) along the Coastal Cordillera of Chile, Catena, 170, 10.1016/j.catena.2018.06.018, 2018.

Oeser, R. A., Stroncik, N., Moskwa, L.-M., Bernhard, N., Schaller, M., Canessa, R., van den Brink, L., Köster, M., Brucker, E., Stock, S., Fuentes, J. P., Godoy, R., Matus, F. J., Oses Pedraza, R., Osses McIntyre, P., Paulino, L., Seguel, O., Bader, M. Y., Boy, J., Dippold, M. A., Ehlers, T. A., Kühn, P., Kuzyakov, Y., Leinweber, P.,

Scholten, T., Spielvogel, S., Spohn, M., Übernickel, K., Tielbörger, K., Wagner, D., and von Blanckenburg, F.: Chemistry and Microbiology of the Critical Zone along a steep climate and vegetation gradient in the Chilean Coastal Cordillera., Catena, 170, 183-203, 10.1016/j.catena.2018.06.002, 2018.

Uhlig, D., and von Blanckenburg, F.: How Slow Rock Weathering Balances Nutrient Loss During Fast Forest Floor Turnover in Montane, Temperate Forest Ecosystems, Frontiers in Earth Science, 7, 10.3389/feart.2019.00159, 2019.

---

## Referee Comment (RC2) · Anonymous Referee #2 · 11 May 2020

In this manuscript, the authors present total and bioavailable stocks of a number of nutrients and some other elements, in soil, saprolith, and vegetation along a steep climatic gradient in the coastal cordillera of Chile. From their data, they calculated a number of fluxes between the various ecosystem compartments and in and out of the whole system. Their main interpretations are that the weathering rates – which they term abiogenic – did not change substantially along the transect in spite of the marked climatic differences while the nutrient uptake rates increased with standing biomass following the rising rainfall rates. I think that the authors produced high-quality data

from an interesting environmental gradient but I disagree with large parts of their data interpretation.

1. It is hard to believe that weathering rates do not respond to the strong climatic gradient. From the explanation in l. 603-607, I read that the "most negative tau values from the shallowest mineral soil sample of each regolith profile were used" [to calculate weathering rates] and that these values are shown in red in Fig. A1. I do not understand this. The red symbols in Fig. A1 do neither refer to the shallowest horizon nor are they consistently the most negative value. Why did you choose the most negative tau value and not the thickness-weighted average tau value of the whole regolith profile? I anyway think that the tau values cannot be used to estimate weathering, because they lump together a complex mixture of many processes including e.g., leaching losses, deposition input, plant uptake, or soil-internal redistribution. The latter is particularly pronounced in Nahuelbuta, where Podzols occur which show strong depletion (E) horizons and strong accumulation (Bhs) horizons. Instead, I think that an equation such as that proposed by Likens (2013), Biogeochemistry of a forested ecosystem (3rd ed). Springer, New York, USA should be used: $TD_i + W_i = ST_i \pm DeltaB_i \pm DeltaOM_i \pm DeltaX_i \pm DeltaM_i$ where TD is total deposition, i a selected element, W is weathering release from primary minerals, Delta is change, B is storage in biomass, OM is long-term storage in soil organic matter, X is the exchangeable pool and M is the secondary mineral pool. I particularly think that the immobilization in soil organic matter and the formation of secondary minerals belong to the weathering rates. An advancing weathering from N to S is also reflected by the soil development ranging from Regosols (initial A-C soils) via Cambisols (A-B-C soils) to Podzols (A-E-Bhs-(Bw-)C soils), which goes along with increasing mineral formation and in the most advanced stage also translocation of organic matter and soil minerals. This is in line with the cited soil production rates.

2. I have also difficulties to understand what your "ecosystem nutrient uptake fluxes" are. Is this the current annual uptake or is it a mean of a certain period? In l. 613,

you mention net primary production (NPP) and one line later "GrowthRate". Total nutrient uptake would, however, be related with gross primary production and growth rate sounds as if the standing nutrient stock is disregarded. Please clarify. What you call "recycling rate" (i.e. uptake rate divided by weathering rate) seems to me to be more of an "accumulation rate", because the vegetation accumulates part of the nutrients released by weathering and deposited from the atmosphere. When enough nutrients have accumulated to support a mature forest, the majority of these nutrients is recycled with losses smaller than the weathering release (or the deposition from the atmosphere on very poor soils) until the forest starts to break down and rejuvenate.

3. Because all sites are about 80 km away from the Pacific Ocean, they should be similarly affected by Sea Spray. I would even expect an increasing Sea Spray deposition with increasing rainfall. This can, however, possibly not be detected because of the simultaneously higher leaching rates at the wetter sites. Nevertheless, the Sr isotope ratio in the plants can be explained as a mixture of the isotope ratio of the rock and Sea Spray at three of the four sites. The only exception is La Campana, where perhaps indeed the uptake from the subsoil dominates the Sr isotope ratio in the plant. Please also mind that plants can take up their nutrients with all surfaces and forests therefore retain nutrients in their canopy. Besides Sea Spray, I would also expect aeolian redistribution of soil material at the driest site. How can this be considered in your budgets?

4. The question "Are nutrient sources setting plant stoichometry?" can be clearly answered with "no" based on well-established textbook knowledge. As an example, I cite an introductory sentence from the textbook "Regulation of Plant Nutrient Uptake" by G.N. Mitra (2015, Springer): "To cope with wide variations in mineral concentrations in soil, plants have evolved mechanisms so that net intake of a nutrient depends on the plant's need for this element rather than its concentration in the rooting medium." I could have cited any other textbook on plant nutrition. The author's use of plant nutritional terms is in large parts wrong. While Fe is an essential micronutrient (not only

beneficial), Al can only be considered as beneficial for some plants at low concentrations in soil solution. For most plants it can instead be toxic. All essential elements are needed at the same time so that there is not a "most plant-essential nutrient". However, the plant requires different amounts of each nutrient. In plant tissue, e.g., the mean molar ratio of the K:P concentrations is 4.2 and the mass ratio is 5.3 (Marschner, 1995, Nutrition of Higher Plants, Academic Press). Nevertheless, P is more frequently growth limiting than K. In most cases, forests are limited by one nutrient or co-limited by two and sometimes even three nutrients. You cite Stock et al. (2019) who stated that your study ecosystems are N-limited, which is in line with the global mapping of nutrient limitation by Du et al. (2020, Nat. Geosci. 13, 221-225). If this is true, no other element than N will be growth limiting. Finally, nutrient concentrations in plants usually vary by less than one order of magnitude. If you now allow for a range of two orders of magnitude (as done in Fig. 6), you will for sure get an overlap between the soil and plant stoichiometry, which is, however, meaningless. Although the organic layer will more closely reflect the plant stoichiometry, its composition is still different from that of the plants, because some nutrients are to a large degree retranslocated prior to leaf/needle abscission (e.g., P) and others are quickly leached from the organic layer (e.g., K). Overall, I think that this part of the discussion must be completely revised or omitted. It would likely make sense to seek the support of a plant nutrition specialist.

5. I don't think that your view of "abiogenic" weathering is correct. The weathering in the deep subsoil (which you call saprolith) is strongly influenced by the acids produced by biological activity. The $CO_2$ concentration in soil air is frequently one order of magnitude higher than in the free atmosphere resulting in the formation of carbonic acid which is leached to the weathering front together with organic acids driving the chemical weathering. Moreover, it is highly unlikely that your saprolith is free of biological activity (i.e. plant roots, fungi, bacteria, soil animals), which would result in a direct biological acidification by the release of protons during cationic nutrient uptake, the release of organic acids, and the production of $CO_2$ by respiration. If there are roots, as suggested for La Campana, there would even be mechanical weathering by the plants.

The source of acids originating from CO2 dissolved in rainfall is much smaller than the biological CO2 sources and it is already buffered in the vegetation canopy and the topsoil and does not reach the weathering front. I would therefore even turn around your conclusion, stating that in vegetated areas, there is likely hardly any abiogenic weathering. The latter might be restricted to not vegetated areas and mainly driven by insolation, salt and ice blasting.

6. The cation-exchange/carbonic acid proton buffer system has a small capacity and a high buffer rate. It is therefore quickly passed. More important are the carbonate/carbonic acid (pH 7-8), the Al oxide/strong acid (pH 3-5), and in the E horizon of the Podzol at Nahuelbuta the Fe oxide/strong acid (pH 2-3) buffer systems. All soils acidify in the course of their development because of a number of proton sources of which the carbonic acid formed by the much higher CO2 concentration in the soil air than in the free atmosphere is the largest one. This acidification results in acid soils with a pH < 5 in the dissolution of Al oxides and the subsequent replacement of exchangeable base metals by Al3+. I don't understand how the cation-exchange capacity can exceed the bioavailable cation pool (determined as the sum of water-extractable and NH4OAc-extractable fractions – if I understood the description of the methods correctly, where it is not described that the two fractions are summed up), because in one of the standard methods to determine the cation-exchange capacity NH4OAc is used as extractant (and anyway any salt added in excess will exchange all exchangeable cations from the soil). The apparent difference can possibly be explained by the spatial heterogeneity but not by any mechanistic background. Exchangeable cations are entirely considered plant-available. The cation-exchange process is very fast so that the composition of the ion mixture on the cation exchanger surfaces is always in equilibrium with the soil solution.

7. The finding that the more developed the soils are, the larger is the part of the nutrients needed to satisfy the plant demand that is directly cycled between the vegetation and the organic layers/mineral topsoils and thus decoupled from weathering is known

since the early 1980s going back to the work of e.g., Jordan (1982, Ecology 63, 647-654).

8. Finally, a merely formal issue: You present parts of your results threefold, i.e. in a figure, a table, and mention some numbers even additionally in the text (e.g., Fig. 4 and Table 3 or Fig. 5 and Table 6). I suggest to keep the figures and move the tables to the Supplementary Material to render the manuscript more concise.

---

## Referee Comment (RC3) · Anonymous Referee #3 · 14 May 2020

This ambitious study adds to the existing body of evidence that the relationships between plants and silicate weathering are complex, and that the concept of "biotic enhancement of weathering" in some cases is misleading. The starting point is that the 4 sites in the EarthShape project should provide evidence for an impact of vegetation on silicate weathering rate, due to the gradient in NPP and precipitation along the transect of similar lithologies and erosion rates. This is contingent on the premise that confounding factors associated with nutrient cycling by the different vegetation types along the climatic transect can be properly disentangled from silicate weathering rates.

[Figure]

The theoretical framework for this has already been established in previous work, and the authors focus on the application. The authors find that there is no positive correlation between vegetation growth and silicate weathering rates, in spite of a strong precipitation gradient. They use this to infer that plants may have negative impacts on silicate weathering rates and that the global silicate-weathering cycle may not be driven by the impacts of plants on weathering. The results and conclusions drawn are certainly interesting and valuable, but they could be better served by a more concise data presentation. Only data that are needed to support the discussion and conclusion should be included, and reference to the tables instead of repetition of data in the text. Further, it would strengthen the discussion if the focus is on the main nutrients, as they are enough to support the main points of the paper.

Line by line comments: L11: "..these two drivers..". It is unclear what is meant – is it biogenic vs. abiogenic?

L20: Ecohydrological controls of partitioning of water between drainage and evapotranspiration may explain some of this discrepancy

L25: Taylor 2009 gives a good review of biotic impacts on weathering

L27: "weatherability" should be defined, as it may mean different things in different contexts.

L36: plants possibly affect a negative feedback that is also there without land plants. Otherwise the silicate weathering thermostat would not have worked prior to the colonization of land by plants

L100: "along the" appears twice

L125: Santa Gracia is affected by livestock gracing, which would add to nutrient export. This should be considered in the discussion later.

L129: Eco-systems are primarily N-limited. What does this imply for P-weathering and P nutrient supply? Should be added in the discussion.

L144: The sentence starting "They are thus towards the lower end of global cosmo-genic nuclide-derived soil production rates. . ." should be clarified. Do you mean overall for all 4 locations?

L256: Sentence starting with "Because. . ." should be revised for readability, e.g. by removing ". . .throughout this article. . ."

L257: add "in" before Appendix A

L273: "kinetically limited weathering regime" is an interpretation and should be included in the discussion. Have you considered that it may be "thermodynamically limited" (Winnick and Maher 2018)?

L283: Probably not all nutrients are available to ecosystems, as some leave soils in dissolved form. The statement in line 610f should be included in some form in the main text.

L298: This paragraph is hard to read. Stick to describing the trends and exclude the numbers from the text. P and K being the most important nutrients should not be called an exception to a trend. It would clarify the message overall to focus on the most important nutrients and leave the evaluation of the other elements to the appendix.

L325: In my opinion the Sr ratios mentioned here are not distinct.

L392: Why does increasing P concentration along the gradient hint at P limitation? Where is P limiting? And what is the impact of livestock on the P budget in Santa Gracia?

L446-460: In Oeser et al 2018 it was concluded that the weathering is not limited by mineral supply. This does not necessarily imply kinetically limited weathering. Equilibrium with regolith fluid characterizes a thermodynamic limit (Winnick and Maher 2018). That being said, the Nahuelbuta site could be in a kinetically limited weathering regime. It would improve the manuscript to clarify this and what role plants may play in different weathering regimes.

L577: "..to be limit. . ." - remove "be".

L581: I would revise the end statement. It is a leap to upscale from a local/regional spatial study to the global temporal cycle. Plants are not the driver of the global silicate-weathering-carbonate cycle, only a modifier in as much as they affect the atmospheric $CO_2$ level at which the silicate weathering $CO_2$ sink balances $CO_2$ sources. Therefore biotic enhancement of weathering at the global scale does not increase silicate weathering rates (in steady state).

L618: GrowthRate should be defined

Figure 2: The left panel does not correspond to the text. Is litter layer and biota one box called plants?

Table 2: Eq(4) Does this assume no recycling internally in the plant?

Eq(6) is the notation tau_x correct here?

References: Taylor et al 2009, Geobiology 7, 171-191 Winnick and Maher 2018, EPSL 485, 111-120

―――――――――――――――――――――――

---

## Author Comment (AC3) · 17 May 2020

**Reply to Referee 2**

**COMMENT:** the authors present total and bioavailable stocks of a number of nutrients and some other elements, in soil, saprolite, and vegetation along a steep climatic gradient in the coastal cordillera of Chile. From their data, they calculated a number of fluxes between the various ecosystem compartments and in and out of the whole

system. Their main interpretations are that the weathering rates – which they term abiogenic – did not change substantially along the transect in spite of the marked climatic differences while the nutrient uptake rates increased with standing biomass following the rising rainfall rates. I think that the authors produced high-quality data from an interesting environmental gradient but I disagree with large parts of their data interpretation.

**REPLY** We are grateful for the reviewer to appreciate the production of high-quality data from a spectacular vegetation gradient. Critical Zone inorganic chemical data of both the weathering zone and the plants that grow on it are still not common, in particular of the rates and fluxes involved. Presenting these quantifications is one of the main objectives of this paper. Independent of our own interpretation our large data set shall allow other scientists to evaluate hypotheses of geo-bio links that was hitherto rarely possible for lack of data that include the rates of these processes that are very hard to measure.

Yet the reviewer disagrees with large parts of our interpretation, and in doing so implicitly disputes major concepts widely accepted in weathering geochemistry. In particular, the reviewer objects to our use of the term "abiogenic weathering". We note that almost any textbook on weathering begins on the premise that rock weathering takes place through inorganic chemical reactions. In practice this was the case early in Earth's history and is possible the case at our arid field site where there is barely any vegetation (but microbes), and is likely the case on Mars...). However, we never state in the manuscript that abiogenic weathering is the only process. The objective of this paper is to evaluate whether biota accelerates, damps, or changes in any way the weathering chemistry that would have happened anyway by abiotic reactions. This is one of the grand questions regarding the long-term climate regulation of the Earth (Berner et al., 2003; Pagani et al., 2009). We will make this objective very explicit in a revised version to avoid this misunderstanding.

Furthermore, the reviewer highlights the different states of soil development between the sites (last part of review section 1) and thus contradicts anonymous reviewer 1 who

suggests that all sites are within the same "pedogenic process domain". It would be disconcerting having to embark in this discussion which seems to be about definitions, rather than about the quantification of biogeochemical fluxes and their interpretation. Of importance may be here the starting viewpoint: both the reviewers' views seem to be based on the perspective of "soil development" on stable surfaces of different ages; implying continuous development through different states. But this is not the system we analyze here: our sites are continuously eroding and thus are not subject to different stages of one evolution. Material is continuously turned over. Such a system has no age, but rather a residence time. This is the contrast between two suggested models of soil development – steady-state and continuous evolution (Lin, 2010). In a revised version we will emphasize this important difference, and that here we deal within the steady state perspective.

In general, we realize that this MS requires better introduction of paradigms and assumptions that are the basis of our interpretation and how these differ from other common perspectives. Such introductory text will make this study more easily accessible to a multi-disciplinary readership. We thank the reviewer for making us aware of this deficit.

We reply point by point to the reviewer's comments.

**COMMENT 1.1** It is hard to believe that weathering rates do not respond to the strong climatic gradient.

**REPLY** Indeed, this is really hard to believe, and it is a stunning result of this study. In particular the weathering rates between the barely vegetated semi-arid site and the fully vegetated humid site (where topographic relief and hence total denudation is similar) are almost the same, despite huge differences in climate. There is key information of biotas role in weathering here.

**COMMENT 1.2** From the explanation in l. 603-607, I read that the "most negative tau values from the shallowest mineral soil sample of each regolith profile were

used" [to calculate weathering rates] and that these values are shown in red in Fig. A1. I do not understand this. The red symbols in Fig. A1 do neither refer to the shallowest horizon nor are they consistently the most negative value. Why did you choose the most negative tau value and not the thickness-weighted average tau value of the whole regolith profile?

**REPLY** We select the shallowest sample not affected by atmospheric input or plant recycling as it reflects integrated elemental release over each entire regolith profile in an eroding regolith column. This principle is discussed extensively for eroding soils in many publications (Granger and Riebe, 2014; Riebe et al., 2004; Dixon et al., 2009; Hewawasam et al., 2013; Wackett et al., 2018). Thus, we used the most negative tau value from the shallowest mineral soil sample (B Horizon) as the metric for the weathering-column integrated fractional mass loss.

**COMMENT 1.3** I anyway think that the tau values cannot be used to estimate weathering, because they lump together a complex mixture of many processes including e.g., leaching losses, deposition input, plant uptake, or soil-internal redistribution. The latter is particularly pronounced in Nahuelbuta, where Podzols occur which show strong depletion (E) horizons and strong accumulation (Bhs) horizons.

**REPLY** There are many different ways to define, study, and quantify weathering, and thus it is no surprise that the reviewer defines estimates of weathering differently than we do. However, we disagree with the suggestion that tau values cannot be used to estimate weathering rate (where we emphasize that rates are our main objective). Deriving weathering rates from CDF (fraction of total dissolved mass loss) or from tau (fraction of elemental dissolved mass loss) in conjunction with a denudation or soil production rate from cosmogenic nuclides is now common practice in many studies of Critical Zone Geochemistry (see also Brantley and Lebedeva, 2011; Ferrier et al., 2010; Uhlig and von Blanckenburg, 2019 in addition to the references named above), and the background paper on which this publication is based on (Oeser et al., 2018). Weathering rate is defined here as the net loss of solutes from regolith and is a figure

that can be compared to e.g. river solute fluxes. Ultimately it is a metric used to calibrate global weathering fluxes. The reviewer instead seems to refer to "weathering" to denote the quantification of internal redistribution fluxes. This is an equally valid, albeit entirely different way of quantifying the state of weathering, but it is not what this study (and many other weathering studies) aims for.

**COMMENT 1.4** Instead, I think that an equation such as that proposed by Likens (2013), Biogeochemistry of a forested ecosystem (3rd ed). Springer, New York, USA should be used: TDi + Wi = STi $\pm$ DeltaBi $\pm$ DeltaOMi $\pm$ DeltaXi $\pm$ DeltaMi where TD is total deposition, i a selected element, W is weathering release from primary minerals, Delta is change, B is storage in biomass, OM is long- term storage in soil organic matter, X is the exchangeable pool and M is the secondary mineral pool.
**REPLY** We acknowledge that the reviewer proposes a different suite of metrics to quantify the internal material fluxes in the Critical Zone (i.e. Eq. (2); Likens, 2013). Indeed, it would be very insightful to establish such budget. However, due to the setup of our study we are unable to determine all the variables required to solve this equation, which was never the aim of the study. We are aware that one of the successful studies in this regard (e.g. Wilcke et al., 2017) was based on many years of careful field monitoring of many ecosystem and water variables – something that is totally beyond the scope of this project. Some of these variables, like weathering release from primary minerals, are in any case very hard to determine in a field setting. We maintain that our method to determine budgets is equally valid, even though it targets different components of the system.

**COMMENT 1.5** I particularly think that the immobilization in soil organic matter and the formation of secondary minerals belong to the weathering rates.
**REPLY** We do not disagree that it would be beneficial to know these parameters for a full characterization of the weathering system. Besides the difficulties arising in establishing rates of these parameters we suggest that whether these parameters

belong to the weathering rates is purely a matter of definition. For the question we ask – namely what are the net weathering release rates from regolith, they are not required.

**COMMENT 1.6** An advancing weathering from N to S is also reflected by the soil development ranging from Regosols (initial A-C soils) via Cambisols (A-B-C soils) to Podzols (A-E-Bhs-(Bw-)C soils), which goes along with increasing mineral formation and in the most advanced stage also translocation of organic matter and soil minerals. This is in line with the cited soil production rates.

**REPLY** It is not entirely clear to us what the suggested link between soil production rates and soil development would be telling us, besides descriptions of soil state variables that have been reported elsewhere (Bernhard et al., 2018). Maybe there is an underlying misunderstanding. As all our sites are continuously eroding (Oeser et al., 2018; Schaller et al., 2018; van Dongen et al., 2019) they do not reflect a continuous series of soil development in the sense of a "chronosequence". To quantify the permanent material turnover, we report soil production rates in L140f and Table 1. They are lowest in the arid site and highest in the mediterranean site. In the semi-arid and the humid-temperate site, they are similar. In contrast to reviewer 1, reviewer 2 (this review) seems to attribute the soils to different process domains. We appreciate these contrasting views on the pedogenetic processes at the EarthShape sites. That the reviewers suggest to employ such opposing frameworks to argue against our determination of the degree of weathering makes designing a useful revision strategy difficult. We hope for further advice from the editor.

**COMMENT 2.1** I have also difficulties to understand what your "ecosystem nutrient uptake fluxes" are. Is this the current annual uptake or is it a mean of a certain period?

**REPLY** As reported in Table 2, Eq. (4) and Table 3, and explained in detail in appendix A the total nutrient uptake fluxes are reported in mg m$^{-2}$yr$^{-1}$. They are thus reported as annual uptake fluxes. However we also describe that NPP was derived from the

LPJ GUESS model (Werner et al., 2018) and they thus reflect average Holocene values – a time scale relevant to regolith weathering.

**COMMENT 2.2** In l. 613, you mention net primary production (NPP) and one line later "GrowthRate".
**REPLY** Thank you for pointing out this inconsistency which we will correct accordingly.

**COMMENT 2.3** Total nutrient uptake would, however, be related with gross primary production and growth rate sounds as if the standing nutrient stock is disregarded. Please clarify.
**REPLY** The derivation of these equations are thoroughly described in Uhlig and von Blanckenburg (2019) (as cited in L256). These authors state that $U_{total}^X$ is calculated from NPP. This is a flux estimate, for which knowledge of the standing biomass stock is not required, since we do not evaluate short-term fluctuations in biomass.

**COMMENT 2.4** What you call "recycling rate" (i.e. uptake rate divided by weathering rate) seems to me to be more of an "accumulation rate", because the vegetation accumulates part of the nutrients released by weathering and deposited from the atmosphere. When enough nutrients have accumulated to support a mature forest, the majority of these nutrients is recycled with losses smaller than the weathering release (or the deposition from the atmosphere on very poor soils) until the forest starts to break down and rejuvenate.
**REPLY** This statement is correct if we would be exploring an ecosystem featuring a growing stock of biomass, such that nutrient accumulate in this biomass. Of course, this may well be the case after e.g. deforestation, sustained drought, or wildfires. However, with the metrics we use, we evaluate, per definition, a much longer timescale. Our weathering release rates average over millennia, the model-derived NPP estimate over the Holocene, and even the residence time of the bioavailable pool is centuries or more (inevitably only the plant chemical concentrations we measure do not integrate over

a longer time scale). Over these long timescales, elements can be safely assumed to be returned from biomass, taken up again, or, as the reviewer states, returned in a small fraction to the weathering flux. They thus do not reflect an accumulation rate.

**COMMENT 3.1** Because all sites are about 80 km away from the Pacific Ocean, they should be similarly affected by Sea Spray. I would even expect an increasing Sea Spray deposition with increasing rainfall. This can, however, possibly not be detected because of the simultaneously higher leaching rates at the wetter sites.
**REPLY** We do not understand this comment. We have determined atmospheric contribution relative to weathering input from Sr isotopes using a two-component mixing calculation using bedrock and seaspray as endmembers. Accordingly, up to 93 and 43% of Sr derived from seaspray are incorporated in the regolith profiles of Pan de Azúcar and Santa Gracia, respectively. In La Campana, this seaspray contribution is zero. In Nahuelbuta Sr isotopes do not discriminate such input. However, we regard it as unlikely as we see no elemental increase at the surface of the profiles that cannot be explained by bio-lifting. In particular Na as the most-abundant cation in seaspray is distributed uniformly with depth in the bio-available fraction at all sites except Pan de Azúcar (Table S4), which we interpret do show the absence of major net input of marine aerosol-derived elements. We discuss the impact of these results in the discussion and quantify them in Table 6. We further assume that even if there were an undetected component of that sea spray its input would immediately be flushed away at the wetter sites such that it may not be as relevant as the weathering input. In a revised version we will make these inferences more explicit.

**COMMENT 3.2** Nevertheless, the Sr isotope ratio in the plants can be explained as a mixture of the isotope ratio of the rock and Sea Spray at three of the four sites. The only exception is La Campana, where perhaps indeed the uptake from the subsoil dominates the Sr isotope ratio in the plant. Please also mind that plants can take up their nutrients with all surfaces and forests therefore retain nutrients in their canopy.

Besides Sea Spray, I would also expect aeolian redistribution of soil material at the driest site. How can this be considered in your budgets?

**REPLY** We agree that in part Sr can be taken up by plants through leaves and cite the review by Burger and Lichtscheidl (2019) for this purpose. If the plants' radiogenic Sr composition would indeed be determined by a two-component mixing between Sr derived from rock and through atmospheric deposition, one would expect higher proportions of atmospheric derived Sr in leaves than in e.g. twig and stem samples. This would lead to gradients in $^{87}$Sr/$^{86}$Sr between the different plant compartments of a single plant. However, $^{87}$Sr/$^{86}$Sr in the different plant organs are (mostly) identical within uncertainties (see associated data set; Oeser and von Blanckenburg, 2020). Thus, we rule out direct input of atmospheric Sr into leaves. We will clarify this point. Concerning internal aeolian redistribution there are, unfortunately, preciously little means to directly quantifying this. In absence of such means a common view is: if material leaves the system it would be included in the cosmogenic nuclide-derived denudation rate. If material is internally redistributed without gains or losses it would not affect the mass balance.

**COMMENT 4** The question "Are nutrient sources setting plant stochiometry?" can be clearly answered with "no" based on well-established textbook knowledge. As an example, I cite an introductory sentence from the textbook "Regulation of Plant Nutrient Uptake" by G.N. Mitra (2015, Springer): "To cope with wide variations in mineral concentrations in soil, plants have evolved mechanisms so that net intake of a nutrient depends on the plant's need for this element rather than its concentration in the rooting medium." I could have cited any other textbook on plant nutrition. The author's use of plant nutritional terms is in large parts wrong. While Fe is an essential micronutrient (not only beneficial), Al can only be considered as beneficial for some plants at low concentrations in soil solution. For most plants it can instead be toxic. All essential elements are needed at the same time so that there is not a "most plant-essential nutrient". However, the plant requires different amounts of each

nutrient. In plant tissue, e.g., the mean molar ratio of the K:P concentrations is 4.2 and the mass ratio is 5.3 (Marschner, 1995, Nutrition of Higher Plants, Academic Press). Nevertheless, P is more frequently growth limiting than K. In most cases, forests are limited by one nutrient or co-limited by two and sometimes even three nutrients. You cite Stock et al. (2019) who stated that your study ecosystems are N-limited, which is in line with the global mapping of nutrient limitation by Du et al. (2020, Nat. Geosci. 13, 221-225). If this is true, no other element than N will be growth limiting. Finally, nutrient concentrations in plants usually vary by less than one order of magnitude. If you now allow for a range of two orders of magnitude (as done in Fig. 6), you will for sure get an overlap between the soil and plant stoichiometry, which is, however, meaningless. Although the organic layer will more closely reflect the plant stoichiometry, its composition is still different from that of the plants, because some nutrients are to a large degree retranslocated prior to leaf/needle abscission (e.g., P) and others are quickly leached from the organic layer (e.g., K). Overall, I think that this part of the discussion must be completely revised or omitted. It would likely make sense to seek the support of a plant nutrition specialist.

**REPLY** We see the point that we may have over-interpreted Fig. 6 with respect to nutrient limitation and plant stoichiometry. As this section is not essential for our analysis and we can remove this without loss in any to the papers' conclusions. Thus, we will follow this advice and remove this part of the discussion, with the exception of the use of Fig.6 as these correlations are evidence that the plant-available pool is indeed the mineral nutrient source.

**COMMENT 5.1** I don't think that your view of "abiogenic" weathering is correct. The weathering in the deep subsoil (which you call saprolith) is strongly influenced by the acids produced by biological activity. The $CO_2$ concentration in soil air is frequently one order of magnitude higher than in the free atmosphere resulting in the formation of carbonic acid which is leached to the weathering front together with organic acids driving the chemical weathering. Moreover, it is highly unlikely that your saprolith is

free of biological activity (i.e. plant roots, fungi, bacteria, soil animals), which would result in a direct biological acidification by the release of protons during cationic nutrient uptake, the release of organic acids, and the production of $CO_2$ by respiration. If there are roots, as suggested for La Campana, there would even be mechanical weathering by the plants.

**REPLY** We are puzzled that this comment is even made. The term "abiogenic weathering" appears only once in the manuscript (in the abstract). In the remainder of the paper we explicitly summarize the potential influence of plants on weathering (L503ff). Our list of mechanisms includes the respiratory release of $CO_2$ (point 4). We never state that the saprolite in the EarthShape sites is free of biological activity. Rather, we discuss a variety of mechanisms on how plants might accelerate weathering, including the forces introduced by plant roots and the effects of mycorrhizal fungi and soil microbiota.

However, as stated above, common weathering geochemistry is treating the weathering zone as a conceptual abiogenic endmember model, onto which biological processes are superimposed either conceptually or from observation. Deciphering whether weathering is more intense, or its fluxes are higher in the presence of plants has never been explicitly resolved. This is the objective of this study. This is a major question in weathering Geochemistry (e.g. Berner et al., 2003; Calmels et al., 2014; D'Antonio et al., 2019; Kump et al., 2000; Porder, 2019; Quirk et al., 2012).

**COMMENT 5.2** The source of acids originating from $CO_2$ dissolved in rainfall is much smaller than the biological $CO_2$ sources and it is already buffered in the vegetation canopy and the topsoil and does not reach the weathering front. I would therefore even turn around your conclusion, stating that in vegetated areas, there is likely hardly any abiogenic weathering. The latter might be restricted to not vegetated areas and mainly driven by insolation, salt and ice blasting.

**REPLY** This is pretty much EXACTLY what we say (Line 547 point 4). A view is suggested here that we never made. The title of this section is: "Is weathering modulated

by biota?". The question whether all weathering is "abiogenic" is an artificial one not posed in this paper. Rather, the question is whether weathering RATES depend on the production of biomass, above- and belowground. We refer to a recent hypothesis paper exposing this question explicitly (Brantley et al., 2011).

**COMMENT 6** The cation-exchange/carbonic acid proton buffer system has a small capacity and a high buffer rate. It is therefore quickly passed. More important are the carbonate/carbonic acid (pH 7-8), the Al oxide/strong acid (pH 3-5), and in the E horizon of the Podzol at Nahuelbuta the Fe oxide/strong acid (pH 2-3) buffer systems. All soils acidify in the course of their development because of a number of proton sources of which the carbonic acid formed by the much higher $CO_2$ concentration in the soil air than in the free atmosphere is the largest one. This acidification results in acid soils with a pH < 5 in the dissolution of Al oxides and the subsequent replacement of exchangeable base metals by $Al^{3+}$. I don't understand how the cation-exchange capacity can exceed the bioavailable cation pool (determined as the sum of water-extractable and $NH_4OAc$- extractable fractions – if I understood the description of the methods correctly, where it is not described that the two fractions are summed up), because in one of the standard methods to determine the cation-exchange capacity $NH_4OAc$ is used as extractant (and anyway any salt added in excess will exchange all exchangeable cations from the soil). The apparent difference can possibly be explained by the spatial heterogeneity but not by any mechanistic background. Exchangeable cations are entirely considered plant-available. The cation-exchange process is very fast so that the composition of the ion mixture on the cation exchanger surfaces is always in equilibrium with the soil solution.

**REPLY** We not understand the point the reviewer is making. We will address this issue by removing the section on cation exchange capacity. The main point is that the bioavailable pool is smaller at the humid site. This can be easily explained by the lower pH and the higher runoff.

**COMMENT 7** The finding that the more developed the soils are, the larger is the part of the nutrients needed to satisfy the plant demand that is directly cycled between the vegetation and the organic layers/mineral topsoils and thus decoupled from weathering is known since the early 1980s going back to the work of e.g., Jordan (1982, Ecology 63, 647- 654).

**REPLY** The cited paper by Jordan (1982) does not show recycling. The paper reports on an imbalance in atmospheric and dissolved export fluxes (interpreted to show net forest growth) but not nutrient recycling. We wonder how recycling rates were determined in the 1980's, as soil profile estimates of weathering fluxes (like from cosmogenic nuclides) did not exist, and high-quality data on plant stoichiometry was rare. We would be grateful for recommendations of other literature than Jordan (1982). In any case, filling these gaps by suggesting means to quantify nutrient recycling is one of the major objective of this paper.

**References**

Berner, E. K., Berner, R. A., and Moulton, K. L.: Plants and Mineral Weathering: Present and Past, in: Treatise on Geochemistry, 169-188, 2003.

Bernhard, N., Moskwa, L.-M., Schmidt, K., Oeser, R. A., Aburto, F., Bader, M. Y., Baumann, K., von Blanckenburg, F., Boy, J., van den Brink, L., Brucker, E., Canessa, R., Dippold, M. A., Ehlers, T. A., Fuentes, J. P., Godoy, R., Köster, M., Kuzyakov, Y., Leinweber, P., Neidhard, H., Matus, F., Mueller, C. W., Oelmann, Y., Oses, R., Osses, P., Paulino, L., Schaller, M., Schmid, M., Spielvogel, S., Spohn, M., Stock, S., Stroncik, N., Tielbörger, K., Übernickel, K., Scholten, T., Seguel, O., Wagner, D., and Kühn, P.: Pedogenic and microbial interrelations to regional climate and local topography: new insights from a climate gradient (arid to humid) along the Coastal Cordillera of Chile, Catena, 170, 10.1016/j.catena.2018.06.018, 2018.

Brantley, S. L., and Lebedeva, M.: Learning to Read the Chemistry of Regolith to Understand the Critical Zone, Annual Review of Earth and Planetary Sciences, 39,

387-416, 10.1146/annurev-earth-040809-152321, 2011.

Brantley, S. L., Megonigal, J. P., Scatena, F. N., Balogh-Brunstad, Z., Barnes, R. T., Bruns, M. A., Van Cappellen, P., Dontsova, K., Hartnett, H. E., Hartshorn, A. S., Heimsath, A., Herndon, E., Jin, L., Keller, C. K., Leake, J. R., McDowell, W. H., Meinzer, F. C., Mozdzer, T. J., Petsch, S., Pett-Ridge, J., Pregitzer, K. S., Raymond, P. A., Riebe, C. S., Shumaker, K., Sutton-Grier, A., Walter, R., and Yoo, K.: Twelve testable hypotheses on the geobiology of weathering, Geobiology, 9, 140-165, 10.1111/j.1472-4669.2010.00264.x, 2011.

Burger, A., and Lichtscheidl, I.: Strontium in the environment: Review about reactions of plants towards stable and radioactive strontium isotopes, Sci. Total Environ., 653, 1458-1512, 10.1016/j.scitotenv.2018.10.312, 2019.

Calmels, D., Gaillardet, J., and François, L.: Sensitivity of carbonate weathering to soil $CO_2$ production by biological activity along a temperate climate transect, Chem. Geol., 390, 74-86, 2014.

D'Antonio, M., Ibarra, D. E., and Boyce, C. K.: Land plant evolution decreased, rather than increased, weathering rates, Geology, 10.1130/g46776.1, 2019.

Dixon, J. L., Heimsath, A. M., and Amundson, R.: The critical role of climate and saprolite weathering in landscape evolution, Earth Surface Processes and Landforms, 34, 1507-1521, 10.1002/esp.1836, 2009.

Ferrier, K. L., Kirchner, J. W., Riebe, C. S., and Finkel, R. C.: Mineral-specific chemical weathering rates over millennial timescales: Measurements at Rio Icacos, Puerto Rico, Chem. Geol., 277, 101-114, 10.1016/j.chemgeo.2010.07.013, 2010.

Granger, D., and Riebe, C.: Cosmogenic nuclides in weathering and erosion, in: Treatise on Geochemistry, edited by: Holland, H. D., and Turekian, K. K., Elsevier, 1-43, 2014.

Hewawasam, T., von Blanckenburg, F., Bouchez, J., Dixon, J. L., Schuessler, J. A.,

and Maekeler, R.: Slow advance of the weathering front during deep, supply-limited saprolite formation in the tropical Highlands of Sri Lanka, Geochim. Cosmochim. Acta, 118, 202-230, 10.1016/j.gca.2013.05.006, 2013.

Kump, L. R., Brantley, S. L., and Arthur, M. A.: Chemical Weathering, Atmospheric CO2, and Climate, Annual Review of Earth and Planetary Sciences, 28, 611-667, 10.1146/annurev.earth.28.1.611, 2000.

Likens, G. E.: Biogeochemistry of a Forested Ecosystem, 3rd ed., 2013. Lin, H.: Linking principles of soil formation and flow regimes, Journal of Hydrology, 393, 3-19, 10.1016/j.jhydrol.2010.02.013, 2010.

Oeser, R. A., Stroncik, N., Moskwa, L.-M., Bernhard, N., Schaller, M., Canessa, R., van den Brink, L., Köster, M., Brucker, E., Stock, S., Fuentes, J. P., Godoy, R., Matus, F. J., Oses Pedraza, R., Osses McIntyre, P., Paulino, L., Seguel, O., Bader, M. Y., Boy, J., Dippold, M. A., Ehlers, T. A., Kühn, P., Kuzyakov, Y., Leinweber, P., Scholten, T., Spielvogel, S., Spohn, M., Übernickel, K., Tielbörger, K., Wagner, D., and von Blanckenburg, F.: Chemistry and Microbiology of the Critical Zone along a steep climate and vegetation gradient in the Chilean Coastal Cordillera., Catena, 170, 183-203, 10.1016/j.catena.2018.06.002, 2018.

Oeser, R. A., and von Blanckenburg, F.: Dataset for evaluation element fluxes released by weathering and taken up by plants along the EarthShape climate and vegetation gradient, GFZ Data Services, 2020.

Pagani, M., Caldeira, K., Berner, R., and Beerling, D. J.: The role of terrestrial plants in limiting atmospheric CO(2) decline over the past 24 million years, Nature, 460, 85-88, 10.1038/nature08133, 2009.

Porder, S.: How Plants Enhance Weathering and How Weathering is Important to Plants, Elements, 15, 241-246, 10.2138/gselements.15.4.241, 2019.

Quirk, J., Beerling, D. J., Banwart, S. A., Kakonyi, G., Romero-Gonzalez, M. E., and

Leake, J. R.: Evolution of trees and mycorrhizal fungi intensifies silicate mineral weathering, Biol. Lett., 8, 1006-1011, 10.1098/rsbl.2012.0503, 2012.

Riebe, C. S., Kirchner, J. W., and Finkel, R. C.: Sharp decrease in long-term chemical weathering rates along an altitudinal transect, Earth. Planet. Sci. Lett., 218, 421-434, 10.1016/s0012-821x(03)00673-3, 2004.

Schaller, M., Ehlers, T. A., Lang, K. A. H., Schmid, M., and Fuentes-Espoz, J. P.: Addressing the contribution of climate and vegetation cover on hillslope denudation, Chilean Coastal Cordillera (26°–38°S), Earth. Planet. Sci. Lett., 489, 111-122, 10.1016/j.epsl.2018.02.026, 2018.

Uhlig, D., and von Blanckenburg, F.: How Slow Rock Weathering Balances Nutrient Loss During Fast Forest Floor Turnover in Montane, Temperate Forest Ecosystems, Frontiers in Earth Science, 7, 10.3389/feart.2019.00159, 2019.

van Dongen, R., Scherler, D., Wittmann, H., and von Blanckenburg, F.: Cosmogenic 10Be in river sediment: where grain size matters and why, Earth Surface Dynamics, 7, 393-410, 10.5194/esurf-7-393-2019, 2019.

Wackett, A. A., Yoo, K., Amundson, R., Heimsath, A. M., and Jelinski, N. A.: Climate controls on coupled processes of chemical weathering, bioturbation, and sediment transport across hillslopes, Earth Surface Processes and Landforms, 43, 1575-1590, 10.1002/esp.4337, 2018.

Werner, C., Schmid, M., Ehlers, T. A., Fuentes-Espoz, J. P., Steinkamp, J., Forrest, M., Liakka, J., Maldonado, A., and Hickler, T.: Effect of changing vegetation and precipitation on denudation – Part 1: Predicted vegetation composition and cover over the last 21 thousand years along the Coastal Cordillera of Chile, Earth Surface Dynamics, 6, 829-858, 10.5194/esurf-6-829-2018, 2018.

Wilcke, W., Velescu, A., Leimer, S., Bigalke, M., Boy, J., and Valarezo, C.: Biological versus geochemical control and environmental change drivers of the base metal budgets of a tropical montane forest in Ecuador during 15 years, Biogeochemistry, 136, 167-189, 10.1007/s10533-017-0386-x, 2017.

---

## Author Comment (AC4) · 19 May 2020

**REPLY** We thank the reviewer for acknowledging the value of this study value in deciphering the complex interactions between plants and silicate weathering. We appreciate these constructive comments that point at possible improvements in the data presentation and the discussion that can be dealt with as suggested.
We reply point by point to the reviewer's comments.

**COMMENT L11:** "..these two drivers..". It is unclear what is meant – is it bio-

genic vs. abiogenic?

**REPLY** Yes, we mean the relative impact of biogenic vs. abiogenic weathering. We will clarify this accordingly.

**COMMENT L20:** Ecohydrological controls of partitioning of water between drainage and evapo- transpiration may explain some of this discrepancy

**REPLY** We discuss this partitioning in Line 528ff. According to Ibarra et al. (2019) total runoff can decrease by up to 23% as vegetation cover raises from barely to highly vegetated sites. However, we find that this reduction is a minor effect when compared to the 100-fold increase in precipitation over the entire EarthShape gradient.

**COMMENT L25:** Taylor 2009 gives a good review of biotic impacts on weathering

**REPLY** Thank you for directing us at this useful review. We already discuss many of the mechanisms of biotic weathering in discussion section "is weathering modulated by biota?" (Line 497ff). However, Table 1 presented in this paper will serve as a useful resource that we will cite to summarize previous field studies done to explore these interactions.

**COMMENT L27:** "weatherability" should be defined, as it may mean different things in different contexts.

**REPLY** With "weatherability" we refer to the susceptibility of minerals to weathering, i.e. dissolution. We will clarify this in a future version of the text accordingly.

**COMMENT L36:** plants possibly affect a negative feedback that is also there without land plants. Otherwise the silicate weathering thermostat would not have worked prior to the colonization of land by plants

**REPLY** We will rewrite this statement to emphasize that plants have strengthened the negative feedback that already existed by abiogenic weathering.

**COMMENT L125:** Santa Gracia is affected by livestock grasing, which would add to nutrient export. This should be considered in the discussion later.
**REPLY** Grazing would indeed lead to increased nutrient export. We just wonder whether the more recent advent of grazing has already contributed to weathering given that the timescale over which the bio-available fraction resides is a few centuries –longer than grazing (see turnover times, Table 8). Nonetheless, we will add this point in the discussion.

**COMMENT L129:** Eco-systems are primarily N-limited. What does this imply for P-weathering and P nutrient supply? Should be added in the discussion.
**REPLY** It is a common observation that the study sites are primarily N-limited (Stock et al., 2019). With increasing NPP along the gradient, however, a N-P co-limitation on plant growth might develop, because Stock et al. (2019) found an increased activity of P-acquiring enzymes in the mediterranean and humid-temperate site compared to the (semi-) arid sites.

**COMMENT L144:** The sentence starting "They are thus towards the lower end of global cosmogenic nuclide-derived soil production rates. . ." should be clarified. Do you mean overall for all 4 locations?
**REPLY** We mean all four locations. Global cosmogenic nuclide-derived soil production rates are up to 20-fold higher than those reported for La Campana (see e.g. compilation by Dixon and von Blanckenburg, 2012)

**COMMENT L273:** "kinetically limited weathering regime" is an interpretation and should be included in the discussion. Have you considered that it may be "thermodynamically limited" (Winnick and Maher 2018)?
**REPLY** "Kinetically limited weathering regime", meaning that here are primary minerals left in regolith because erosion is sufficiently high such that weatherable minerals still

remain and the weathering rate is limited by mineral dissolution kinetics (e.g. Dixon et al., 2012), is in our opinion a factual observation, and hence we think it is not a discussion item. We consider the "Thermodynamic limited" weathering regime to be a subset of the kinetic limit, namely the balance between dissolving primary minerals and precipitating secondary phases at a metastable equilibrium. In the absence of concentration-discharge data we have no means to investigate the chemostatic behavior that would result from the thermodynamic limit. This is why we do not discuss this topic.

**COMMENT L283:** Probably not all nutrients are available to ecosystems, as some leave soils in dissolved form. The statement in line 610f should be included in some form in the main text.
**REPLY** We agree with this comment and will emphasize that the parameterized weathering fluxes are upper estimates of potential nutrient uptake.

**COMMENT L298:** This paragraph is hard to read. Stick to describing the trends and exclude the numbers from the text. P and K being the most important nutrients should not be called an exception to a trend. It would clarify the message overall to focus on the most important nutrients and leave the evaluation of the other elements to the appendix.
**REPLY** We can follow this suggestion, but we note that many readers wish to see at least some data in the results section.

**COMMENT L325:** In my opinion the Sr ratios mentioned here are not distinct.
**REPLY** We will rephrase the sentence accordingly.

**COMMENT L392:** Why does increasing P concentration along the gradient hint at P limitation? Where is P limiting? And what is the impact of livestock on the P budget in Santa Gracia?

**REPLY** Please see our comment to L129. As suggested by reviewers #1 and #2, we will remove the section "Are nutrient sources setting plant stoichiometry?" from the discussion. This section is not essential for our analysis and we can remove this without loss in any of the manuscript's conclusions.

**COMMENT L446-460:** In Oeser et al 2018 it was concluded that the weathering is not limited by mineral supply. This does not necessarily imply kinetically limited weathering. Equilibrium with regolith fluid characterizes a thermodynamic limit (Winnick and Maher 2018). That being said, the Nahuelbuta site could be in a kinetically limited weathering regime. It would improve the manuscript to clarify this and what role plants may play in different weathering regimes.
**REPLY** Is it not valid to assume that in these sites primary mineral dissolution is limited by the kinetics of mineral dissolution reactions (Dixon et al. 2012)? We consider the "Thermodynamic limited" weathering regime to be a subset of the kinetic limit, namely the balance between dissolving primary minerals and precipitating secondary phases at a metastable equilibrium. In the absence of concentration-discharge data we have no means to investigate the chemostatic behavior that would result from the thermodynamic limit. This is why we do not discuss this topic. However, to do justice to this comment we can reword "kinetically limited" into "supply limited".

**COMMENT L581:** I would revise the end statement. It is a leap to upscale from a local/regional spatial study to the global temporal cycle. Plants are not the driver of the global silicate-weathering-carbonate cycle, only a modifier in as much as they affect the atmospheric CO2 level at which the silicate weathering CO2 sink balances CO2 sources. Therefore, biotic enhancement of weathering at the global scale does not increase silicate weathering rates (in steady state).
**REPLY** It is good to point out that plants cannot produce a weathering flux that exceeds CO2 supply by volcanic emissions. However, because of the many reactions that plants and subsoil microbiota induce they have been suggested to make the delivery

of alkalinity into the oceans more efficient. Thus biogenic weathering would impact climate by setting lower atmospheric CO2 levels and are thus thought potentially drive global cooling (Berner et al., 2003; Pagani et al., 2009). We can clarify this point, but we wish to emphasize that the ability to recycle elements damps this response.

**COMMENT L618:** GrowthRate should be defined
**REPLY** Here we use NPP and Growth rate synonymously. This is indeed misleading, and we will use NPP instead.

**COMMENT Figure 2:** The left panel does not correspond to the text. Is litter layer and biota one box called plants?
**REPLY** We will rectify this in a future version of the figure and its caption.

**COMMENT Table 2:** Eq(4) Does this assume no recycling internally in the plant? Eq(6) is the notation tau$_x$ correct here?
**REPLY** Eq (4):Our parameterization of the nutrient uptake rate is independent on internal nutrient recycling. Eq (6) Indeed, it should be $\tau^X$. We will correct this accordingly.

**References**

Berner, E. K., Berner, R. A., and Moulton, K. L.: Plants and Mineral Weathering: Present and Past, in: Treatise on Geochemistry, 169-188, 2003.

Dixon, J. L., Hartshorn, A. S., Heimsath, A. M., DiBiase, R. A., and Whipple, K. X.: Chemical weathering response to tectonic forcing: A soils perspective from the San Gabriel Mountains, California, Earth. Planet. Sci. Lett., 323-324, 40-49, 10.1016/j.epsl.2012.01.010, 2012.

Dixon, J. L., and von Blanckenburg, F.: Soils as pacemakers and limiters of global silicate weathering, Comptes Rendus Geoscience, 344, 597-609, 10.1016/j.crte.2012.10.012, 2012.

Ibarra, D. E., Rugenstein, J. K. C., Bachan, A., Baresch, A., Lau, K. V., Thomas, D. L., Lee, J.-E., Boyce, C. K., and Chamberlain, C. P.: Modeling the consequences of land plant evolution on silicate weathering, Am. J. Sci., 319, 1-43, 10.2475/01.2019.01, 2019.

Pagani, M., Caldeira, K., Berner, R., and Beerling, D. J.: The role of terrestrial plants in limiting atmospheric CO(2) decline over the past 24 million years, Nature, 460, 85-88, 10.1038/nature08133, 2009.

Stock, S. C., Köster, M., Dippold, M. A., Nájera, F., Matus, F., Merino, C., Boy, J., Spielvogel, S., Gorbushina, A., and Kuzyakov, Y.: Environmental drivers and stoichiometric constraints on enzyme activities in soils from rhizosphere to continental scale, Geoderma, 337, 973-982, 10.1016/j.geoderma.2018.10.030, 2019.

---

## Author Comment (AC5) · 28 May 2020

**Final response**

The interactive discussion in Biogeosciences has resulted in a two-faced impression. One anonymous reviewer and the short comment from Marjin van de Broek endorse our efforts in development of novel ways to quantify the relationship between rock weathering and nutrient uptake. These efforts are met with skepticism from two other reviewers. Given that what we do is a new way to look at a well-, but hitherto

differently researched object, we are not surprised by this outcome. Yet we are grateful for all reviewers for such a detailed critique of our work, undoubtedly allowing us to design a better structured and much more accessible manuscript. Our aim is to make these new methods emerging from geochemistry aware to an interdisciplinary readership such as found in *Biogeosciences*. Time will tell whether these approaches lead to new trajectories in thinking about Critical Zone processes.

From the reviews, we identified several priority revision items. In particular these include the addition of *introductory text* aiming at clarifying the viewpoint of different disciplines and thereby avoiding the trap of misunderstandings apparent in some of the reviews:

- The comparison between fundamental concepts in weathering geochemistry and soil formation. Namely the concept of steady state (which we apply in our study) where regolith is constantly rejuvenated by regolith production at depth and its removal through erosion from above, and e.g. the "pedogenic threshold" or other more common soil science concepts which have mostly been developed for chronosequences, where the soils have a distinct age and undergo several phases of soil development.

- Key in this comparison is emphasis on the different time scales over which the various metrics integrate. Fluxes from the way we conduct weathering geochemistry integrate over millennia. The processes we decipher are thus long-term patterns set by the drivers that average or integrate over these time scales. This thinking differs (and likely might well lead to different views) from that of e.g. soil-ecological plot experiments where processes over the annual time scale can be monitored.

We further identified these potential modifications addressing major reviewer com-

ments:

- We are convinced by the referees' concern of over-interpretation of our data on plant stoichiometry. We will thus remove this section from the manuscript. This section is not essential for our analysis and the removal will not infer a loss in the conclusions.

- We will strongly clarify the misunderstanding in that we never suggested that plants to do contribute to silicate weathering. The question is how much do plants modify weathering under different degrees of plant cover and primary production. These relationships and their quantification are major unresolved issues in weathering Geochemistry and the weathering controls over the global carbon cycle.

- We are very much aware that resolving the interactions between ecosystem productivity and silicate weathering is not an easy task. However, we are convinced that our data set on chemistry in the weathering zone, soil, and plants in four study sites along the Earth's most extreme climate and biological gradient provides an excellent basis to tackle this task. We aim at addressing the involved confounding issues, in particular relief and erosion as the possibly stronger weathering flux driver, in a brief separate section.

- Because two reviewers were not convinced by the thrust of our interpretation, we will tune down the conclusion and rephrase the manuscripts' title as a question. For example: Do silicate weathering degree and flux depend on plant primary productivity?

- We note that some major reviewer comments are not justified. One reviewer was concerned by the lack of replication, which is wrong. We analyzed four replicate soil profiles at each site, and two of these in greater detail. There was concern over erosion being the larger control, inspired the high rate at one of the sites. We

addressed this concern already by basing our main comparison on the two sites that are similar in erosion rate and relief. One comment questioned our choice of tau values, used to calculate elemental weathering fluxes. However, what we do is standard practice in weathering rate calculation. The input of seaspray was questioned, even though we address this by Sr radiogenic isotope ratios.

- However, we will strongly revise the text for clarity, brevity and logic, as suggested.

---

## Author Response (AR1)

Dear Prof. Dr. Jack Middelburg,

please find enclosed our revised manuscript with the revised title "**Do degree and rate of silicate weathering depend on plant productivity?**" (bg-2020-69) by Oeser and von Blanckenburg.

We would like to thank you and the reviewers for the lively and interactive discussion in *Biogeosciences*, allowing us to design a better structured and much more accessible manuscript.

From the reviews, we identified several priority revision items (see also our final response submitted on May 28[th]) that we addressed in this revised version of the manuscript. These are:

(1) The text was strongly revised for clarity and logic.

(2) We addressed the misunderstandings arising from possibly contrasting, but not contradictory viewpoints emerging from different scientific disciplines. A new section called "2 *Conceptual perspectives*" describing the different models of soil development and weathering was added. In this section, we focus on the continuous evolution model and the steady state model of soil development and weathering, and put emphasis on the relevant timescales over which the different metrics integrate.

(3) We were convinced by the reviewers' concerns regarding the over-interpretation of our data on plant stoichiometry. This section has been removed from the manuscript without loss for the conclusions of the manuscript.

(4) We strongly clarified that we never suggested that plants do not contribute to silicate weathering.

(5) We tested the main question, whether plants accelerate weathering more so than other drivers, by a statistical analysis. We first performed an ANOVA to statistically test for significant differences in expressions of weathering (i.e. degree and rate of weathering) between the sites. We then determined Pearsons's correlation coefficients between denudation rate D, mean annual precipitation MAP, and net primary productivity NPP, respectively, on the one hand, and the metrics quantifying weathering on the other hand, to single out the possible biogenic weathering driver from confounding effects. This statistical

analysis was further complemented by an assessment of possible confounding effects of differences in bedrock mineralogy on weathering.

(6) The section dealing with the impact of plants on weathering has been revised. We amended the analysis presented in the previous version by a concept in which plants actively discriminate for or against silicon, hence setting the solubility limits of silicates – the main constituents in regolith on granitoid substrate.

(7) Figures were checked for readability for color-blind readers using the software Color Oracle (http://colororacle.org). The color schemes were changed in a way that they are now readable in grayscale and by readers affected by Deuteranopia.

(8) Appendix B was added. Here we provide information on sample replication and the applied statistical methods. The Tables A1 – A3 show these statistical evaluations.

(9) We toned down our main conclusion – essentially asking whether plant activity might dampen and not necessarily accelerate silicate weathering, depending on overall setting.

(10) To reflect this change in thrust we have changed the title from "**Decoupling silicate weathering from primary productivity – how ecosystems regulate nutrient uptake along a climate and vegetation gradient**" to "**Do degree and rate of silicate weathering depend on plant productivity?**"

We do note that we here do not response point by point to the reviewers' comments which we have already done in our author responses (AC1: May 05; AC2: May 08; AC3: May 17; and AC4: May 19). Instead we summarize the changes in the manuscript below:

**Revision notes**

**L90 – 95:** We explicitly address the challenge in disentangling biotic from abiotic weathering drivers caused by confounding effects in a dedicated paragraph.

**L185 – 395:** We added a section "*2 Conceptual perspectives*" describing the fundamental different concepts that describe the relationship between regolith formation and time. In this section we compare the continuous evolution model with the steady-state model and highlight the different timescales over which the various metrics integrate. The subsequent sections' numbering has been changed accordingly.

**L516 – 521:** A paragraph was added to point out that our study sites do indeed cross of several pedogenic thresholds along the north-south transect.

**L551 – 561:** Additional description of *vegetation sampling* has been added.

**L678:** Text was added to clarify the use of one $10^{12}$ Ω resistor instead of a $10^{11}$ Ω resistor to detect the intensities of the ion beams measuring to $^{82}$Kr. However, whether a $10^{11}$ Ω or a $10^{12}$ Ω resistor is used has no effect on the $^{87}$Sr/$^{86}$Sr isotope composition

**L682:** We added text to describe how the correction for a session offset were performed.

**L690:** Text was added to refer to Appendix B (statistical analysis)

**L713 – 720:** We emphasized that whether a mineral nutrient is beneficial or essential to plants is, to a certain extent, species-dependent.

**L758 – 761:** We added a paragraph describing the range in soil weathering rates W along the north-south transect. Further, elemental weathering fluxes in this section do include associated uncertainties.

**L810:** In the section's header, "*ecosystems*" has been replaced by "*plants*".

**L810 – 829**: Further text on the total inventories in saprolite, soil and bulk regolith was added.

**L881 – 891:** Text that explains the high elemental Al and Na concentration in the northernmost site that was formerly contained in section "Nutrient recycling as buffering mechanism" was moved into this section.

**L892 – 898:** Uncertainties of nutrient uptake fluxes were added.

**L903 – 905:** We rectified that the radiogenic Sr composition in Pan de Azúcar indeed differs between the single regolith profiles.

**L1004 – 1146:** Text formerly contained in the section "*Are nutrient sources setting plant stoichiometry*" was included in this section and toned down. We use the X:P ratio in plants and the bio-available fraction (Fig. 6), respectively, and the $^{87}$Sr/$^{86}$Sr ratio in the bio-available fraction to locate the nutrient pool of plants rather than inferring nutrient limitation.

**L1147 ff:** The section's header was changed from "*Nutrient recycling as buffering mechanism*" to "*An increase in nutrient recycling with NPP*". We link the fluxes of the geogenic nutrient pathway and the organic nutrient cycle and describe the increasing recycling rates along the north-south transect, and how increasing recycling affects he X:P ratio in the bio-available fraction.

**L1287:** This section's header was changed from "*How the organic and the geogenic nutrient pathway set the size of the bioavailable pool*" to "*Processes that set the size of the bio-available pool.*"

**L1287 – 1349:** This section was strongly revised and now discusses the main processes that set the size of the bio-available pool at conceptual steady state. We further tuned down the importance of $CEC_{eff}$ in setting the size of that pool and conclude that retention capacity and water flow ultimately set its size.

**L1350 – 1551**: This section's header was changed from "*How the organic and the geogenic nutrient pathway set the size of the bioavailable pool*" to "*Concepts for biotas role in setting fluxes in the geogenic and the organic nutrient cycle*". The section was strongly revised. Here we discuss the role of biota in setting the delicate balance between the organic and geogenic nutrient pathway. We discuss how recycling and biogenic weathering can in general contribute towards this balance.

**L1552ff:** This section's header was changed from "*Is weathering modulated by biota?*" to "*Is weathering modulated by biota? A statistical approach.*" In this section, we discuss correlational statistical methods to single out the possible biogenic weathering driver from the confounding factors. We define three hypothesis on the potential drivers of weathering and its potential drivers: (1) Where denudation rate D is high bulk weathering fluxes are high… (2) At sites at which MAP and hence runoff is high, weathering fluxes are high… (3) If NPP is high the degree and rate of weathering will be high… . Using this statistical approach, we found that neither MAP nor NPP have a significant effect of rate and degree of weathering. Instead, denudation rate the is the main driver. We further expand the analysis of confounding factors to include the differences in initial bedrock composition and mineralogy that exist between Santa Gracia and Nahuelbuta, and the resulting effects of weathering. In summary, we discuss that in Santa Gracia the mineral suite is more susceptible to weathering than in Nahuelbuta and might result in the observed similarities in weathering fluxes between the two sites despite massive differences in vegetation cover, NPP, and MAP.

**L1613ff:** A subsection named "*Do negative feedbacks decouple biomass growth from weathering rate and degree?*" was added and returns to the analysis of the suite of processes how plants and their associated biota directly and indirectly impact weathering. We particularly focus in point (D) on how plants can increase or decrease solubility of silicate minerals by Si accumulation or by discriminating against Si during uptake. The Si-uptake flux is minor compared to its release by weathering. The ecosystems at the EarthShape sites can be regarded to be below the threshold considered for Si accumulators. We thus conclude that Si uptake by plants does not contribute towards increasing weathering rates. Given this

observation together with the analysis of pedogenic oxides, we argue that in the humid-temperate site of Nahuelbuta, Si weathering rates are subdued despite the elevated solubility of primary minerals due to increased $CO_2$ respiration by roots in regolith.

**L1771 – 1779:** We toned down the summary of this section. We state that we did not find evidence for coupling of silicate weathering fluxes with the nutrient demands of biota to an extent that exceeds other controlling factors of weathering.

**L1812 – 1820:** We toned down our main conclusion so as to direct the reader at possibly rethinking common views, which is the aim of our study.

Best regards,

Ralf Oeser and Friedhelm von Blanckenburg

[revised manuscript text omitted]

**hat gelöscht:** Oeser et al. (2018)
**hat gelöscht:** study
**hat gelöscht:** i.e. τ, almost 60
**hat gelöscht:** of Ca and Na
**hat gelöscht:** a.s.l.
**hat gelöscht:** mild

**hat gelöscht:** Table 1).
**hat gelöscht:** for example
**hat gelöscht:** accordingly
**hat gelöscht:** (south-facing, Fig. A1; Table S2; Brantley and Lebedeva, 2011)
**hat gelöscht:** a.s.l.
**hat gelöscht:** Table 1).
**hat gelöscht:** Podsol
**hat gelöscht:** Umbrisol

**hat gelöscht:** ¶

**hat gelöscht:** eight regolith profiles'
**hat gelöscht:** Table 1

**4 Methods**

**4.1 Sampling**

Regolith samples were collected in a continuous sequence of depth increments from bottom to top. Increments amount to a thickness of 5 cm for the uppermost two samples, 10 cm for the 3rd sample from top, and increase to 20 cm thickness for the 4th sample onwards. To account for the dependence on solar radiation, two regolith profiles on adjacent hillslopes (north- and south-facing) were sampled at each study site (see Appendix B for further information on sample replication).

The underlying unweathered bedrock was not reached in any of the regolith profiles and the depth to bedrock remains unknown. Thus, bedrock samples were collected from nearby outcrops. This sample set comprises the 20 bedrock samples already reported in Oeser et al. (2018) and 15 additional bedrock samples (in total 12 in Pan de Azúcar, 8 in Santa Gracia, 10 in La Campana, and 5 in Nahuelbuta).

Vegetation samples from representative shrubs and trees (grasses have been excluded) of each study site were sampled in the austral summer to autumn 2016. The sample set comprises material from mature plants of the prevailing species: *Nolana mollis* (Pan de Azúcar), *Asterasia* sp., *Cordia decandra*, *Cumulopuntia sphaerica*, and *Proustia cuneifolia* (Santa Gracia), *Aristeguietia salvia*, *Colliguaja odorifera*, *Cryptocarya alba*, and *Lithraea caustica* (La Campana), *Araucaria araucana*, *Nothofagus antarctica*, and *Chusquea coleu* (Nahuelbuta). From each sampled plant (n = 20), multiple samples of leaves, twigs and stem were collected, pooled together, and homogenized prior to analysis. These samples were either taken using an increment borer (stem samples) or plant scissors (leaf and twig samples) equipped with a telescopic arm to reach the higher parts of trees. As is commonly the case in field studies, roots could not be sampled in a representative manner though we account for their influence on plant composition (see Appendix A). The litter layer in La Campana and Nahuelbuta was also sampled.

**4.2 Analytical methods**

**4.2.1 Chemical composition of regolith and bedrock**

The concentration of major and trace elements in bedrock and regolith samples were determined using a X-Ray Fluorescence spectrometer (PANalytical AXIOS Advanced) at the section for "Inorganic and

**Margin annotations (tracked changes – "hat gelöscht" = deleted):**
- hat gelöscht: 3
- hat gelöscht: 3
- hat gelöscht: .
- hat gelöscht: has
- hat gelöscht: been
- hat gelöscht: either
- hat gelöscht: Oeser et al. (2018)
- hat gelöscht: ) from within the respective study sites.
- hat gelöscht: have been
- hat gelöscht: .
- hat gelöscht: stem-, twig-, and leaf-samples
- hat gelöscht: ), respectively.
- hat gelöscht: In addition,
- hat gelöscht: 3
- hat gelöscht: 3

Isotope Geochemistry", GFZ German Research Centre for Geosciences. A detailed description of the analytical protocols and sample preparation is given in Oeser et al. (2018).

**4.2.2 Chemical composition of vegetation**

Major and trace element concentrations of vegetation samples were determined using a Varian 720-ES axial ICP-OES at the Helmholtz Laboratory for the Geochemistry of the Earth Surface (HELGES), GFZ German Research Centre for Geosciences (von Blanckenburg et al., 2016) with relative uncertainties smaller than 10%. Prior to analysis, all samples were oven-dried at 120°C for 12 hrs. Subsequently, leaves were crushed and homogenized. About 0.5 g of leaf and 1 g of woody samples were digested in PFA vials using a microwave (MLS start) and ultra-pure concentrated acid mixtures comprising $H_2O_2$ and $HNO_3$, HCl and $HNO_3$, and HF. In some plant samples Si-bearing precipitates formed upon evaporation after digestion. These sample cakes were redissolved in a mixture of concentrated HF and $HNO_3$ to ensure complete dissolution of Si prior to analysis. As some Si might have been lost by volatilization as $SiF_4$ in this process, we do not include these samples (indicated by a * in Table S5) for the compilation of the plants' Si budget. With each sample batch, the international reference material NIST SRM 1515 Apple leaves and a procedural blank were processed.

**4.2.3 Extraction of the bio-available fraction and its chemical analyses**

The bio-available fraction of regolith samples was extracted using a sequential extraction procedure adapted from Arunachalam et al. (1996), He et al. (1995), and Tessier et al. (1979). The sequential extraction was performed in parallel on two regolith aliquots, and the supernatants were pooled together for analyses. About 2 g of dried and sieved (< 2 mm) sample material were immersed in 14 ml 18 MΩ Milli-Q $H_2O$ (water-soluble fraction) and then in 1M $NH_4Oac$ (exchangeable fraction; maintaining a sample/reactant ratio of ca. 1:7), and gently agitated. After each extraction, the mixture was centrifuged for 30 min at 4200 rpm and the supernatant was pipetted off. The remaining sample was then rinsed with 10 ml Milli-Q $H_2O$ and centrifuged again (4200 rpm, 30 min) and the rinse solution added to the supernatant. Subsequently, the supernatants were purified using a vacuum-driven filtration system (Millipore®; 0.2 μm acetate filter), evaporated to dryness, and redissolved with ultra-pure concentrated

Isotope Geochemistry", GFZ German Research Centre for Geosciences. A detailed description of the analytical protocols and sample preparation is given in Oeser et al. (2018).

**4.2.2 Chemical composition of vegetation**

Major and trace element concentrations of vegetation samples were determined using a Varian 720-ES axial ICP-OES at the Helmholtz Laboratory for the Geochemistry of the Earth Surface (HELGES), GFZ German Research Centre for Geosciences (von Blanckenburg et al., 2016) with relative uncertainties smaller than 10%. Prior to analysis, all samples were oven-dried at 120°C for 12 hrs. Subsequently, leaves were crushed and homogenized. About 0.5 g of leaf and 1 g of woody samples were digested in PFA vials using a microwave (MLS start) and ultra-pure concentrated acid mixtures comprising $H_2O_2$ and $HNO_3$, HCl and $HNO_3$, and HF. In some plant samples Si-bearing precipitates formed upon evaporation after digestion. These sample cakes were redissolved in a mixture of concentrated HF and $HNO_3$ to ensure complete dissolution of Si prior to analysis. As some Si might have been lost by volatilization as $SiF_4$ in this process, we do not include these samples (indicated by a * in Table S5) for the compilation of the plants' Si budget. With each sample batch, the international reference material NIST SRM 1515 Apple leaves and a procedural blank were processed.

**4.2.3 Extraction of the bio-available fraction and its chemical analyses**

The bio-available fraction of regolith samples was extracted using a sequential extraction procedure adapted from Arunachalam et al. (1996), He et al. (1995), and Tessier et al. (1979). The sequential extraction was performed in parallel on two regolith aliquots, and the supernatants were pooled together for analyses. About 2 g of dried and sieved (< 2 mm) sample material were immersed in 14 ml 18 MΩ Milli-Q $H_2O$ (water-soluble fraction) and then in 1M $NH_4Oac$ (exchangeable fraction; maintaining a sample/reactant ratio of ca. 1:7), and gently agitated. After each extraction, the mixture was centrifuged for 30 min at 4200 rpm and the supernatant was pipetted off. The remaining sample was then rinsed with 10 ml Milli-Q $H_2O$ and centrifuged again (4200 rpm, 30 min) and the rinse solution added to the supernatant. Subsequently, the supernatants were purified using a vacuum-driven filtration system (Millipore®; 0.2 μm acetate filter), evaporated to dryness, and redissolved with ultra-pure concentrated
* * *
**Margin comments (tracked changes — "hat gelöscht" = "deleted"):**

- hat gelöscht: 3
- hat gelöscht: (von Blanckenburg et al., 2016)
- hat gelöscht: redissoled
- hat gelöscht: evaporation
- hat gelöscht: have
- hat gelöscht: included
- hat gelöscht: plant
- hat gelöscht: was
- hat gelöscht: 3
- hat gelöscht: Arunachalam et al. (1996)
- hat gelöscht: He et al. (1995)
- hat gelöscht: Tessier et al. (1979)
- hat gelöscht: of which
- hat gelöscht: either
- hat gelöscht: or
- hat gelöscht: ), thus
- hat gelöscht: to
- hat gelöscht: about
- hat gelöscht: . Each extraction step was performed with mild agitation…
- hat gelöscht: ). This
- hat gelöscht: was

[revised manuscript text omitted]

**hat gelöscht:** Taylor, L. L.,

**hat gelöscht:** Weathering by tree-root-associating

**hat gelöscht:** diminishes under simulated Cenozoic atmospheric CO2 decline, Biogeosciences, 11, 321-331,

**hat gelöscht:** .5194/bg-11-321-

**hat gelöscht:** Redfield, A. C.: On the proportions of organic derivatives in sea water and their relation to the composition of plankton, James Johnstone memorial volume, 176-192, 1934.¶

**hat gelöscht:** Rosenstock, N. P.: Can ectomycorrhizal weathering activity respond to host nutrient demands?, Fungal Biology Reviews, 23, 107-114, 10.1016/j.fbr.2009.11.003, 2009.¶ Rosling, A., Lindahl, B. r. D., Taylor, A. F. S., and Finlay, R. D.: Mycelial growth and substrate acidification of ectomycorrhizal fungi in response to different minerals, FEMS Microbiology Ecology, 47, 31-37, 10.1016/s0168-6496(03)00022-8, 2004.¶

**hat gelöscht:** Sardans

**hat gelöscht:** Rivas-Ubach

**hat gelöscht:** Peñuelas,

**hat gelöscht:** The elemental stoichiometry of aquatic

[revised manuscript text omitted]

---

## Author Response (AR2)

To: Prof. Jack Middelburg

Dear Jack,

Please find enclosed our revised manuscript entitled **"Do degree and rate of silicate weathering depend on plant productivity?"** by Oeser and von Blanckenburg

We would like to thank you and the reviewers for the very constructive reviews, allowing us to improve our manuscript and better readable for a wider audience.

In this revision we followed many of the reviewer's comments but not all as some of the structure and content was requested by the other reviewers. Below you will find the point by point replies to the comments and suggestions of the reviewer.

Despite the hard work for you, the reviewers, and for us, we enjoyed the discussion process at Biogeosciences which we hope in this case resulted in a hopefully though-provoking, yet acceptable version for an interdisciplinary readership. We would like to express our thanks for guiding us through this process.

In the name of Friedhelm von Blanckenburg and Ralf A. Oeser, yours sincerely,

Ralf

I had already reviewed the first version of the manuscript and appreciate the complete overhaul, which rendered the paper much clearer. However, I have still concerns with respect to (i) the unusual use of soil scientific terms, (ii) partly sloppy and even wrong use of plant nutritional terminology, (iii) the statistical data treatment, and (iv) the general structure of the paper, which is still overly long for its message. I regret that the authors seem to be resistant against some of the reviewer comments without presenting arguments that I find convincing.

REPLY: We are grateful nevertheless for the comments from this reviewer and the other three reviewers. They considerably helped to shape this paper, that was written from the view point of Geochemistry, but will hopefully be read by readers from other fields, for example ecologists and soil scientists. This task involved a whole range of issues, including time scales, terminology, and general scientific perspective. In this revision we followed many of the reviewer comments – but not all, as some of the structure and contents was requested by the other reviewers, whereas other content we deem necessary to make this work accessible to an interdisciplinary audience.

**i.** I think that the use of saprolite is not consistent with its usual meaning in Soil Science. Saprolite is the lowermost part of the deep, strongly chemically weathered soil in the humid inner tropics where desilication prevails. Saprolites are characterized by their intensive degree of weathering while at the same time the original structure is maintained because of the location under tens of meters of weathered material. Saprolites surround Woolsacks and occur as relics of the Tertiary at erosion-protected locations in temperate soils but according to the soil description at your study sites in Chile, there are no saprolites. As a consequence, the use of the term regolith does not seem to be necessary, because there is no part of the weathering mantle which is free of living organisms. What you are studying is the soil with A, B, C and perhaps even R horizons. I understand what you mean by "weathering front" when referring to deep cracks in the granitoid rocks, where initial weathering indeed takes place but in Soil Science the weathering front is usually seen at the contact interface between soil and bedrock, i.e. between the Cw and the (nearly) unweathered rock (i.e. R horizon). The latter does not need to be changed but it should be clear what you mean.

REPLY: We acknowledge that the use of certain terms differs between scientific fields. However, the terminology used in this manuscript is in line with a large number of geomorphic

articles and in particular with the article from Bernhard et al. (2018) on which the classification of the regolith profiles in the EarthShape study sites is based on. In order to be compatible with the other EarthShape papers and other geomorphic papers we retain the terminology used.

**ii.** Although the authors removed the part that I most criticized in my past review, there are still several issues with the plant nutritional terminology. First of all, it must be clearly recognized that the most limiting nutrient is N, which is not rock-derived. The demand for the rock-derived nutrients follow the N supply in little variable stoichiometric ratios. All essential nutrients are required at the same time. A common differentiation would be to distinguish plant mineral macronutrients (N, P, K, Ca, Mg, S) from micronutrients (Fe, B, Cl, Mn, Zn, Cu, Mo, Ni). It is problematic to generally include Al in the list of plant beneficial elements because for most plants Al is not beneficial but even more likely toxic. Moreover, as I already stated in my previous review, nutrient concentrations are mostly regulated in plants so that their concentrations vary by much less than a factor of 10. One of the most extreme cases is the comparison of straw with grain of wheat where the straw contains 5% N and the grain 20%, i.e. a variation by a factor of four. If the whole plant is considered, the variation of nutrient concentrations is usually even smaller. The same is true for element ratios. Consequently, if there is a deviation of a factor of 10 of element ratios between the so-called plant-available nutrient pool in soil and the plants, then this is clearly dissimilar. Moreover, it is not easily possible to characterize plant availability with a single extract, because a single extract does not consider the kinetic replenishment of nutrients after plants have taken up a nutrient and because the extract cannot mimick nutrient acquisition strategies like enhanced mineralization of organic matter by exoenzymes or local acidification by roots and mycorrhizae. This is particularly true for P.

REPLY: Based on the previous reviews we decided to remove any reference to nutrient limitation in the text, including the role of N. Following the reviewers' request in this revised version we introduce the general importance of N to plant nutrition despite not being rock-derived. We further decided to remove any reference on nutrient classification and have erased the terms 'plant-beneficial' and 'plant-essential' throughout the manuscript, figures, and tables accordingly.

We agree that sequential extractions do not mimic nutrient acquisition strategies by plants. Yet estimating plant available nutrients using sequential extraction methods is common

practice in geosciences (e.g. Brucker and Spohn, 2019; Bullen and Chadwick, 2016; Lang et al., 2017). We are not aware of other possibilities to determine plant availability within a reasonable effort.

**iii.** In principle, I find the approach to single out biological weathering with a statistical approach great. However, I see problems in the small number of independent data you have, i.e. n = 4. You are not transparent about the number of data considered in your analysis and I suspect that you have included more the four data pairs. If one considered the two soil profiles on the different slopes as independent from each other – which is debatable – you had n = 8. (But I would not recommend to do this.)

REPLY: We regret that our statistical data set is not sufficient in the reviewer's eyes. However, we cannot overcome these limitations by any means. Conducting this extensive analysis on even eight profiles represented a formidable effort- In order to address the concerns of the reviewer, we will tone down the corresponding text passages (i.e. removal of direct comparison of r-values as suggested by the reviewer).

We performed these statistical analyses on the basis that the two soil profiles at each site that are considered to be independent from each other. Further we did report p-values in APA style including the degree of freedoms which are directly linked to the independent variables. We believe that this approach was thus completely 'statistically honest'. In the revised version of the manuscript and tables we however explicitly mention the number of statistically independent variables instead.

**iv.** I think that the conceptual perspectives should be included into the introduction to avoid references to later sections such as in l. 78-79 and 91-94 and because you present your methods before you justify them. I also think that the introduction would become more concise (alone deleting the references to later section saves 5 lines). I therefore strongly suggest to combine and shorten Sections 1 and 2. Moreover, all results are still presented threefold, i.e. in tables, figures, and as numbers in the text. I had already previously suggested that the tables should be moved to the appendix and the numbers in the text deleted, while the figures should be kept and I still insist on this. This could be a really interesting paper. However, it is still cumbersome to read, because it is not sufficiently concise.

REPLY: After the first round of open discussion, we identified several priority revision items. One of which was the need for a section dedicated to the different concepts used in soil- and geosciences to characterize weathering profiles. We would prefer to retain 'Section 2

Conceptual perspectives' to make our concept of weathering and ecosystem nutrition accessible to a broader audience including soil scientists.

Another revision item that was raised by anonymous referee #1. The referee asked for a more rigorous presentation of uncertainties within the text. Therefore, we added uncertainties to the text which we retained in this version. However, to shorten the manuscript and address this referees' concern we suggest transferring the former Tables 4, 6, and 8 into the Appendix. They now become Table A1, A2, and A3.

We will be happy to receive guidelines from the editor on how to proceed in this respect.

**Minor comments:**

**l. 11, 13, 167:** Delete "substantial". If the work was not substantial, it should not be published.

REPLY: Done accordingly; the first "substantial" (describing the climate gradient) replaced by "major"

**l. 82-83:** Move to acknowledgments.

REPLY: Done accordingly

**l. 90:** Replace "nutritive element" by "mineral nutrient".

REPLY: Done accordingly

**l. 97-98:** Delete. At this point, this is an unsubstantiated claim and moreover a repetition from the abstract.

REPLY: Done accordingly

**l. 125-127:** This is not true. In many terrestrial ecosystem, biomass production is limited by N which is rarely (only in bituminous sedimentites) rock-derived but acquired by biological N2 fixation from the air.

REPLY: We agree with the reviewer with regards to N limitation in many terrestrial ecosystems as is the case in the EarthShape sites. We replaced "limitation" by "supply (of a mineral nutrient) needed to sustain an ecosystem over weathering time scales". Furthermore, we suggest adding the following text here: "Note that nitrogen (N), the most limiting nutrient in many ecosystems, is not an element addressed here. Although rocks have recently received attention as source of geogenic N (Houlton et al. 2018) this source is most prominent in sedimentary rock. This study explores ecosystems developed on granitoid rock where N is derived from the inexhaustible atmospheric pool by nitrogen-fixing bacteria, and limitation mostly arises by the energy required for fixation (Chapin III et al, 2011).

**l. 127-129:** I think that this is a too far-reaching generalization. There are e.g., biodiversity hotspots in mountainous areas, which are not particularly nutrient-poor.

REPLY: We reworded to say that supply by a specific mineral nutrient impacts plant diversity and nutrient acquisition strategies, for which there is ample evidence in the literature, even though of course mineral nutrient limitation is not an exclusive condition for biodiversity hotspots.

**l. 156:** The "geogenic nutrient pathway" rarely influences nutrient limitation in native ecosystems. I can only imagine that this would be the case if K was growth limiting. Even the P supply is governed by mineral dissolution and sorption-desorption equilibria, organic matter mineralization and biological acquisition strategies such as local acidification by plant roots and mycorrhiza. Of these processes, only mineral dissolution could be counted as weathering process but mostly of secondary minerals. In strongly weathered humid inner tropical ecosystems it is thought that the small losses of nutrients escaping from the close ecosystem cycling are replenished by deposition, while weathering does not play any role.

REPLY: We do not entirely understand the reviewer's comment. The loss of elements from ecosystems, amongst them the mineral nutrients through erosion and leaching has been documented by many studies, in particular for sloping landscapes. In ecosystems where atmospheric deposition does not suffice to balance these losses, nutrients must be replenished through weathering and this replenishment occurs over millennia. The reviewer refers to strongly depleted ecosystems in the tropics. In these erosion rates are way lower than in the EarthShape sites and the losses can indeed be compensated by atmospheric deposition (e.g. Arvin et al., 2017; Boy and Wilcke, 2008; Schuessler et al., 2018), even though the main nutrient source is recycling. These pathways (the geogenic nutrient pathway and the orgaic nutrient pathway were recently related to each other in a paper from our group (Uhlig and von Blanckenburg, 2019) and here we build on this conceptual framework.

**l. 166:** Start a new sentence after the references.

REPLY: Done accordingly.

**l. 171:** Mediterranean (with upper scale M).

REPLY: As mediterranean does not refer to the European region but is rather used as an adjective, we will retain the "M" as lower-case letter.

**l. 178** (and in the whole paper): Replace "gC" by g m-2 yr-1 C.

REPLY: Done accordingly.

**l. 229-230:** Umbric Podzols, Orthodystric Umbrisols (mind the spelling including upper case letters).

REPLY: Done accordingly

**l. 230:** Do you mean an organic layer? Usually, there are several organic horizons (Oi, Oe, Oa).

REPLY: We mean the organic layer as we did not differentiate the single organic horizons. Changed accordingly.

**l. 260:** Delete "As is commonly the case in field studies", because this is not true. There are many field studies on root distribution even in forests.

REPLY: Deleted accordingly.

**l. 262:** What is the "litter layer"? Only the Oi horizon, i.e. the freshly fallen litter of the same or perhaps two years? Or the whole organic layer? Use consistent terminology.

REPLY: The litter layer comprised small woody debris and leaves fallen within the last two years. We amended the sentence accordingly.

**l. 281:** Where are the results of these quality controls?

REPLY: All data (including analysis of reference materials) are reported in an open access data repository (Oeser and von Blanckenburg, 2020). We mention the existence of this data supplement in the text.

**l. 286** (and in the whole paper): The correct SI unit is mL (with upper case L).

REPLY: This journal asks to follow the recommendations of the SI and IUPAC. According to SI, both terms ml and mL are correct. Hence, we will retain ml throughout the manuscript.

**l. 287** (and in the whole paper): Milli-Q is a brand name, which should not be used. Replace by "deionized water".

REPLYY: Done accordingly

**l. 290 and l. 303:** "rounds per minute" are meaningless without the knowledge of the geometry of your centrifuge or shaker. Delete or replace by g.

REPLY: Although we partially agree with reviewer we still prefer to retain the rpm as most lab centrifuges are similar and the reader thus gets a feel for the method.

**l. 351:** P and K are not the "most important nutrients" in a N-limited system like yours. It is N, and N supply does not depend on weathering at your study sites. This must be acknowledged and discussed.

REPLY: We are a little puzzled by this comment. We do explicitly use the term 'mineral nutrients' and although N can be derived though geogenic sources (Houlton et al., 2018), we

did not aim to include nitrogen into that group. However, we will highlight the particular role of N on gross primary production in line 133ff (see above).

**l. 354:** It is wrong to consider Al as a generally beneficial element for plants, only for a few and only at low concentrations. In Marschner (2012) it is stated: "Aluminum is beneficial to some plants, such as tea, and may alleviate proton toxicity and increase the activity of antioxidant enzymes."

REPLY: We did tone down this paragraph accordingly and removed any reference to plant-essential or plant-beneficial elements from the text and figures (see previous comment to point ii).

**l. 355:** It is wrong to include the micronutrient Fe into the group of beneficial elements. It does not make sense to refer to an arbitrary selection of nutrients as "plant-essential elements".

REPLY: Please see our comment above.

**l. 356:** It is wrong to consider Sr as a nutrient. It is at best a surrogate. However, its metabolism in plants differs from that of Ca for which it is used as surrogate (see e.g., Blum et al., 2012, Plant Soil 356, 303-314).

REPLY: Please see our comment above.

**l. 364:** I do not understand what you mean by "the shallowest mineral soil"?

REPLY: We mean the subsoil. Changed accordingly.

**l. 382:** element-specific

REPLY: Changed accordingly.

**l. 395:** The total stock of elements would include the organic layer, which plays a particular important role for plant nutrition, where present.

REPLY: The reviewer is right, elements contained in the organic layer would indeed contribute to the total stock of nutrients available. However, their total stock might be small compared to the mineral nutrients contained in bulk regolith. This is the theoretically maximum available amount of nutrients available. We changed the text to 'The maximum amount of mineral nutrients...'.

**l. 403:** If you just sum up the stocks of macro and micronutrients the result will be (almost) identical to that for the macronutrients. I do hard in seeing an additional value in this metric.

REPLY: Wo do not understand the reviewer's comment in that context. We believe it is worth mentioning that the total inventory of bio-available nutrients is highest in the arid site (where

demand in terms of NPP is lowest among the four sites) but lowest in the high-NPP site of Nahuelbuta. We discuss this pattern later in the manuscript.

**l. 433-435:** This is an example where numbers are mentioned in the text, which are additionally shown in Table 3 and Figure 4. It is sufficient to show the data once (preferably in the figure).

REPLY: Please see our comment to the referee's point iv. We are happy to receive guidelines from the editor on how to proceed.

**l. 448:** Is such a small difference really significant?

REPLY: In the framework of (radiogenic) isotope geochemistry even smaller differences are significant when resolvable. Resolution is dependent on the method applied but commonly variations on the 5$^{th}$ decimal are easily detectable.

**l. 481:** I believe that your estimate of the contribution of sea spray is not correct. At the short distance of 80 km to the ocean there must be a visible input of sea spray and this input should increase with increasing surface area of the canopy, i.e. from N to S. Couldn't the deviation of the Sr isotope ratio in the so-called bioavailable pool be caused by incorporation of Sr deposited from the sea? What is the Sr isotope ratio of sea spray? – I mentioned this already in my previous review and am not satisfied by the answer.

REPLY: We still do not entirely understand this comment. We have determined atmospheric contribution relative to weathering input from Sr isotopes using a two-component mixing calculation using bedrock and seaspray as endmembers. This is a commonly used approach. Accordingly, up to 93 and 43% of Sr derived from seaspray are incorporated in the regolith profiles of Pan de Azúcar and Santa Gracia, respectively. No such an input was found in the other two sites. Although we could not detect seaspray input in these sites, that does not mean this input is not existent. However, the **flux** of Sr derived from weathering likely exceeds the seaspray flux by orders of magnitude such that we cannot resolve this incorporation. Sr seaspray ratio is mentioned in Table A2 and Figure 5.

**l. 489:** This is pure speculation.

REPLY: The referee is right inasmuch as we have no direct mineral-specific measurements of $^{87}Sr/^{86}Sr$. However, we have no indications of a significant incorporation of external sources into the regolith profiles of La Campana and Nahuelbuta (neither $^{87}Sr/^{86}Sr$ nor τ-values would permit such a conclusion). If not from weathering of rock, what is the Sr-source? We will tone down this sentence.

**l. 495:** The most-demanded (in terms of limitation and quantitatively) is N, not P.

REPLY: Please see our previous comments on N. We do not regard N as a mineral nutrient simply because the majority of the N stem from Earth's largest N inventory, the atmosphere and is not limited by weathering rather than fixation. We addressed this misunderstanding by adding 'rock-derived'.

**l. 501:** One order of magnitude is more than nutrient concentrations and ratios in plants vary.

REPLY: We do not understand this comment. We found that with increasing NPP (and from saprolite to soil) the X:P ratio between the bio-available pool and plants increasingly approach the 1:1 line. We interpret these shifts as an increasing signature of plant material in the bio-available fraction, caused by higher recycling rates. We omitted making any further conclusions on plant stoichiometrical pattern.

**l. 523:** A large part of the nutrient recycling occurs via the organic layer which is not considered here. This could at least be acknowledged in the discussion.

REPLY: We acknowledge the loci of recycling by specifically mentioning organic bearing soil horizons: 'In the remaining sites $Rec^X$ **occurs in all organic-bearing soil horizons** and increases…'

**l. 524-525:** The plants take up nutrients from greater depth, primarily because they only find water there. Without water, there is no nutrient uptake via the roots.

REPLY: Deep nutrient uptake as nutrient acquisition strategy has only recently been shown by Uhlig et al. (2020). However, in order to account for the reviewer's concern, we changed the sentence into '…forage nutrients and water by deep…'.

**l. 561:** Organo-Al complexes are not bioavailable (and non-toxic).

REPLY: The wording is indeed a little bit confusing. We therefore change 'available' to 'abundant'

**l. 567:** This "dilution effect" is called "leaching" in Soil Science. **l. 570:** It is the pH value and leaching.

REPLY: We mean the "dilution effect" of solutes in water, as described in concentration-discharge relationships.

**l. 595:** Weathering only needs to replace the loss from plant nutrient cycling, not supply the whole required plant nutrients.

REPLY: This is exactly at the heart of the "geogenic nutrient pathay" - replacing the loss from recycling.

**l. 636-641:** What is the number of replicates? You need to mind that these replicates need to be statistically independent from each other. You need to add this number to Table A1. The maximum number of such data you have is in my view n = 4 (see above).

REPLY: Please see our response to reviewer's point iii. We treat the two regolith profiles at each site as natural (hence independent) replicates. In our view, n = 8. In Appendix B and the statistical tables, we now clearly state the number of replicates.

**l. 667-669:** Given the low number of independent correlation pairs, I would refrain from such a quantitative comparison of r values.

REPLY: We did refrain from mentioning r values in the text. Instead we only reference to the Table(s).

**l. 685:** How was this tested? Again, what was your number of replicates per site? Two?

REPLY: The description of the statistical procedures can be found in Appendix B. We tested for statistically significant differences in weathering pattern using Tukey's HSD test. The number of replicates per site is two.

**l. 754-756:** This is typical for the weathering regime of temperate soils where Si released from weathering is not leached but forms secondary minerals such as clay minerals. Thus nothing new.

REPLY: Please to see that interpretations from soil sciences and weathering geochemistry converge.

**l. 769:** Even though/Although

REPLY: Changed accordingly.

**Table 2, Eq. (5):** I think that you need to exchange numerator and denominator.

REPLY: We believe the equation is correct.

Arvin, L.J., Riebe, C.S., Aciego, S.M., Blakowski, M.A., 2017. Global patterns of dust and bedrock nutrient supply to montane ecosystems. Science Advances, 3: 1-10.

[revised manuscript text omitted]

**Seite 52: [1] hat gelöscht**  author  28.08.20 16:09:00

**Seite 52: [2] hat gelöscht**  author  28.08.20 16:09:00

**Seite 52: [2] hat gelöscht**  author  28.08.20 16:09:00